# Role of sea spray aerosol at the air-sea interface in transporting aromatic acids to the atmosphere

Yaru Song, Jianlong Li, Narcisse Tsona Tchinda, Kun Li, and Lin Du*

Qingdao Key Laboratory for Prevention and Control of Atmospheric Pollution in Coastal Cities, Environment Research Institute, Shandong University, Qingdao, 266237, China

*Correspondence to*: Lin Du (lindu@sdu.edu.cn)

**Abstract**

Aromatic acids are ubiquitous in seawater (SW) and can be transported to the atmosphere via sea spray aerosol (SSA). Despite their importance in affecting the global radiative balance, the contribution of marine aromatic acids and their transport mechanisms through SSA remain unclear. Herein, the distribution of particle size and number concentration of SSA produced in SW containing nine different aromatic acids (i.e., benzoic acids,

benzenedicarboxylic acids, hydroxybenzoic acids, vanillic acid, and syringic acid) was studied using a custom-made SSA simulation chamber; moreover, the enrichment of aromatic acids in SSA and their emission flux to the atmosphere were analyzed. Transmission electron microscopy (TEM) images clearly revealed that aromatic acids can be transferred to the nascent SSA. Interestingly, the morphology associated with benzenedicarboxylic acids-coated particles showed that aromatic acids can promote the growth of other surfaces of sea salt, thus making the

sea salt core spherical. Aromatic acids showed a significant enrichment behavior at the air-sea interface, which clearly indicated that SSA represent a source of aromatic acids in the atmosphere. Vanillic acid had the largest global emission flux through SSA (962 tons $yr^{-1}$), even though its concentration in SW was lower. The calculated results indicated that the global annual flux of aromatic acids was not only affected by the concentration in SW, but also by their enrichment factor (EF). These data are critical for further quantifying the contribution of organic

acids to the atmosphere via SSA, which may provide an estimate of the potential influence of the atmospheric feedbacks to the ocean carbon cycle.

## 1. Introduction

Aromatic acids are considered to be important environmental pollutants due to their potential toxicity and persistence (Zhao et al., 2019). They are ubiquitous in seawater (SW) and the atmosphere, and have an important influence on the marine and atmospheric environment. On the one hand, aromatic acids can readily be taken up by marine organisms, leading to their enrichment within these organisms or transportation to remote areas (Fu et al., 2011; Shariati et al., 2021; Wang and Kawamura, 2006). This process poses health risks to the endocrine system of aquatic organisms and the overall marine ecosystem (Saha et al., 2006). Other than the influence on marine systems, they have also been reported to have negative impacts on human health, bioaccumulation, environmental persistence, and climate change (Lu et al., 2021).

Sea spray aerosol (SSA) may contain organic acids that play a major role in Earth's climate. Previous studies suggested that these organic acids can alter the composition of SSA, subsequently influencing atmospheric processes such as cloud condensation nuclei (CCN) or ice nuclei (IN) activities (Moore et al., 2011; Zhu et al., 2019). In addition, organic acids, such as carboxylic acids, aromatic acids also contribute significantly to ocean acidification (Kumari et al., 2022). Therefore, due to their importance in upper ocean biogeochemistry and their environmental risk, the transport processes and accumulation potential of aromatic acids at the air-sea interface require a comprehensive understanding.

In various observations, aromatic acids have been detected in both natural and anthropogenic sources (Zhao et al., 2019; Zangrando et al., 2019; Dekiff et al., 2014). Many studies have confirmed that aromatic acids cannot only be produced by biological processes but also be considered as potential sources of oil spill, marine plastics, and sunscreen product discharges (Kristensen et al., 2021; Aitken et al., 2018). An important source of marine aromatic acid is the acidic degradation of spilled oil that occurs extensively in human-impacted areas, including coastal oceans and harbors (Lu et al., 2021). Importantly, in addition to biodegradation, the identification of phthalic acid in organisms raises the possibility of its origin from the ingestion of marine plastics (Almulhim et al., 2022). This is consistent with previous studies that phthalic acid is primarily derived from anthropogenic sources, such as plasticizer, biomass burning and fossil fuel combustion (Boreddy et al., 2022; Ding et al., 2021; Ren et al., 2023; Sanjuan et al., 2023; Shumilina et al., 2023; Yang et al., 2020; Zhu et al., 2022), whereas hydroxybenzoic acid has both anthropogenic and natural sources (Castillo et al., 2023; Liao et al., 2019; Lu et al., 2021; Lu et al., 2023; Zhao et al., 2019).

Recent laboratory studies have shown that personal-care products in SW, especially sunscreen (e.g., *o*-hydroxybenzoic acid), are reduced in levels during algal blooms (Franklin et al., 2022). This phenomenon has prompted discussions about the concept of "missing aromatic acids". Nevertheless, the currently available data do not provide a conclusive explanation for the existence of these "missing aromatic acids". One possible reason for the disappearance of these aromatic acids is their release into the atmosphere. The existing datasets obtained from remote marine areas offer evidence of the presence of these compounds in the atmosphere (Fu et al., 2010). Although recent studies outline the critical sources of aromatic acids, much less is known about its transportation via SSA (Castillo et al., 2023; Hu et al., 2022).

SSAs, generated by breaking waves and bubble bursting, are among the major sources of atmospheric particles (Andreae and Rosenfeld, 2008; Angle et al., 2021; Hasenecz et al., 2020; Malfatti et al., 2019). SSAs can act as carrier agents for the vertical transport of much more than just sea salt and often include organic surfactants in the ocean, as already shown by field and laboratory studies (Cochran et al., 2016; Franklin et al., 2022; Rastelli et al.,

2017).

Recent data indicate that the surface activity and octanol-water partitioning coefficients ($K_{ow}$) of organic compounds may affect their transport efficiency from the water phase to the atmosphere (Olson et al., 2020). Moreover, the molecular structure may induce changes in organic acids' properties (i.e., surface tension, toxicity), which further affect their transport potential and global emission flux (Lee et al., 2017; Rastelli et al., 2017; Frossard et al., 2019; Acker et al., 2021a). While field studies have demonstrated the presence of aromatic acids in both the ocean and atmosphere (Boreddy et al., 2017), the specific mechanisms influencing their transport at the air-sea interface require further investigation. Therefore, understanding the factors influencing the enrichment behavior of aromatic acids at the air-sea interface and determining the emission flux of aromatic acids is crucial.

In a previous study, we observed that the transfer of short-chain organic acids between the air and sea through SSA may be influenced by seawater surface tension. This factor could, in turn, impact the enrichment behavior of organic acids (Song et al., 2022). However, extrapolation of these results to all of the organic acids may not be warranted as the mechanism for enrichment behavior may be closely related to the molecular structure.

In this study, we systematically identified the transport process of aromatic acids from SSA to the atmosphere in the natural seawater environment. To this end, we developed a sea spray aerosol simulation chamber using the plunging jet method, providing the closest proxy to natural SSA currently available in a controlled laboratory monitoring environment, and used it to probe the transport process of aromatic acids. In our SSA simulation chamber, SW and SSA samples were concurrently analyzed for their aromatic acid concentrations, allowing better characterization of any enrichment processes involved in bubble bursting. Finally, the global emission fluxes of aromatic acids via SSA were assessed by combining data on the concentration of aromatic acids in SW and experimental enrichment factor data, which provides unique insights into the enrichment behavior and atmospheric transport of aromatic acids-containing SSA particles, particularly at the air-sea interface.

## 2. Experimental Section

### 2.1 Materials

Aromatic acids were purchased from Shanghai Aladdin Bio-Chem Technology, China. Nine aromatic acids, including benzoic acid, phthalic acids (*o*-, *m*-, and *p*-isomers), hydroxybenzoic acids (*o*-, *m*-, and *p*-isomers), vanillic acid, and syringic acid, were investigated to determine the influence of functional group position and quantity on the transmission of aromatic acids at the sea-air interface (Table 1). Further details are provided in Table S1, which lists sources and concentrations of these aromatic acids identified in SW and atmospheric samples over the ocean. High performance liquid chromatography (HPLC) grade methanol and acetonitrile were supplied by Fisher Scientific, USA. Artificial seawater (ASW) was supplied by Psaitong, China. SW collected from the coast of the Yellow Sea in Qingdao, China, was transported into the laboratory SSA simulation chamber (see Fig. S1). The range of pH values measured by a pH meter (PHS-3C, Shanghai Yidian Scientific Instrument, China) in all experiments was 7.58–7.92. Ultrapure water with a resistivity of 18.2 MΩ cm was generated by a Milli-Q purification system (Merck Millipore, France), which can be used to prepare mobile phases for liquid chromatography. Each aromatic acid was first sonicated in natural seawater for 30 min at a concentration of 1 mM. For simplified consideration, aromatic acids were added separately to ASW to achieve concentrations of 1 μM and 1 mM, in order to verify the effects of background systems and concentrations on the EF of aromatic acids, which may not accurately reflect realistic conditions but provide an approximated trend of EFs instead.

## 2.2 Experimental setup

A jet-based laboratory SSA simulation chamber, made of 316L stainless steel material, was used to mimic the SSA generation (Fig.S2). This chamber was a clamshell cuboid box (length 30 cm, width 20 cm, height 40 cm) with a viewable glass window, which has recently been adapted for air-sea transport process studies for continuously generated plunging jets to generate realistic SSA (Liu et al., 2022; Xu et al., 2023; Zhan et al., 2022). The detailed dimensions of the SSA simulation chamber are provided in Table S2. All the enrichment experiments were conducted in the SSA simulation chamber filled with approximately 9 L of SW. When used, plunging jets were cycled in SW by the pump (Shenchen V6-6, China) at a flow rate of 1 L min$^{-1}$ through a stainless-steel nozzle (inner diameter 4.3 mm). Then the bubble breaking on the surface of the SW was observed through the glass window attached to the SSA simulation chamber. During the experiment, it was assumed that the interaction of the bubble plume generated by the plunging jet with the wall was negligible (Fig. S3).

To avoid any contamination from indoor air, particle free air was supplied by a compressor and zero-air generator (model 111, Thermo Scientific, USA). A mass flow controller (Beijing Sevenstar Electronics, China) was used to adjust the air flow rate entering the SSA simulation chamber, with a flow rate in the range of 3–50 L min$^{-1}$. The outlet was fitted with a three-way valve to transfer the sample airflow and the excess gas. The Nafion drying tube (MD-700-06S-3, Perma Pure, USA) is used to control the relative humidity of the aerosol particles at the sampling port. The relative humidity and temperature were monitored by a 2-channel thermo-hygrometer (Testo HM42, Vaisala, Finland), in the ranges of 30–40% and 20–25 °C, respectively. Surface tension was measured with approximately 20 mL of SW samples. The tensiometer (JK99C, Powereach, China) was calibrated with 20 mL of ultrapure water at 25 °C and the sheet metal was cleaned with ethanol between each measurement. The surface tension represents collections of at least three independent measurements for each aromatic acid in order to guarantee data reproducibility.

## 2.3 Sample collection and instrumental analysis

The experiment consisted of a total of 25 sets with target compound concentrations of 10$^{-3}$ and 1 mM (Table S3). Each aromatic acid was added individually to SW and ASW for the experiment. Particle size distributions were measured at a relative humidity of 34.2±3.9% using a scanning mobility particle sizer (SMPS, TSI, Model 3936). The SMPS consisted of a differential mobility analyzer (DMA, Model 3081, TSI, USA) coupled to a condensation particle counter (CPC, Model 3776, TSI, USA), which measured particles with diameters between 14.1 and 710.5 nm. The SMPS measured particle number concentration and geometric mean diameter (GMD) within the SSA simulation chamber at a time resolution of 3 min, with an inlet and sheath gas flow rate of 0.3 L min$^{-1}$ and 3.0 L min$^{-1}$, respectively. Before turning on the pump, we confirmed that the particle number concentration was < 20 cm$^{-3}$ during the first 30 min of each experiment to ensure no leaking and no background particles. Each group of number concentration monitoring experiments lasted approximately 1 h. After the size distribution stabilized, a total of 40 mL of SW is collected from a tap located 1.5 cm above the bottom of the chamber.

SSA samples were collected on an aluminum foil (25 mm, Jowin Technology, China) with a 14-stage Dekati low-pressure impactor (DLPI+, DeKati, Finland) for 5 h, and then stored at -20 °C until analysis. All samples were analyzed by attenuated total internal reflectance Fourier transform infrared (ATR-FTIR) spectroscopy (Vertex 70, Bruker, Germany) using an automated fitting algorithm and techniques described in a previous study

(Xu et al., 2021). In the range of 4000–600 cm$^{-1}$, the ATR-FTIR spectra of SSA particles were recorded with an average of 64 scans. SW sample was collected as the ATR-FTIR spectra of blank aluminum foil to confirm that there was no infrared absorption of target functional groups on the aluminum foil.

In addition, laboratory SSA transmission electron microscopy (TEM) samples were collected using a single particle sampler (DKL-2, Genstar Electronic Technology, China). TEM was performed using a TEM cryo-mount (Gatan 626) to load the samples, where the TEM grid was immersed in liquid nitrogen and then mounted on the holder by means of a cryo-transfer workstation. TEM with a high-angle annular-dark-field detector was used and then TEM images were obtained at an accelerating voltage of 200 kV. The SSA particles impacted onto copper grids films (T11023, Tianld, China) with a flow rate 1 L min$^{-1}$ for 1 h. TEM images were used to characterize the morphology of coated and uncoated SSA samples.

Submicron SSA particles were impacted onto 25-mm diameter quartz fiber filters (QFF, 1851–025, Waterman, UK), which were baked in a muffle furnace at 450 ℃ for 3 h. Filters collected on stages 6–10 (0.19–0.94 μm) of low-pressure impactor were extracted with 4 mL ultrapure water under ultrasonication for 30 min. Thermo scientific series of high performance liquid chromatography (HPLC, Vanquish, Germany) coupled with Atlantis C18 reversed phase column (2.1 mm × 150 mm, 3 μm particle size, 100 Å, Waters) at 30 ℃ and mass spectrum was performed to determine the concentration of aromatic acids in SSA and SW samples. The mobile phase consisted of 0.1% formic acid aqueous solution (A) and acetonitrile, and was kept at a flow rate of 0.2 mL min$^{-1}$. An eluent gradient program was used as follows (Witkowski and Gierczak, 2017): starting with 5% A for 5 min, increasing A from 5% to 25% in 2 min, keeping A at 25% for 4 min, increasing A from 25% to 95% in 5 min, and holding constant for 2 min, then turning A to 5% in 0.5 min and keeping it at 5% for 6.5 min. ESI was performed in negative ion mode with a capillary voltage of 3.5 kV. The injection volume was set at 10 μL. The standard curves for aromatic acids were constructed over a concentration range of 0.01-1000 μM, with more than seven data points (Fig. S4).

Inorganic ions in SW and SSA samples were filtered through 0.22 μm PTFE filters and analyzed by ion chromatography (Dionex ICS-6000, Thermo Fisher Scientific, USA) coupled with conductivity detection. Cations were isolated on a Dionex IonPac CS12A (4 mm × 250 mm) column, preceded by a guard column, with 20 mM methanesulfonic acid at a flow rate of 1 mL min$^{-1}$. The sample volume was 10 μL for each injection.

**2.4 Data analysis**

SSA production was used to evaluate the particle yield at the air-sea layer. In a previous laboratory study (Christiansen et al., 2019), the production of aromatic acids in SSA was calculated as:

$$\text{SSA Production} = N_{\text{Total}} \times Q_{\text{Sweep}} \tag{1}$$

where $N_{\text{Total}}$ and $Q_{\text{Sweep}}$ are the total number concentration of particles detected from SMPS and the air flow rate through the headspace of the chamber, respectively.

The enrichment factor (EF) of the aromatic acid was then calculated using Eq. (2), where $[X]_{\text{SSA}}$ and $[Na^+]_{\text{SSA}}$ are the concentrations of aromatic acid and Na$^+$ in SSA, $[X]_{\text{SW}}$ and $[Na^+]_{\text{SW}}$ denote the concentrations in SW (Sha et al., 2021a).

$$EF = \frac{[X]_{SSA}/[Na^+]_{SSA}}{[X]_{SW}/[Na^+]_{SW}} \tag{2}$$

An equation to estimate organic acids global flux from SSA emission was developed by Sha et al. (2021b). Both the laboratory-derived EF and the concentration of Na$^+$ data were obtained to quantify the global emission

flux of the aromatic acid.

$$[X]_{SSA} = k_{SSA} \times [Na^+]_{SSA} \qquad (3)$$

$$[X]_{SSA} = EF \times \frac{[X]_{SW}}{[Na^+]_{SW}} \times [Na^+]_{SSA} \qquad (4)$$

$$flux_X = k_{SSA} \times flux_{[Na^+]} \qquad (5)$$

In these equations, $k_{SSA}$ represents the concentration of the aromatic acid transferred to the atmosphere per μg of $Na^+$. In addition, $flux_{[Na^+]}$ means SSA annual production. Note that this quantification assumes a constant concentration of each aromatic acid reported from the literature and no atmospheric aging process of aromatic acids.

## 3. Results and discussion

### 3.1 Surface tension of seawater containing aromatic acids

The surface tension of SW was examined using a surface tensiometer (Sigma 700, Biolin Scientific, Sweden) equipped with a Wilhelmy plate, calibrated at 25 °C with 30 mL of ultrapure water. The variation of seawater surface tensions containing different aromatic acids at room temperature is given in Fig. 1. Surface tension measured in SW ranged from 73.59 to 73.84 mN m$^{-1}$ with a median of 73.75±0.06 mN m$^{-1}$. However, the average surface tension value of ASW was 74.5 mN m$^{-1}$ (Fig. S5), indicating that the presence of surfactants in SW enhances its surface activity. The surface tension of SW containing benzoic acid is 73.67±0.03 mN m$^{-1}$ and certainly lower than that of SW, indicating that benzoic acid slightly reduces the surface tension of SW. Likewise, it is generally observed that surfactants in SW will decrease the surface tension of SW (Cravigan et al., 2020; Liu and Dutcher, 2021; Pierre et al., 2022; Keene et al., 2017; Enders et al., 2023).

The surface tension of both benzenedicarboxylic acids and hydroxybenzoic acids is also lower than that of SW, indicating that the effect of carboxyl and hydroxyl groups on the surface tension of SW is similar. However, differences in the positions of the functional groups of the compounds lead to some differences between the surface tension of SW containing the two types of aromatic acids. SW containing *p*-phthalic acid (73.92±0.14 mN m$^{-1}$) exhibited the highest surface tension among the isomers of benzenedicarboxylic acids. The seawater surface tension varied with the position of the carboxyl group, in the order of *p*-phthalic acid > *m*-phthalic acid > *o*-phthalic acid. It is important to note that benzenedicarboxylic acid used here is amphiphilic, which may be the main cause of results observed above. The current findings on the surface tension of SW containing aromatic acids are also consistent with recent studies that the surface propensity of monocarboxylic acids leads in general to a reduction of surface tension compared to SW, while dicarboxylic acids give higher values (Ozgurel et al., 2022). The theory applies to hydroxybenzoic acids as the surface tension of SW containing hydroxybenzoic acid was lower than that of SW devoid of aromatic acids, with surface tension depressions ranging from 0.01 to 0.17 mN m$^{-1}$.

Unlike the order of seawater surface tension containing benzenedicarboxylic acid, the most obvious inhibition of surface tension was found with the addition of *p*-hydroxybenzoic acid (73.58±0.10 mN m$^{-1}$), while *o*-hydroxybenzoic acid had the weakest effect on seawater surface activity. Similar to the hydroxybenzoic acid, vanillic acid and syringic acid inhibited the surface tension of SW to varying degrees. The results showed that the

hydrophobic group (methoxy group) enhanced the surface activity of SW, in the order of syringic acid > vanillic acid > *p*-hydroxybenzoic acid. As illustrated in Fig. 1D, the seawater surface activity increased with the number of methoxy groups. And the similar trends were also observed in ASW (Fig. S5). Our data demonstrated strong functional groups connections to the seawater surface tension containing aromatic acids.

## 3.2 Transfer of individual aromatic acids to submicron SSA particles

### 3.2.1 SSA size distribution in the different seawater system containing individual aromatic acids

The size distribution of SSA generated from the SSA simulation chamber was measured to study the influence of molecular structure of aromatic acids on SSA production. Fig. S6 shows the number size distribution of SSA generated with the SSA simulation chamber in this study, which could be fitted to a lognormal-mode distribution as observed in previous studies (Quinn et al., 2017; Saliba et al., 2019; Xu et al., 2022). In Fig. 2A, the size distribution of sea salt particles and benzoic acid-added particles shows a peak number concentration near 186.26 nm. The number size distribution showed no significant changes in response to SSA particles after adding benzoic acid, with an SSA production rate of $1.08 \times 10^7$ particles s$^{-1}$.

To explore the response of SSA formation to the number of carboxyl groups in aromatic acids, we measured the particle size distribution in the system after adding different benzenedicarboxylic acids. The number concentration of SSA particles with added benzenedicarboxylic acids was much higher than that with added benzoic acids. As shown in Fig. 2B, *p*-phthalic acid promoted a much higher SSA particles number concentration, which could be observed visually through the SSA simulation chamber window as an increase in bubble bursting. The observed bursting phenomenon is in accordance with the conclusion that higher seawater surface tension could promote the SSA production. The most common explanation is that *p*-phthalic acid has two hydrophilic carboxyl groups at both ends, and the increase of particle number concentration cannot be attributed to the inhibition effect of surfactants on bubbles bursting. Obviously, the presence of benzenedicarboxylic acid increases the particle number concentration, as described in a recent study (Dubitsky et al., 2023). For the particle size distribution, the size distribution of particles with added benzenedicarboxylic acid was much narrower in contrast to the system with added benzoic acids. The morphology of SSA particles seems to change after the addition of benzenedicarboxylic acid, which is very similar to the SSA chamber studies described by Lv et al. (2020) and Lee et al. (2020a), and further indicating that the number and position of carboxylic groups in aromatic acids could significantly affect the formation of SSA.

Further, different functional groups were investigated to determine their effect on particle generation. The distributions show a decrease in the particle number concentrations when adding hydroxybenzoic acids, compared with the system with added benzoic acids, leading to the conclusion that hydroxybenzoic acids, acting as surfactants, could inhibit SSA production (Fig. 2C). Compared with the SW with added benzenedicarboxylic acids, the particles number concentration with added hydroxybenzoic acids decreased significantly. The hydroxybenzoic acids enhanced the seawater surface activity, which would increase the bubble lifetime and then, decrease the SSA production. Importantly, Fig. 2D shows that the particle number concentration decreased proportionally with the increase of the number of methoxy groups, providing further ground that organic matter with hydrophobic functional groups have preferential atomization ability. Previous studies also showed that particles number concentration and GMD increased with the seawater surface tension (Guzmán et al., 2014; Liu et al., 2022), further

indicating that the types of functional groups and the seawater surface tension both affect SSA generation. Moreover, the results showed that the effect trends of aromatic acids on SSA production in ASW were consistent with those observed in SW (Fig. S7), eliminating the influence of organic matter.

Integrated particle size, SSA production and mass concentration based on SMPS are shown in Fig. 3. The mode diameter of SSA particles containing *m*-hydroxybenzoic acid was the largest among all the hydroxybenzoic acid isomers, suggesting that *m*-hydroxybenzoic acid was more likely attached to sea salt particles. In all of the benzenedicarboxylic acids position isomers, *p*-phthalic acid-containing particles gave the largest mode diameter while SSA particles containing *o*-phthalic acid expressed the smallest mode diameter. Rather, the SSA particle diameters of vanillic acid and syringic acid did not change significantly, with a deviation of less than 3.75%. For SSA production, the rate of *p*-phthalic acid-added particles ($1.54 \times 10^7$ particles s$^{-1}$) is higher than that of *m*-phthalic acid-added ($1.33 \times 10^7$ particles s$^{-1}$) and *o*-phthalic acid-added particles ($1.05 \times 10^7$ particles s$^{-1}$). In addition, the SSA production of particles when adding *o*-hydroxybenzoic acid ($1.13 \times 10^7$ particles s$^{-1}$) was obviously higher than that of other isomers (*m*-hydroxybenzoic acid: $0.90 \times 10^7$ particles s$^{-1}$, *p*-hydroxybenzoic acid: $0.98 \times 10^7$ particles s$^{-1}$) shown in Fig. 3, because the surface tensions of SW when adding *m*-hydroxybenzoic acid and *p*-hydroxybenzoic acid were lower than that of SW. These differences in SSA production can likely be attributed to the selective transfer of aromatic acids at the air-sea phases. As described in previous studies, the bubble formation and breakout were controlled by seawater surface tension (Lee et al., 2020b; Liu and Dutcher, 2021). Furthermore, the mass concentration was increased with the SSA production, which is similar to previous findings (Sha et al., 2021a).

### 3.2.2 Morphologies of SSA particles

The particle size and number concentration of particulate matter obtained by SMPS were varied with adding different aromatic acids, which may be closely related to the interaction between aromatic acids and sea salt. To test this hypothesis, SSA particles from SW experimental systems with addition of different aromatic acids were collected and the morphologies of individual particles were characterized (Fig. 4). The TEM images visually show that the cores of the sea salt particles were coated to varying degrees, forming a typical core-shell structure. Furthermore, aluminum foil samples were also characterized by ATR-FTIR for the qualitatively analysis of organic coating. The broad band in the range of 3700–3000 cm$^{-1}$ is O–H stretching in water or acids (Diniz et al., 2018). The other absorption band peak was observed at approximately 1643 cm$^{-1}$, representing the O–H stretching in acids (Jin et al., 2013). The image from Fig. S8A provides the evidence that SSA produced by bubble bursting can transfer acids. The absorbances in the ranges 1900–1670, 1600–1480, 1475–1300 cm$^{-1}$ are the stretching vibration of C=O in aromatic carboxylic acid, C=C stretching in aromatic ring, C–H symmetrical carboxylate stretching (Andreeva and Burkova, 2017; Diniz et al., 2018; Geng et al., 2009; Koutstaal and Ponec, 1993).

As can be seen from Figs. 4A–B, the sea salt particles had a cubic shape, and the benzoic acid coated sea salt particles formed a core-shell structure. With this perspective, the finding of TEM images is in agreement with previously reported morphology for SSA particles (Unger et al., 2020). Figs. 4C–E clearly show the single particle morphology characteristics of benzenedicarboxylic acids. Notably, it is evident that the core morphology of the salt particles underwent a significant change, with the cubic structure being transformed into a sphere structure. The core of SSA particles containing *p*-phthalic acid on the TEM grid became more round compared with SSA

particles containing *m*-phthalic acid and *o*-phthalic acid (Fig. 4A). As previously shown by a different study, these morphologies may suggest that *o*-phthalic acids-containing SSA form round particles via promoting the growth of other surfaces of sea salt particles (Ballabh et al., 2006). The organic coating of sea salt is a well-known process, and has been shown specifically from phytoplankton bloom to produce more organics that likely further lead to spherical structures (Ault et al., 2013). Moreover, the organic coating was getting thicker, in the order of *p*-phthalic acid > *m*-phthalic acid > *o*-phthalic acid. This order in the organic coating clearly explained the result discussed above that the GMD of *p*-phthalic acid-containing particles was markedly larger than those of other isomers of benzenedicarboxylic acids in SSA particles. According to the spectrum in Fig. S8B, for SSA particles containing benzenedicarboxylic acids, the very weak absorption band featuring range of 1000–650 cm$^{-1}$ represents the C–H out of plane bending in the present study and in the literature (Chang et al., 2022; Lin et al., 2014; Świsłocka et al., 2013). The in-situ ATR-FTIR identified SSA particles functional groups provide the key information on the transfer of aromatic acids from SW to the atmosphere.

Fig. 4B show the morphology images of SSA particles containing hydroxybenzoic acids, where the core maintained the crystalline phase of sea salt with its cubic structure and the coatings mainly consisted of hydroxybenzoic acids. Notably, the organic coating of *m*-hydroxybenzoic acid is the thickest among all the isomers. This is consistent with the result that the GMD of *m*-hydroxybenzoic acid is the largest measured by SMPS. The TEM images of Fig. 4C show that individual nascent SSA containing vanillic acid and syringic acid also have the cubic sea salt core, despite their coating thickness is similar. Both characteristic absorption peaks of hydroxybenzoic acid, vanillic acid and syringic acid are observed in Figs. S8C–D, consistent with the results for benzenedicarboxylic acids discussed above. Besides, we expressed herein that sea salt particles surrounded by the aromatic acid coatings would be also inferred from TEM images and ATR-FTIR spectra.

**3.3 Enrichment of individual aromatic acids to submicron SSA particles**

The enrichment factors (EFs) of different aromatic acids in SSA are characterized to visually demonstrate the influence of different functional groups on the accumulation degree of aromatic acids from SW to the atmosphere through SSA. As shown in Fig. 5, all aromatic acids exhibit varying degrees of enrichment in SSA particles. The EF ranged from 5.97 to 24.69, with the largest and smallest being of *m*-hydroxybenzoic acid and *o*-phthalic acid, respectively. The reason for the difference in EFs may be related to the difference in surface tension of SW and the difference in octanol-water partitioning coefficients (log(Kow)) (Olson et al., 2020). In detail, the EF of benzoic acid was approximately 10, indicating that benzoic acid can be significantly transferred to the atmosphere, which is consistent with the findings of ASW (Fig. S9). In combination with the TEM images, not only is the *p*-phthalic acid coating the thickest, but its EF is also the largest among its three isomers. Interestingly, the EFs show the log(Kow) dependence for the three isomers of benzenedicarboxylic acids, in the order of *p*-phthalic acid > *m*-phthalic acid > *o*-phthalic acid. A negative correlation between EF and the log(Kow) of surfactants has been observed (Olson et al., 2020), and the detailed data are available on the website (https://comptox.epa.gov/dashboard/) and summarized in Table S4. The EFs order of benzenedicarboxylic acids is opposite to that of surfactants, probably due to the amphiphilicity of *o*-phthalic acid. However, the EF of *p*-phthalic acid was significantly lower in ASW, which may be due to the depletion of *p*-phthalic acid acts as an ·OH scavenger (Li et al., 2023). In contrast, organic matter in SW reacts preferentially with ·OH and therefore has less

effect on $p$-phthalic acid enrichment (Anastasio and Newberg, 2007).

    Notably, the correlation between the EF containing the isomers of hydroxybenzoic acid and the corresponding organic coating was also strong, among which the EF of $m$-hydroxybenzoic acid (EF = 24.7±8.2) was the highest. Meanwhile, the GMD of SSA particles containing $m$-hydroxybenzoic acid was also the highest. One possible reason could be that the log($K$ow) for $m$-hydroxybenzoic acid (1.50) supports preferential transfer of $m$-
hydroxybenzoic acid to SSA particles. The fundamental role of log($K$ow) in the transfer of organic matter from aqueous phase to particulate phase has been proved (Olson et al., 2020; Mccord et al., 2018; Maagd et al., 1999). However, the enrichment behavior of $o$-hydroxybenzoic acid is weaker than those of others. Therefore, log($K$ow) is not the only factor affecting the EF, which should be combined with the octanol-air partitioning coefficients (log($K$oa)). Just as $o$-hydroxybenzoic acid is easily transferred to the seawater surface microlayer according to
log($K$ow), it is not easily transferred to the atmosphere according to log($K$oa). Compared with $p$-hydroxybenzoic acid, vanillic acid has one more hydrophobic group (methoxy) to its molecular structure. Accordingly, the EF of vanillic acid (13.3±2.5) in SSA is slightly higher than that of $p$-hydroxybenzoic acid (12.1±2.5). The EFs of vanillic acid and syringic acid depend on their surface activities. Syringic acid has strong surface activity, making it to be more easily transported to the atmosphere through SSA than vanillic acid. For most of the target aromatic
acids, the enrichment trends are less affected by the compound concentration (Fig. S9). The above EF data indicates that aromatic acids have significant transport potential at the air-sea interface.

    We hypothesized that the enrichment of aromatic acids is largely related to the bridging effect of $Ca^{2+}$ as well and, as a result, the EFs of $Ca^{2+}$ were calculated. The observed selectivity in cation transfer shown in Fig. 6 has the same trend as the cation binding affinity of aromatic acids. Moreover, another reason for the effective transfer
of $m$-hydroxybenzoic acids to SSA particles is their binding to $Ca^{2+}$, as evidenced by the significant enrichment of $Ca^{2+}$ (Fig. 6). It is known that $Ca^{2+}$ has the potential to transfer organics (Shaloski et al., 2015; Hasenecz et al., 2019). The fact is that aromatic acids can form complexes with cations which are then transferred to SSA by bubble bursting. The strong binding abilities of divalent cations to phenolic −OH and aromatic C=C have been reported (Jayarathne et al., 2016). Thus, the enrichment of major cations acts also as a standard for assessing the
abundance of organic matter. In Fig. 6, the enrichment of $Ca^{2+}$ was the most obvious, followed by those of $Mg^{2+}$ and $K^+$ among all the experiments. In SSA, $Ca^{2+}$ always exhibited high enrichment (EF > 1), while the EFs for $K^+$ and $Mg^{2+}$ were around and just below 1. The above observation of the enrichment capacity of $Ca^{2+}$ is consistent with recent studies (Salter et al., 2016; Unger et al., 2020; Lee et al., 2021; Acker et al., 2021b). Interestingly, the greater the EF for $Ca^{2+}$, the larger the EF for the corresponding aromatic acids. Taking $m$-hydroxybenzoic acid as
an example, we calculated the enrichment factors of $Ca^{2+}$ and $m$-hydroxybenzoic acid in NaCl solution, ASW and SW, and found that both follow the pattern: $EF_{NaCl} < EF_{ASW} < EF_{SW}$ (Fig. S10). Hence, the current results further confirm that $Ca^{2+}$ bridging is important in complexes formation and transport of organics.

**3.4 Estimation of SSA contribution to global aromatic acids emissions**

    The $k_{SSA}$ for aromatic acids ranged from 0.0002 to 0.623 (Table S5). These values were strongly influenced by
the concentration of aromatic acid in SW, with differences of 1–4 orders of magnitude. For example, the seawater concentrations of syringic acid, $p$-hydroxybenzoic acid, benzoic acid are 0.3, 4.58, 34 ng $L^{-1}$, respectively (Zhao et al., 2019). Such large variations highlight the importance of concentration variations in SW of aromatic acid in

assessing the SSA contribution. More than that, benzoic acid and *p*-hydroxybenzoic acid concentrations in the ocean were 205 ng L$^{-1}$ and 8.66 ng L$^{-1}$, respectively (Zhao et al., 2019). Therefore, it is necessary to estimate their atmospheric fluxes. Notably, for syringic acid, $k_{SSA}$ is equivalent to that of *o*-hydroxybenzoic acid, while the concentration of syringic acid in SW is obviously lower. As a result, EFs of aromatic acids also contribute to SSA emission flux. Furthermore, the enrichment process in the field is much more complicated than in controlled laboratory experiments. For example, seawater temperature affects SSA release, while wind speed at the sea surface may influence the amount and size of SSA particles emitted, etc. As such, further research on the environmental enrichment mechanism of aromatic acids in SSA is required to reduce the uncertainty in the estimation of aromatic acid emissions.

On the basis of $k_{SSA}$ and modeled annual SSA, the global emission fluxes of aromatic acids from SW to the atmosphere through SSA can be estimated as eq. 4. The flux of Na$^+$ was obtained by using the annual emission of SSA calculated by Textor et al. (2006) and Gliss et al. (2021). The estimate of aromatic acids fluxes using the above Na$^+$ global emission is included in Table S6 for reference, and the comparison between aromatic acids fluxes is also shown in Fig. 7B. For example, the laboratory-derived annual flux of benzoic acids was 27 tons yr$^{-1}$ for SSA emission of approximately $3.65 \times 10^{12}$ kg yr$^{-1}$, but increased to 71 tons yr$^{-1}$ when SSA emission were about 9.7 kg yr$^{-1}$. Therefore, the SSA emission used in the global annual emission of aromatic acids may be one possible reason of observed differences. Similarly, the atmospheric fluxes measured for aromatic acids are of similar order of magnitude with those for perfluorocarboxylic acids in a previous laboratory study based on field samples (Sha et al., 2021b). It follows that using lab-derived global annual fluxes gives results that are close to the modeling results. As illustrated in Fig. 7, vanillic acid has the highest global emission among all aromatic acids studied, although it has lower concentration in SW than others. This demonstrates that the EF might play a very important role in the global emission fluxes of organic matter, in addition to concentration. Moreover, an in-depth quantification of the organic matter global emission flux transferred to the atmosphere through SSA will help estimating the CCN influence of the organic contribution.

## 4. Conclusion

Based on our experimental data, we highlight that aromatic acids can be transferred from SW to the atmosphere through bubble bursting. The air-sea transfer efficiency of aromatic acids was evaluated by simulating the SSA generation with a plunging jet and collecting the SSA particles. First and foremost, aromatic acids transferred from SW to the SSA particles were confirmed as possible by combining infrared spectra and TEM images. TEM images intuitively show that aromatic acids are coated with sea salt particles to form a core-shell structure, of which further proved that there exists a good correlation between EF of organic acids and organic coating. Our data confirm that the enrichment of aromatic acid was enhanced by the increase of hydrophobic functional groups, while the corresponding number concentration of SSA particles decreased. As a whole, the transfer capacity of aromatic acids may depend on their functional groups and on the bridging effect of cation, as well as their concentration in SW, as these factors influence the global emission flux of aromatic acids via SSA. Overall, our research helps to close an existing knowledge gap in studying the global annual flux of aromatic acids from SSA particles to the atmosphere. Though the current results deepen the understanding of aromatic acids transport, much detailed work is still needed to explore how different aromatic acids affect the marine carbon cycle.

**Data availability**

The data used in this study can be found online at https://doi.org/10.5281/zenodo.10903140 (Song et al., 2024).

**Supplement**

Additional information includes 6 tables and 10 figures. The supplement related to this article is available online at:

**Author contributions**

**Yaru Song:** Conceptualization, Methodology, Investigation, Formal analysis, Writing- Original draft
preparation. **Jianlong Li:** Writing- Reviewing and Editing. **Narcisse Tsona Tchinda:** Writing- Reviewing and Editing. **Kun Li:** Writing- Reviewing and Editing. **Lin Du:** Investigation, Conceptualization, Funding acquisition, Project administration, Resources, Writing- Reviewing and Editing.

**Competing interest**

The authors declare that they have no known competing financial interests or personal relationships that
could have appeared to influence the work reported in this paper.

**Financial support**

This work was supported by National Natural Science Foundation of China (22076099, 22376121).

**Acknowledgments**

We would like to thank Xiaoju Li from State Key laboratory of Microbial Technology of Shandong
University for help and guidance in TEM.

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

**List of Tables**

**Table 1.** Summary of aromatic acids used in the experiments.

| | Aromatic acids | | | |
|---|---|---|---|---|
| | benzoic acid | *o*-phthalic acid | *m*-phthalic acid | *p*-phthalic acid |
| Position of –COOH |  |  |  |  |
| | benzoic acid | *o*-hydroxybenzoic acid | *m*-hydroxybenzoic acid | *p*-hydroxybenzoic acid |
| Position of –OH |  |  |  |  |
| | *p*-hydroxybenzoic acid | vanillic acid | syringic acid | |
| Number of –CH$_3$O |  |  |  | |

**List of Figures**

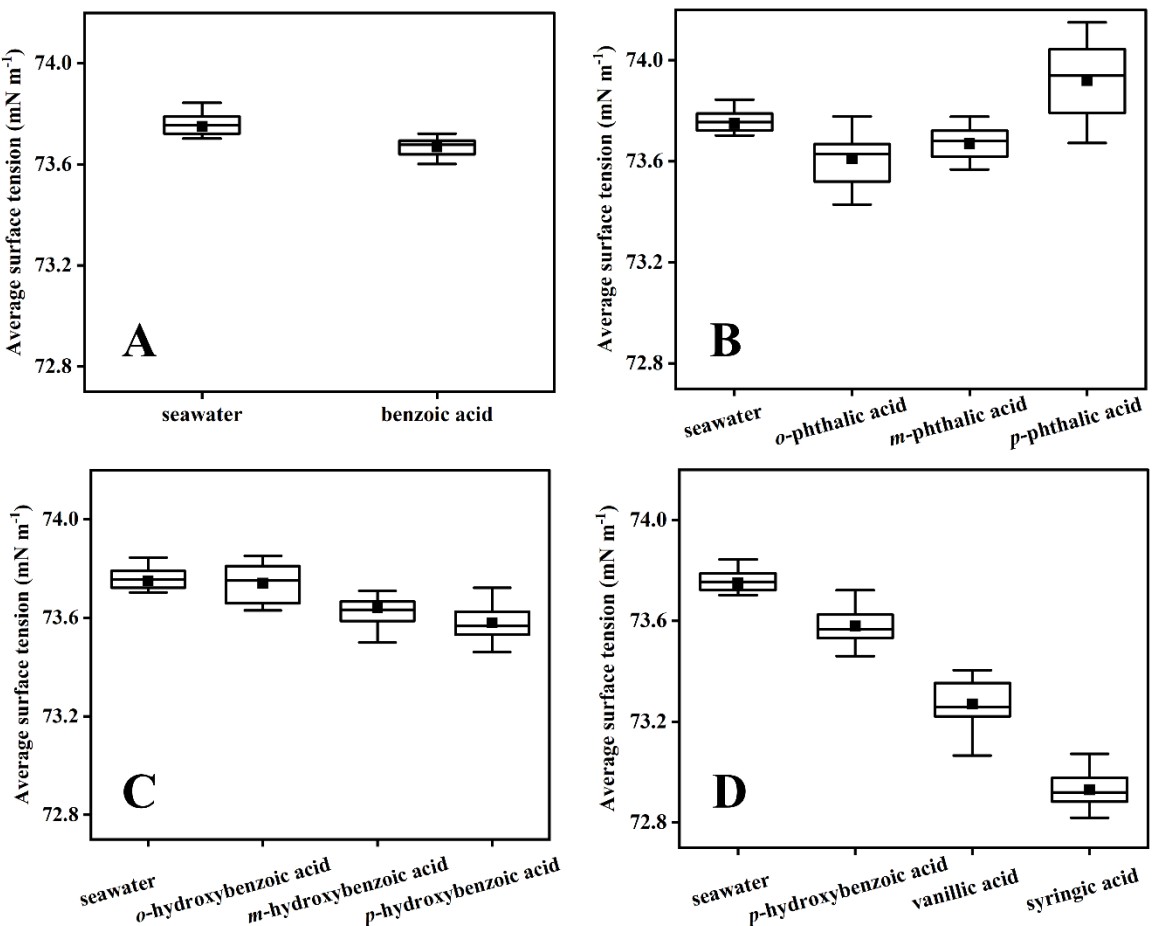

Fig. 1. Measured surface tension values of natural seawater and aromatic acid-containing seawater: benzoic acids (A), benzenedicarboxylic acids (B), hydroxybenzoic acids (C), *p*-hydroxybenzoic acid, vanillic acid, and syringic acid (D). The dark spots represent the mean values of at least 9 data points, the boxes represent the ranges of 25th−50th−75th percentiles, and the whiskers represent the maximum and minimum values.

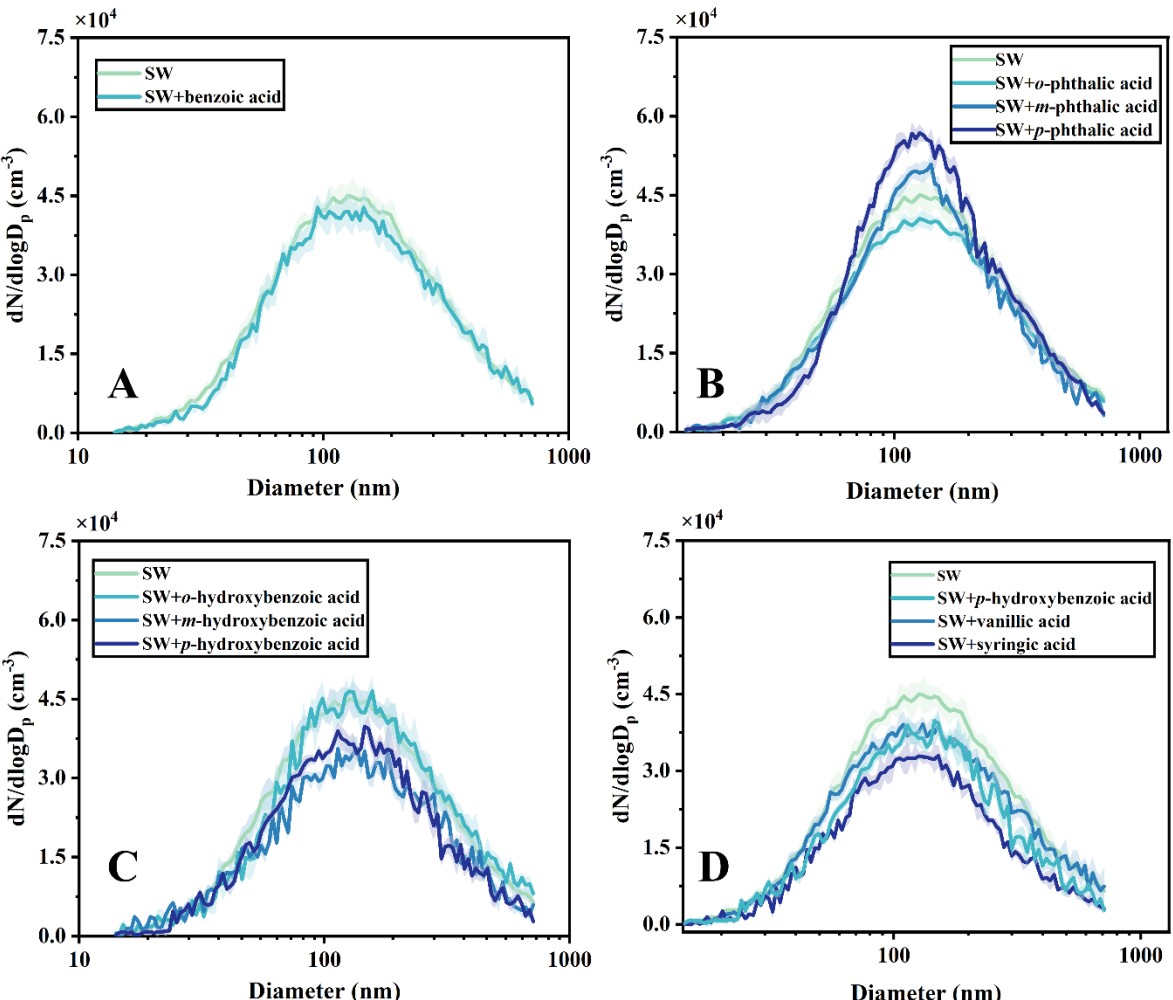

Fig. 2. Number concentration distribution of sea salt particles and SSA particles containing benzoic acids (A), benzenedicarboxylic acids (B), hydroxybenzoic acids (C), *p*-hydroxybenzoic acid, vanillic acid, and syringic acid (D). SW represents natural seawater.

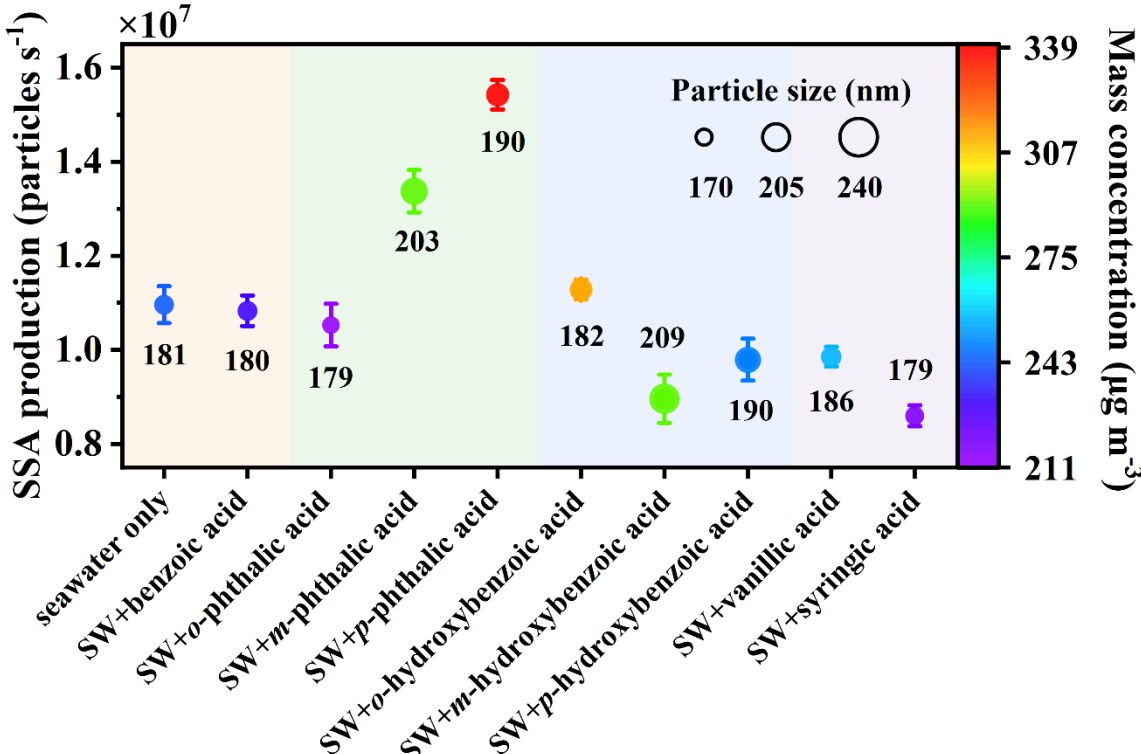

**Fig. 3. SSA production, particle size, mass concentration distribution of aromatic acids. The symbol size**
**represents the geometric mean diameter of SSA particles, with the numbers below or above the points giving**
**the geometric mean diameter (in nm), and the error bars are standard deviation. The symbol color indicates**
**the particle mass concentration, with SW representing natural seawater.**


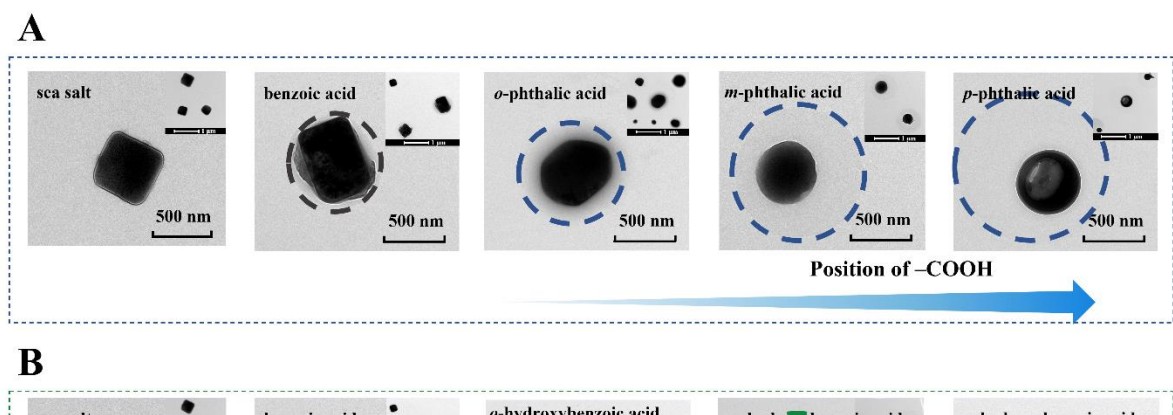

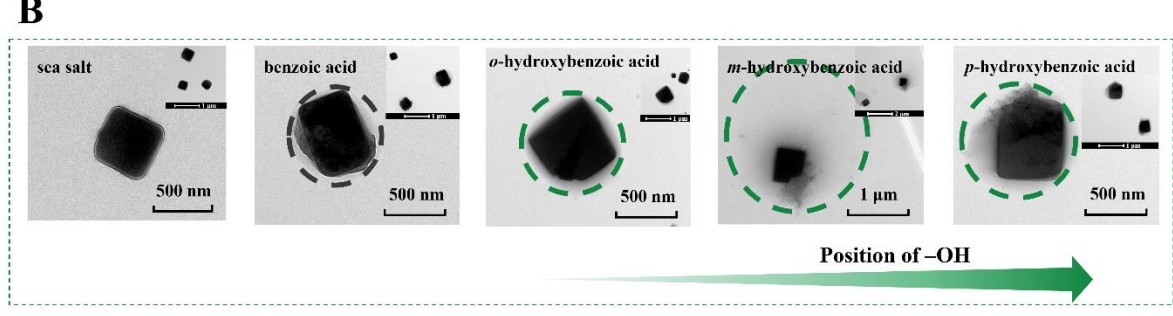

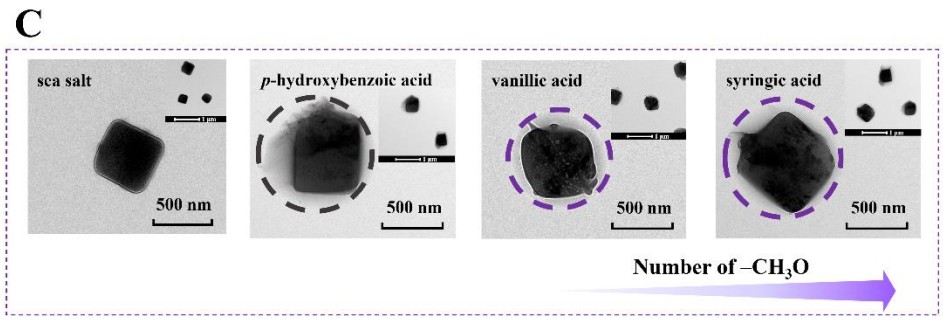

**Fig. 4. Particle morphology observed using TEM of sea salt particles, and benzenedicarboxylic acids- (A), hydroxybenzoic acids- (B), *p*-hydroxybenzoic acid-, vanillic acid-, and syringic acid-coated sea salt particles (C).**

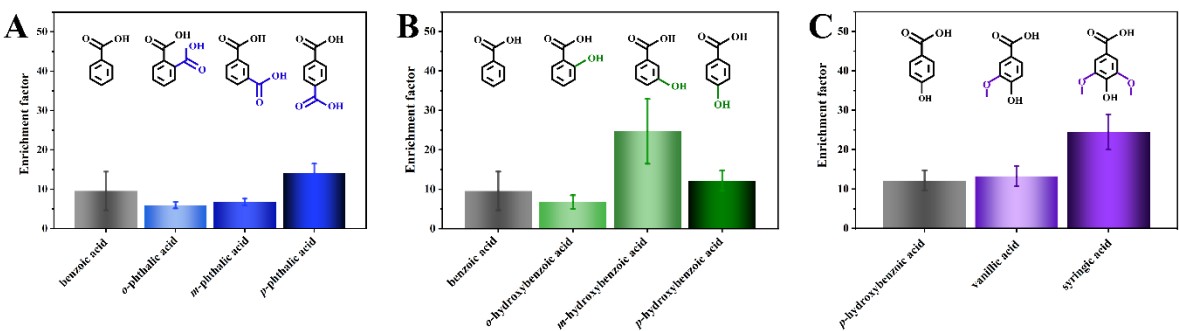

**Fig. 5. Enrichment factors of benzenedicarboxylic acids (A), hydroxybenzoic acids (B), *p*-hydroxybenzoic acid, vanillic acid, and syringic acid (C) from seawater to the atmosphere.**

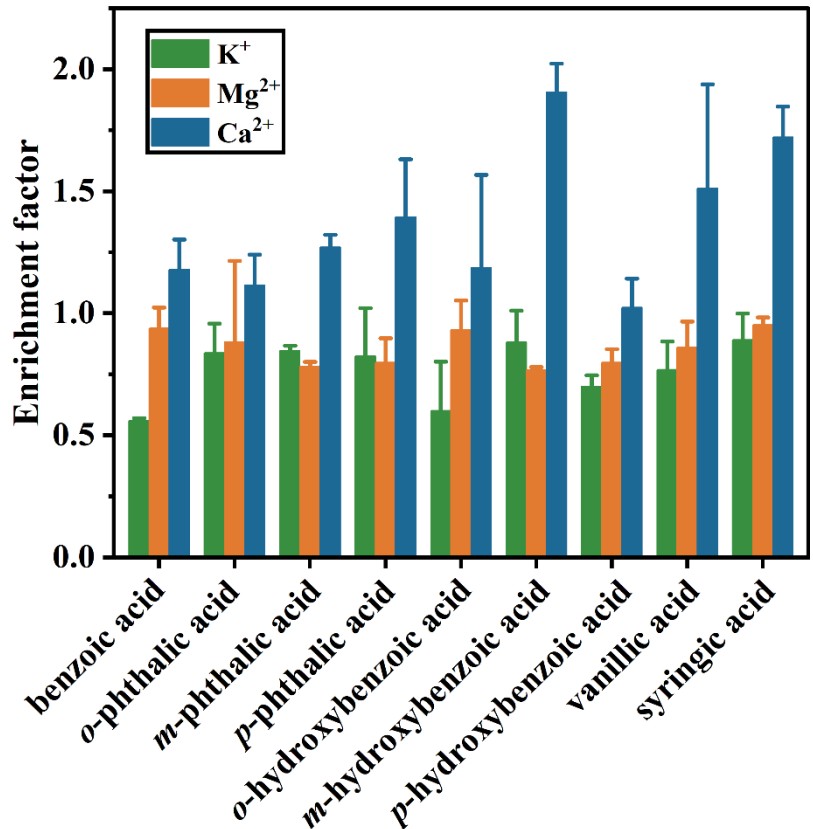

**Fig. 6. Enrichment factors of K⁺, Mg²⁺, and Ca²⁺ in submicron SSA during the experiment.**

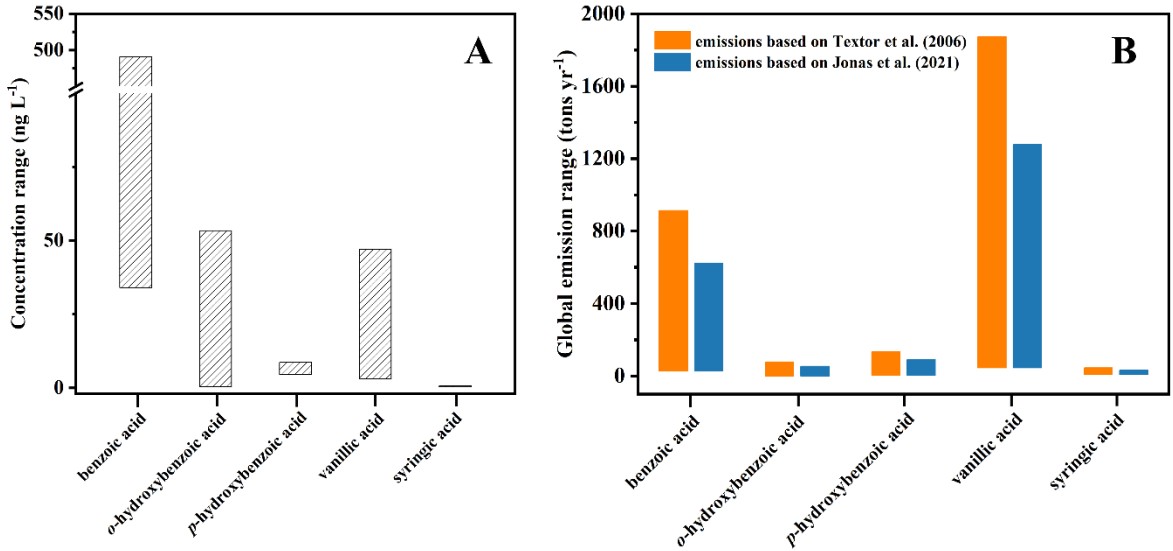

**Fig. 7. Concentration range of aromatic acids in seawater (A) and the estimated range of annual global aromatic acids emission (tons yr⁻¹) via SSA (B). Yellow and blue stacked columns represent emissions based on Textor et al. (2006) and Gliss et al. (2021), respectively.**
