# Peer review of "Role of sea spray aerosol at the air-sea interface in transporting aromatic acids to the atmosphere"

_EGUsphere, 2023_

## Referee Comment (RC2)

**Review of "Role of sea spray aerosol at the air-sea interface in transporting aromatic acids to the atmosphere" by Yaru Song *et al.**

**Summary**

Yaru Song *et al.* conducted a study to investigate the potential transport of organic acids, such as benzoic acids, from the ocean to the atmosphere via sea spray aerosols. The authors generated nascent sea spray aerosols in a laboratory setting using a sea spray simulation chamber equipped with a plunging water jet, which is a well-established method.

The authors aimed to determine the enrichment of the target organic acids in the aerosols compared to a sea spray tracer ion, sodium. Although the experimental principle aligns with previous successful studies (e.g., Johansson et al., 2019; Sha et al., 2021), the current manuscript lacks crucial details about the experimental procedures and quality control. This omission hinders the ability to assess the measurement quality.

The manuscript fails to provide the necessary information to evaluate the reasonableness of the results and the value of the upscaled estimates of global emissions. Consequently, it lacks scientific significance in its present form. Therefore, I recommend rejecting the manuscript for publication in ACP.

To enhance the manuscript's scientific rigor, I suggest the authors make a series of major improvements which I have outlined below. Subsequently, I outline some more minor points for improvement.

**Major points**

- In their introduction, the authors discuss the concept that the aromatic acids at the centre of their study can be of both natural and anthropogenic origin. Although this is an interesting and important discussion, the authors could expand on the relative importance of the different sources. As it stands, it is unclear whether most of the acids considered in this study are of natural or anthropogenic origin or whether they differ for the different acids investigated. A table summarising this information could help clarify the discussion.

- The description of the process of sea spray aerosol formation could be improved. For example, the initial description on line 57 is rather cumbersome and lacks accuracy. The authors can draw inspiration from the wealth of literature available on this topic to improve this section.

- The authors used a common approach to mimic the process of sea spray aerosol formation in the laboratory, namely a sea spray simulation chamber. Although such systems have

become quite common, there is diversity among the systems in use to the extent that it is important to be very precise in describing the particulars of the system in use. Since the characteristics of the different systems used can influence the properties of the aerosols generated, certain information must be conveyed to the reader. For example, the authors used a plunging jet-type system. It is important to include all the relevant details of the system used, such as the diameter of the nozzle through which the seawater flowed into the chamber, the type of pump used to generate the plunging jet, the distance between the exit of the nozzle and the surface of the seawater in the chamber, and the material from which the chamber itself was fabricated.

- It is also important to note that seawater temperature has been shown to influence the process of sea spray aerosol formation. Therefore, it would be helpful to know whether the seawater sample temperature in the chamber was controlled or monitored. Additionally, it would be useful to know how the chamber was cleaned between experiments. Furthermore, in the schematic shown in Figure S1, a 'sweep flow' of particle-free air into the chamber is shown, while the only outlet was to the SMPS." The authors did not specify where the excess air went. Did they not operate with an "overflow" exit to ensure that the chamber was always operated at atmospheric pressure? Additionally, was the Dekati LPI connected to the same outlet as the SMPS system or was it connected to another outlet so that both measurements were conducted simultaneously?

- When evaluating data from sea spray simulation systems, it is important to consider the size of the bubble plume generated by the plunging jet compared to the chamber itself. For instance, did the bubble plume reach the bottom of the chamber? If the surface foam patch is also relatively large in comparison to the surface area of the water, then "wall effects" can occur whereby bubbles reach the side and burst faster than they would if the sides had not affected the bubbles. Therefore, some insight here (perhaps some photographs) would add value to the manuscript.

- Regarding the use of a Dekati LPI and a TEM sampler to obtain offline samples for subsequent analysis, the authors did not mention obtaining blank samples to check for handling impacts. It is unclear whether blank substrates were loaded into the samplers and then analysed in the same way as the samples.

- When it comes to the results of their experiments, the authors first report the impact of different organic acids on surface tension. However, they did not provide enough information to discern how the experiments were carried out. It is assumed that the authors added or "spiked" a known concentration of the different compounds to the seawater samples they collected from the Yellow Sea. However, the concentration of the different compounds used, whether the seawater samples already contained some or all of the compounds, and the variability of the concentration of the analytes of interest in these seawater samples are not mentioned. Additionally, the presence of other surfactants in the form of organic matter derived from natural and anthropogenic sources, and how they could affect the results, is not discussed. All of this information is critical to helping the reader discern the implications of the results. Since natural seawater has variable amounts of these compounds, it is suggested that the authors could have "spiked" the analytes of interest into artificial seawater to try to negate this effect.

- The authors then report the impact of different acids on the size and number of particles that are emitted as SSA. It is unclear what the concentrations of the different analytes in these experiments were. It is also unclear whether the same seawater was used for all experiments or whether different seawater samples were spiked. The concentration of the analytes in the seawater sample and the level of organic matter in the seawater are not mentioned.

- Following this, the authors present enrichment factors of the different organic acids on the nascent sea spray aerosol they generate. However, many details are again missing, which makes it impossible to evaluate the results. For example, it is not clear to me whether the presented results are the average over all stages of the impactor or if they only represent a single stage. Furthermore, the authors did not mention how they quantified the sodium ion and the concentration of the organic acids in the same sample. Did the authors extract the substrates in ultrapure water and then subsample these extracts for the different analyses, or did they use a different approach? To generate their enrichment factor estimates, the authors will have used a concentration value for each of the acids in the seawater used to generate the aerosols. Was this value quantified, or did the authors simply use the "spike" concentration they aimed for? This is important given that surface active species, such as the organic acids used, have a tendency to stick to the walls of chambers and the actual concentration of the organic acids in the seawater may well have been well below the "spike" value the authors aimed for. Along the same lines, assuming that the authors did quantify the actual concentration of the organic acids in the seawater, it would be good to know at what point of the experiment they obtained these samples. For example, the authors state that the LPI was run for 5 hours - were water samples taken at the start or end of this period? In similar experiments using perfluoroalkyl acids, Johannsson et al. (2019) observed that the concentration of the substances in the seawater from which they were generating aerosol during their experiment decreased as these substances were "lost" to the aerosol.

- On Line 84, the authors introduce the idea of upscaling their measurements using literature values for the seawater concentration of these compounds. However, the reader requires more information at this juncture. Questions arise, such as: What are the typical seawater concentrations of these compounds? How much do they vary globally? How extensive is the dataset of such measurements? Incorporating this information into the previously introduced table (major point 1) would be beneficial. While the authors eventually provide some literature values in section 3.4 of the manuscript, it remains unclear which specific values were utilised. Was it an average of the literature values or a different approach? Considering the likely high spatial (and potentially temporal) variability in these concentrations, it is crucial to understand how the authors addressed this variability. Presently, it is nearly impossible to evaluate the magnitude of the emissions proposed by the authors due to the uncertainty in the conducted measurements.

**Minor points**

- Line 30: I don't think "captured" is the right word here. Perhaps "taken up" would be better.

- Line 31: Would read better as: "...leading to their enrichment within these organisms or transportation to remote areas (Fu et al.,..."

- Line 31: I would also break this sentence to aid readability, e.g.: "This process poses health risks to the endocrine system of aquatic organisms and the overall marine ecosystem (Saha et al., 2006)."

- Line 35: Would read better as "Previous studies suggest that these organic acids can alter the composition of SSA, subsequently influencing atmospheric processes such as cloud condensation nuclei (CCN) or ice nuclei (IN) activities.

- Line 36: Needs rephrasing. It is not the "organic acids" that are the source of the aerosol. Rather, it is SSA that may contain organic acids that play a role in Earth's climate.

- Line 50: Would read better as "...the identification of phthalic acid in organisms raises the possibility of its origin from the ingestion of marine plastics."

- Line 53: Would read better as "This phenomenon has prompted discussions about the concept of "missing aromatic acids."

- Line 54: Would read better as "Nevertheless, the currently available data do not provide a conclusive explanation for the existence of these "missing aromatic acids."

- Line 55: This paragraph could be better linked to the previous discussion. For example, "One possible reason for the disappearance of these aromatic acids is their release into the atmosphere. Existing datasets obtained from remote marine areas offer evidence of the presence of these compounds in the atmosphere (Fu et al., 2010)."

- Line 66: Would read better: "While field studies have demonstrated the presence of aromatic acids in both the ocean and atmosphere (Boreddy et al., 2017), the specific mechanisms influencing their transport at the air-sea interface require further investigation."

- Line 69: Here the authors introduce the air-sea transport of perfluoroalkyl acids (PFAAs) via sea spray aerosol. Although this is an interesting topic, the link to the work carried out in this study is unclear. I urge the authors to better describe this link or remove this reference.

- Line 74: Would read better as "In a previous study, we observed that the transfer of short-chain organic acids between the air and sea through SSA may be influenced by seawater surface tension. This factor could, in turn, impact the enrichment behaviour of organic acids (Song et al., 2022)."

- Line 77: The authors state "...other factors were discussed in our recent review..." What are the other factors? If this discussion is important, then the authors should do it justice and include all relevant information. Otherwise, I see no need to mention this.

- Line 78: There should be a new paragraph here: "In this study..."

**References**

Johansson, Jana H., et al. "Global transport of perfluoroalkyl acids via sea spray aerosol." Environmental Science: Processes & Impacts 21.4 (2019): 635-649.

Sha, B., Johansson, J. H., Benskin, J. P., Cousins, I. T., and Salter, M. E.: Influence of water concentrations of perfluoroalkyl acids (PFAAs) on their size-resolved enrichment in nascent sea spray aerosols, Environ. Sci. Technol., 55, 9489–9497, 10.1021/acs.est.0c03804, 2021

---

## Author Comment (AC1)

We thank the Reviewer for the insightful comments. We have revised our manuscript according to the suggestions of the Reviewer's comments and our responses to the comments are as follows: Reviewer's comments are in black, authors' responses are in blue, and changes to the manuscript are in red color text. Figures prepared for reply are named as Figure R-.

**Reviewer #1:**

**General comments**

I find this manuscript well-written / well-referenced, scientifically interesting for the SSA community, and the experimental quality is good. I find it great that the authors try connect SSA experiments with functional group level chemistry.

I have some minor concerns, that should be easily to address. I would recommend publication with only minor revisions.

**Author reply:**

We thank the Reviewer for the positive assessment of our manuscript and the constructive comments. We have revised our manuscript according to the suggestions of the Reviewer's comments and our responses to the comments are as follows.

**Specific comments**

1. Real seawater composition. In line 94 you write that you sample and transport seawater to the SSA laboratory. Is it possible to get more details? Conditions at sampling site (is it a productive area?), temperature, duration of storage, volume, was it filtered? Was it sampled on the same or different days? What time of year? You can add this information to the SI or just expand Table S1.

   **Author reply:**

   Based on the global net primary productivity estimated by Dai et al. (2023), the sampling site is a high-productivity area. We have indicated the storage conditions

and time in the revised SI and expanded Table S3 to make the experimental process clearer.

**S1.** Quality assurance/quality control

Seawater was collected from the coastal area of Shazikou on March 27, 2023, with a volume of 500 L (Fig. S1). Considering the storage inconvenience caused by huge consumption of seawater, all our seawater was pre-filtered through a polyethersulfone filter (47 mm diameter, 0.2 μm pore size, Supor®-200, Pall Life Sciences, USA) and stored in the dark at 18 ℃ for less than one month. Quinn et al. (2015) have shown that the fraction that passes through the filter is regarded as dissolved organic carbon and includes colloidal and truly dissolved materials.

[Figure]

**Fig. S1.** Sampling site at Shazikou along the Yellow Sea coast, Qingdao, China.

**Table S3.** Summary of experimental conditions.

| Exp. No. | Experiment type | Concentration (mM) | pH | Salinity (psu) | Sampling time (h) | RH (%) | Temperature difference (°C) [a] |
|---|---|---|---|---|---|---|---|
| 1 | SW | 0 | 7.92 | 34.2 | 5 | 35 | 2.0 |
| 2 | SW+benzoic acid | 1 | 7.72 | 34.3 | 5 | 34 | 1.5 |
| 3 | SW+$o$-hydroxybenzoic acid | 1 | 7.60 | 34.5 | 5 | 36 | 1.0 |
| 4 | SW+$m$-hydroxybenzoic acid | 1 | 7.68 | 34.1 | 5 | 40 | 2.0 |
| 5 | SW+$p$-hydroxybenzoic acid | 1 | 7.84 | 34.3 | 5 | 38 | 1.5 |
| 6 | SW+$o$-phthalic acid | 1 | 7.58 | 34.2 | 5 | 36 | 2.0 |
| 7 | SW+$m$-phthalic acid | 1 | 7.80 | 34.5 | 5 | 37 | 2.5 |
| 8 | SW+$p$-phthalic acid | 1 | 7.85 | 34.4 | 5 | 42 | 2.0 |
| 9 | SW+vanillic acid | 1 | 7.81 | 34.2 | 5 | 43 | 3.0 |
| 10 | SW+syringic acid | 1 | 7.84 | 34.3 | 5 | 39 | 2.0 |
| 11 | ASW | 0 | 7.96 | 35.1 | 5 | 33 | 1.5 |
| 12 | ASW+benzoic acid | 1 | 7.68 | 34.6 | 5 | 35 | 1.0 |
| 13 | ASW+$o$-hydroxybenzoic acid | 1 | 7.76 | 34.9 | 5 | 34 | 0.5 |
| 14 | ASW+$m$-hydroxybenzoic acid | 1 | 7.99 | 35.3 | 5 | 36 | 1.5 |
| 15 | ASW+$p$-hydroxybenzoic acid | 1 | 7.85 | 34.7 | 5 | 38 | 2.0 |
| 16 | ASW+$o$-phthalic acid | 1 | 7.93 | 34.5 | 5 | 35 | 1.0 |

| 17 | ASW+*m*-phthalic acid | 1 | 7.88 | 34.9 | 5 | 36 | 1.0 |
| 18 | ASW+*p*-phthalic acid | 1 | 7.97 | 34.6 | 5 | 34 | 1.5 |
| 19 | ASW+vanillic acid | 1 | 7.89 | 35.2 | 5 | 35 | 1.0 |
| 20 | ASW+syringic acid | 1 | 7.99 | 34.8 | 5 | 39 | 1.0 |
| 21 | ASW+benzoic acid+*o*-hydroxybenzoic acid+*o*-phthalic acid+vanillic acid+syringic acid | $10^{-3}$ | 7.95 | 35.1 | 20 | 41 | 3.5 |
| 22 | ASW+benzoic acid+*m*-hydroxybenzoic acid+*m*-phthalic acid+vanillic acid+syringic acid | $10^{-3}$ | 7.98 | 34.6 | 20 | 38 | 1.5 |
| 23 | ASW+benzoic acid+*p*-hydroxybenzoic acid+*p*-phthalic acid+vanillic acid+syringic acid | $10^{-3}$ | 7.88 | 34.9 | 20 | 40 | 2.0 |
| 24 | NaCl | 0 | 7.68 | 35.3 | 5 | 38 | 1.0 |
| 25 | NaCl+*m*-hydroxybenzoic acid | 1 | 7.54 | 34.7 | 5 | 36 | 1.5 |

[a] The temperature difference in the SSA simulation chamber before and after the experiment.

       o   Also I have had challenges when I sampled fresh real seawater, that the SSA properties (size and number) changed as a function of time in the SSA chamber (due to microbial activity, degassing) – I therefore sometimes prepared artificial seawater from just inorganic sea salts. Could the authors elaborate on how reproducible the experiments are? And would the authors expect the results being similar using artificial inorganic mixture?

**Author reply:**

1) All our seawater was pre-filtered through a polyethersulfone filter (47 mm diameter, 0.2 μm pore size, Supor®-200, Pall Life Sciences, USA) and stored in the dark at 18 °C for less than one month to minimize microbiological effects.

2) We measured the total particle number concentration and concentration of $Na^+$ of seawater before each experiment, and all the experiments showed good repeatability (Fig. R1). Furthermore, for comparing the properties of SSA particles containing aromatic acids, we normalized the particles size distribution of seawater before adding aromatic acids. Therefore, perhaps it is likely that the same trend would exist in artificial seawater.

[Figure]

Fig. R1. Mean value of total particle number concentration of SSA ($N_{Total}$) and concentration of $Na^+$ ($C(Na^+)$) for each experiment.

3) To demonstrate this conclusion more rigorously, we added aromatic acid to the artificial seawater and observed its effect on the particle size distribution of SSA particles. The following text has been added in the revised manuscript.

Lines 241-242:

Moreover, the results showed that the effect trends of aromatic acids on SSA production in ASW were consistent with those observed in seawater (Fig. S7), eliminating the influence of organic matter.

[Figure]

**Fig. S7.** Number concentration distribution of sea salt particles and SSA particles containing benzoic acids (A), benzene dicarboxylic acids (B), hydroxybenzoic acids (C), vanillic acid and syringic acid (D).

From the results, we can see that the effect trend of aromatic acids on SSA production is generally similar, not only in seawater but also in artificial seawater.

Furthermore, we also measured the artificial seawater surface tension with aromatic acids and the EF of aromatic acids, and the results are as follows.

[Figure]

**Fig. S6.** Measured surface tension values of artificial seawater (ASW) and aromatic acid-containing ASW.

From the results, we can see that the effect trend of aromatic acids on seawater surface tension is generally similar, not only in seawater but also in artificial seawater.

[Figure]

**Fig. S8.** Enrichment factors of aromatic acids at different concentrations from artificial seawater to the atmosphere.

Comparing the EF of aromatic acids in SW and ASW, it was observed that the EF trends of benzene dicarboxylic acids in seawater follows the pattern: *o*-phthalic acid < *m*-

phthalic acid < *p*-phthalic acid. However, in ASW, the EF of *p*-phthalic acid was lower than that of *m*-phthalic acid. Based on the findings of Li et al. (2023), we hypothesize that in ASW, *p*-phthalic acid acts as an ·OH scavenger to produce TAOH. Hence, the EF of *p*-phthalic acid is lower than that of *m*-phthalic acid. Meanwhile, organic compounds in SW preferentially react with ·OH (Anastasio and Newberg, 2007), thus the EF of *p*-phthalic acid is the highest among the benzene dicarboxylic acids. Furthermore, differences in aromatic acid concentration did not change the enrichment pattern of organic acids.

2. Experimental Setup. Would be helpful for the reader if Figure S1 was updated to include schematics of the entire setup, e.g. add where the DLPI+ was connected (before or after dryer?), single particle sampler (TEM). The air flow rate into SSA chamber (Line 109). Why does the range span from 3 all the way to 50 L min$^{-1}$? Is it because you have different setups at different times during a single experiment? Could you elaborate more on this.

**Author reply:**

The Dekati DLPI+ and single particle sampler were connected after dryer and sampling was carried out separately. We updated figure S3 to include schematics of the entire setup in the supplement to make it clearer. For the air flow rate, we would like to express that the zero air can be adjusted in the range of 3–50 L min$^{-1}$, e.g. zero air flow rate into the SSA simulation chamber was set at 3 L min$^{-1}$, while the zero air flow rate of the Nafion dryer tube was set at three times the outlet air flow rate when we measured the size distribution of SSA particles. When the total particle concentration has stabilized, we connected the single particle sampler to sample the SSA particles under the same air flow rate. The inlet flow rate for Dekai DLPI+ sampler sampling was set to 10 L min$^{-1}$ to

supply its pumping flow. We also updated Figure S3 to include air flow rate in the supplement to make it clearer.

[Figure]

Fig. S3. Schematic picture of the plunging jet-sea spray aerosol generator: SMPS sampling (A), single particle sampling (B), and DeKati DLPI+ sampling (C). The red arrows represent the flow direction of seawater, and the purple arrows represent the flow of gases and aerosol particles.

3. TEM details. I am missing details about TEM instrument and conditions. When you do TEM, wouldn't the organic coating of SSA just vaporize in the vacuum? Also, wouldn't the SSA and aromatic acids be more internally mixed, when use a plunging jet / real bubble bursting?

**Author reply:**

We have added details about TEM instrument and conditions in the manuscript. The copper grid film was placed in a liquid nitrogen-frozen vacuum environment through a TEM holder, thereby inhibiting the evaporation of the organic coating of SSA. According to the classification of aerosol particles by Li et al. (2016), a particle consisting of two or more aerosol components can be defined as an internally mixed particle. Otherwise, it will be regarded as an externally mixed particle. For the mixing state of SSA particles, the OM coating and core-shell mixing structure were considered as internally mixed in previous studies (Li et al., 2016; Li et al., 2021).

Lines 138-141:

TEM was performed using a TEM cryo-mount (Gatan 626) to load the samples, where the TEM grid was immersed in liquid nitrogen and then mounted on the holder by means of a cryo-transfer workstation. TEM with a high-angle annular-dark-field detector was used and then TEM images were obtained at an accelerating voltage of 200 kV.

**Technical comments**

Line 58: add mass, so it reads main mass component

**Author reply:**

We simplified the original sentence as follows.

Lines 59-61:

Sea spray aerosols (SSAs), generated by breaking waves and bubble bursting, are one of the major sources of atmospheric particles (Andreae and Rosenfeld, 2008; Angle et al., 2021; Hasenecz et al., 2020; Malfatti et al., 2019).

Line 61: "... disturbing ecological systems... ", sound funny. Maybe change to "further impacting" or "further interaction with"

**Author reply:**

We have rephrased the original text as:

Lines 61-63:

SSAs can act as carrier agents for the vertical transport of much more than just sea salt and often include organic surfactants in the ocean, as already shown by field and laboratory studies (Cochran et al., 2016; Franklin et al., 2022; Rastelli et al., 2017).

Line 163: The **k** look wierd, should be $k_{SSA}$ as in Sha et al, right? Also chemical symbols should be upright, not italic.

**Author reply:**

Yes. We have replaced "**k**" by "$k_{SSA}$" in the revised manuscript.

Line 202: Unit, Part s$^{-1}$, would prefer just s$^{-1}$ or particles s$^{-1}$.

**Author reply:**

We have modified corresponding sentences and figures in the revised manuscript. All the SSA production units previously noted as "part s$^{-1}$" of seawater are now expressed as "particles s$^{-1}$".

Line 207: what does increase in bubble bursting refer to? Is it foam stability or lifetime? Or smaller and more bubbles?

**Author reply:**

The increase in bubble bursting here refers to smaller and more bubbles.

Line 260: change ball to sphere

**Author reply:**

We have modified corresponding sentences in the manuscript.

Lines 275-276:

Notably, it can be seen that the core morphology of salt particles had changed significantly, where the cubic structure has changed into a sphere structure.

Line 323: "lousy" is informal slang, change to "very poor" or just "poor" or "around and just below 1".

**Author reply:**

We thank the Reviewer for pointing this out, a new sentence has been shown in the revised manuscript.

Lines 342-343:

In SSA, Ca$^{2+}$ always exhibited high enrichment (EF > 1), while the EFs for K$^{+}$ and Mg$^{2+}$ were around and just below 1.

Line 351: … plays a very important role… Tone down, add "might play"

**Author reply:**

We have modified corresponding sentences in the manuscript.

Lines 373-374:

This demonstrates that the EF might play a very important role in the global emission fluxes of organic matter, in addition to concentration.

Figure 1: Maybe same range on y-axis? Easier to compare across subplots. I am colorblind, do you need the colors? If you perfer using colors, then should be the same as in Figure 2.

**Author reply:**

We have modified corresponding figures in the manuscript. We used the same range on y-axis and removed unnecessary colors.

[Figure]

**Fig. 1.** Measured surface tension values of natural seawater and aromatic acid-containing seawater: benzoic acids (A), benzene dicarboxylic acids (B), hydroxybenzoic acids (C), *p*-hydroxybenzoic acid, vanillic acid, and syringic acid (D). The dark spots represent Mean values of at least three measurements and the boxes represent the ranges of 25th−50th−75th percentiles.

Figure 2. I would change the colors. Yellow is difficult to see. Look at this website for inspiration: https://colorbrewer2.org/#type=diverging&scheme=BrBG&n=4

**Author reply:**

We thank the Reviewer for providing us with this useful website. We have changed the colors of Figure 2 in the revised manuscript.

[Figure]

**Fig. 2.** Number concentration distribution of sea salt particles and SSA particles containing benzoic acids (A), benzene dicarboxylic acids (B), hydroxybenzoic acids (C), *p*-hydroxybenzoic acid, vanillic acid, and syringic acid (D).

Figure 3. Seawater = Seawater only. And maybe add some errorbar estimation with respect to SSA production? Here color is okay, but that white shadow looks funny.

**Author reply:**

We have added error bar estimation with respect to SSA production of Figure 3 in the revised manuscript.

[Figure]

**Fig. 3.** SSA production, particle size, mass concentration distribution of aromatic acids. The symbol size represents the geometric mean diameter of SSA particles and is marked with numbers, and the symbol color indicates the particle mass concentration.

Figure 6: Maybe make different symbols and use one color? Feel free to ignore this comment.

**Author reply:**

We have tried to make different symbols in Figure 6 in the revised manuscript. Using the same color makes the symbols overlap. Hence, we used different colors.

[Figure]

**Fig. 6.** Enrichment factors of $K^+$, $Mg^{2+}$, and $Ca^{2+}$ in submicron SSA during the experiment.

Figure 7. The space between the bars are not the same. Also I would change colors (Feel free to ignore this comment)

**Author reply:**

We have made the space between the bars same and changed colors of Figure 7 in the revised manuscript.

[Figure]

**Fig. 7.** Estimated annual global aromatic acids emission (tons yr⁻¹) via SSA. Yellow and blue stacked columns represent emissions based on Textor et al. (2006) and Jonas et al. (2021), respectively.

**References**

Anastasio, C. and Newberg, J. T.: Sources and sinks of hydroxyl radical in sea-salt particles, J. Geophys. Res. Atmos., 112, D10306, 10.1029/2006jd008061, 2007.

Andreae, M. O. and Rosenfeld, D.: Aerosol-cloud-precipitation interactions. Part 1. The nature and sources of cloud-active aerosols, Earth Sci. Rev., 89, 13–41, 10.1016/j.earscirev.2008.03.001, 2008.

Angle, K. J., Crocker, D. R., Simpson, R. M. C., Mayer, K. J., Garofalo, L. A., Moore, A. N., Mora Garcia, S. L., Or, V. W., Srinivasan, S., Farhan, M., Sauer, J. S., Lee, C., Pothier, M. A., Farmer, D. K., Martz, T. R., Bertram, T. H., Cappa, C. D., Prather, K. A., and Grassian, V. H.: Acidity across the interface from the ocean surface to sea spray aerosol, Proc. Natl. Acad. Sci. U.S.A., 118, e2018397118, 10.1073/pnas.2018397118, 2021.

Cochran, R. E., Laskina, O., Jayarathne, T., Laskin, A., Laskin, J., Lin, P., Sultana, C., Lee, C., Moore, K. A., Cappa, C. D., Bertram, T. H., Prather, K. A., Grassian, V. H., and Stone, E. A.: Analysis of organic anionic surfactants in fine and coarse fractions of freshly emitted sea spray aerosol, Environ. Sci. Technol., 50, 2477-2486, 10.1021/acs.est.5b04053, 2016.

Franklin, E. B., Amiri, S., Crocker, D., Morris, C., Mayer, K., Sauer, J. S., Weber, R. J., Lee, C., Malfatti, F., Cappa, C. D., Bertram, T. H., Prather, K. A., and Goldstein, A. H.: Anthropogenic and biogenic contributions to the organic composition of coastal submicron sea spray aerosol, Environ. Sci. Technol., 56, 16633–16642, 10.1021/acs.est.2c04848, 2022.

Hasenecz, E. S., Jayarathne, T., Pendergraft, M. A., Santander, M. V., Mayer, K. J., Sauer, J., Lee, C., Gibson, W. S., Kruse, S. M., Malfatti, F., Prather, K. A., and Stone, E. A.: Marine bacteria affect saccharide enrichment in sea spray aerosol during a phytoplankton bloom, ACS Earth Space Chem., 4, 1638–1649, 10.1021/acsearthspacechem.0c00167, 2020.

Jonas, G., Augustin, M., Michael, S., Andrews, E., Balkanski, Y., Bauer, S. E., Benedictow, A. M. K., Bian, H., Checa-Garcia, R., Chin, M., Ginoux, P., Griesfeller, J. J., Heckel, A., Kipling, Z., Kirkevåg, A., Kokkola, H., Laj, P., Le Sager, P., Lund, M. T., Lund Myhre, C., Matsui, H., Myhre, G., Neubauer, D., van Noije, T., North, P., Olivié, D. J. L., Rémy, S., Sogacheva, L., Takemura, T., Tsigaridis, K., and Tsyro, S. G.: AeroCom phase III multi-model evaluation of the aerosol life cycle and optical properties using ground- and space-based remote sensing as well as surface in situ observations, Atmos. Chem. Phys., 21, 87–128, 10.5194/acp-21-87-2021, 2021.

Li, K., Guo, Y., Nizkorodov, S. A., Rudich, Y., Angelaki, M., Wang, X., An, T., Perrier, S., and George, C.: Spontaneous dark formation of OH radicals at the interface of aqueous atmospheric droplets, Proc. Natl. Acad. Sci. U.S.A., 120, e2220228120, 10.1073/pnas.2220228120, 2023.

Li, W., Liu, L., Zhang, J., Xu, L., Wang, Y., Sun, Y., and Shi, Z.: Microscopic evidence for phase separation of organic species and inorganic salts in fine ambient aerosol particles, Environ. Sci. Technol., 55, 2234-2242, 10.1021/acs.est.0c02333, 2021.

Li, W., Sun, J., Xu, L., Shi, Z., Riemer, N., Sun, Y., Fu, P., Zhang, J., Lin, Y., Wang, X., Shao, L., Chen, J., Zhang, X., Wang, Z., and Wang, W.: A conceptual framework for mixing structures in individual aerosol particles, J. Geophys. Res. Atmos., 121, 13784–13798, 10.1002/2016jd025252, 2016.

Malfatti, F., Lee, C., Tinta, T., Pendergraft, M. A., Celussi, M., Zhou, Y. Y., Sultana, C. M., Rotter, A., Axson, J. L., Collins, D. B., Santander, M. V., Morales, A. L. A., Aluwihare, L. I., Riemer, N., Grassian, V. H., Azam, F., and Prather, K. A.: Detection of active microbial enzymes in nascent sea spray aerosol: Implications for atmospheric chemistry and climate, Environ. Sci. Technol. Lett., 6, 171–177, 10.1021/acs.estlett.8b00699, 2019.

Quinn, P. K., Collins, D. B., Grassian, V. H., Prather, K. A., and Bates, T. S.: Chemistry and related

properties of freshly emitted sea spray aerosol, Chem. Rev., 115, 4383-4399, 10.1021/cr500713g, 2015.

Rastelli, E., Corinaldesi, C., Dell'Anno, A., Lo Martire, M., Greco, S., Cristina Facchini, M., Rinaldi, M., O'Dowd, C., Ceburnis, D., and Danovaro, R.: Transfer of labile organic matter and microbes from the ocean surface to the marine aerosol: an experimental approach, Sci. Rep., 7, 11475, 10.1038/s41598-017-10563-z, 2017.

Textor, C., Schulz, M., Guibert, S., Kinne, S., Balkanski, Y., and Bauer, S.: Analysis and quantification of the diversities of aerosol life cycles within AeroCom, Atmos. Chem. Phys., 6, 1777–1813, 10.5194/acp-6-1777-2006, 2006.

---

## Author Comment (AC2)

We thank the Reviewer for the insightful comments. We have revised our manuscript according to the suggestions of the Reviewer's comments and our responses to the comments are as follows: Reviewer's comments are in black, authors' responses are in blue, and changes to the manuscript are in red color text. Figures prepared for reply are named as Figure R-.

**Reviewer #2:**

Review of "Role of sea spray aerosol at the air-sea interface in transporting aromatic acids to the atmosphere" by Yaru Song et al.

**Summary**

Yaru Song et al. conducted a study to investigate the potential transport of organic acids, such as benzoic acids, from the ocean to the atmosphere via sea spray aerosols. The authors generated nascent sea spray aerosols in a laboratory setting using a sea spray simulation chamber equipped with a plunging water jet, which is a well-established method.

The authors aimed to determine the enrichment of the target organic acids in the aerosols compared to a sea spray tracer ion, sodium. Although the experimental principle aligns with previous successful studies (e.g., Johansson et al., 2019; Sha et al., 2021), the current manuscript lacks crucial details about the experimental procedures and quality control. This omission hinders the ability to assess the measurement quality. The manuscript fails to provide the necessary information to evaluate the reasonableness of the results and the value of the upscaled estimates of global emissions. Consequently, it lacks scientific significance in its present form. Therefore, I recommend rejecting the manuscript for publication in ACP.

To enhance the manuscript's scientific rigor, I suggest the authors make a series of major improvements which I have outlined below. Subsequently, I outline some more minor points for improvement.

**Author reply:**

We thank the Reviewer for the thoughtful comments and valuable suggestions that will contribute without doubt to improve our original manuscript. Based on these major and minor points below, we have revised our manuscript.

Major points

• In their introduction, the authors discuss the concept that the aromatic acids at the

centre of their study can be of both natural and anthropogenic origin. Although this is an interesting and important discussion, the authors could expand on the relative importance of the different sources. As it stands, it is unclear whether most of the acids considered in this study are of natural or anthropogenic origin or whether they differ for the different acids investigated. A table summarising this information could help clarify the discussion.

**Author reply:**

For the aromatic acids we studied, benzoic and hydroxybenzoic acids as well as vanillic and syringic acids in seawater have both natural and anthropogenic sources, and benzene dicarboxylic acids are mainly derived from anthropogenic sources. We have rewritten some sentences and summarized in a table as follows.

Line 41-45:

In various observations, aromatic acids have been detected in both natural and anthropogenic sources (Zhao et al., 2019; Zangrando et al., 2019; Dekiff et al., 2014). Among them, natural sources of aromatic acids produced by algal releases account for most of marine aromatic acids, especially benzoic acid, and most of hydroxybenzoic acids (Mostafa et al., 2017; Fotso Fondja Yao et al., 2010a; Castillo et al., 2023; Abdel-Hamid A. Hamdy, 2020).

Line 50-53:

This is consistent with previous researches that phthalic acid is primarily derived from anthropogenic sources (Ren et al., 2023a), whereas hydroxybenzoic acid has both anthropogenic and natural sources (Zhao et al., 2019; Castillo et al., 2023).

Line 89-90:

Further details are provided in Table S1, which lists sources and concentrations of these aromatic acid identified in seawater and atmospheric samples over the ocean.

**Table S1.** Sources and concentrations of aromatic acids identified in seawater and atmospheric samples over the ocean.

| Aromatic acids | Natural sources | Anthropogenic sources |
|---|---|---|
| benzoic acid | <li>sea algae (Abdel-Hamid A. Hamdy, 2020; Al-Zereini et al., 2010; Fotso Fondja Yao et al., 2010b; Liu et al., 2022b)</li><li>sedimentary organic matter (10–65 $\mu$g g$^{-1}$) (Deshmukh et al., 2016)</li><li>bacteria isolated from sea bass viscera (314 ppb) (Martí-Quijal et al., 2020)</li><li>snow pit samples (2.11 ng g$^{-1}$) (Mochizuki et al., 2016)</li> | <li>emerging endocrine disrupting compounds (34–491 ng L$^{-1}$) (Zhao et al., 2019)</li><li>fuel combustion (Boreddy et al., 2017)</li><li>industrial wastewater, automobile exhaust and tobacco smoke (Cuadros-Orellana et al., 2006)</li> |
| *o*-phthalic acid | | <li>plasticizer (16.7–657 ng g$^{-1}$ d.w.) (Ren et al., 2023b; Sanjuan et al., 2023);</li><li>plastic waste burning (8.3–84.9 ng m$^{-3}$) (Zhu et al., 2022)</li><li>the end product of photochemical oxidation of SOA (15.5 ng m$^{-3}$) (Ding et al., 2021)</li><li>biomass burning and fossil fuel combustion sources (0.4–7.9 ng m$^{-3}$) (Shumilina et al., 2023; Yang et al., 2020; Boreddy et al., 2022)</li> |

| | | |
|---|---|---|
| *m*-phthalic acid | | • plasticizer (Ren et al., 2023b)
• the end product of photochemical oxidation of SOA (3.6 ng m$^{-3}$) (Ding et al., 2021)
• biomass burning and fossil fuel combustion sources (0.01–2.3 ng m$^{-3}$) (Yang et al., 2020; Boreddy et al., 2022; Kawamur, 2014) |
| *p*-phthalic acid | | • plasticizer (0.51–6.8 mg kg$^{-1}$) (Ren et al., 2023b; Di Giacinto et al., 2023; Di Renzo et al., 2021)
• plastic waste burning (10.8–80.7 ng m$^{-3}$) (Zhu et al., 2022); the end product of photochemical oxidation of SOA (4.3 ng m$^{-3}$) (Ding et al., 2021)
• biomass burning and fossil fuel combustion sources (0.05–2.5 ng m$^{-3}$) (Yang et al., 2020; Boreddy et al., 2022; Kawamur, 2014) |
| *o*-hydroxybenzoic acid | • sea algae (76.8 mg L$^{-1}$) (Castillo et al., 2023; Mostafa et al., 2017; Klejdus et al., 2017) | • pharmaceuticals and drugs of abuse (0.4–53.3 ng L$^{-1}$) (Alygizakis et al., 2016) |
| *m*-hydroxybenzoic acid | • sea algae (Al-Zereini et al., 2010; Castillo et al., 2023) | |

| | | |
|---|---|---|
| *p*-hydroxybenzoic acid | • sea algae (57.7 mg $L^{-1}$) (Castillo et al., 2023; Klejdus et al., 2017; Tian et al., 2012; Hawas and Abou El-Kassem, 2017)
• sea fungus (Rukachaisirikul et al., 2010; Shao et al., 2007)
• sponge Mycale species (Xuefeng Zhou, 2013); metabolite (Jingchuan Xue, 2015; Liao and Kannan, 2018)
• sediment samples (6.85–437 ng $g^{-1}$ dw) (Liao et al., 2019) | • Pharmaceuticals and personal care products (Lu et al., 2023)
• emerging endocrine disrupting compounds (4.58–49.9 ng $L^{-1}$) (Zhao et al., 2019; Lu et al., 2021; Alygizakis et al., 2016) |
| vanillic acid | • sea algae (3–47 ng $L^{-1}$) (Zangrando et al., 2019; Klejdus et al., 2017)
• lignin decomposition (Wang et al., 2015; Hu et al., 2022; Xu et al., 2017) | • combustion of both softwood and hardwood (Simoneit, 2022) |
| syringic acid | • sea algae (0.3–0.6 ng $L^{-1}$) (Poznyakovsky et al., 2021; Zangrando et al., 2019; Klejdus et al., 2017)
• lignin decomposition (Hu et al., 2022; Xu et al., 2017) | • pharmaceuticals (Fisch et al., 2017)
• hardwood burning (Simoneit, 2022) |

• The description of the process of sea spray aerosol formation could be improved. For example, the initial description on line 57 is rather cumbersome and lacks accuracy. The authors can draw inspiration from the wealth of literature available on this topic to improve this section.

**Author reply:**

We have modified the original text as follows.

Line 59-67:

Sea spray aerosols (SSAs), generated by breaking waves and bubble bursting, are one of the major sources of atmospheric particles (Andreae and Rosenfeld, 2008; Angle et al., 2021; Hasenecz et al., 2020; Malfatti et al., 2019). SSAs can act as carrier agents for the vertical transport of much more than just sea salt and often include organic surfactants in the ocean, as already shown by field and laboratory studies (Cochran et al., 2016; Franklin et al., 2022; Rastelli et al., 2017). Recent data indicated that the surface activity and octanol-water partitioning coefficients ($K$ow) of organic compounds may affect their transport efficiency from the water phase to the atmosphere (Olson et al., 2020). Moreover, molecular structure may induce changes of organic acids properties (i.e., surface tension, toxicity), which further affect their transport potential and global emission flux (Lee et al., 2017; Rastelli et al., 2017; Frossard et al., 2019; Van Acker et al., 2021).

• The authors used a common approach to mimic the process of sea spray aerosol formation in the laboratory, namely a sea spray simulation chamber. Although such systems have become quite common, there is diversity among the systems in use to the extent that it is important to be very precise in describing the particulars of the system in use. Since the characteristics of the different systems used can influence the properties of the aerosols generated, certain information must be conveyed to the reader. For example, the authors used a plunging jet-type system. It is important to include all the relevant details of the system used, such as the diameter of the nozzle through which the seawater flowed into the chamber, the type of pump used to generate the plunging jet, the distance between the exit of the nozzle and the surface of the seawater in the chamber, and the material from which the chamber itself was fabricated.

**Author reply:**

We have added more contents to make the manuscript more complete.

Lines 99-100:

A jet-based laboratory SSA simulation chamber, made of 316L stainless steel material, was used to mimic the SSA generation (Fig.S2).

Lines 104-106:

When used, plunging jets were cycled in seawater by the pump (Shenchen V6-6, China) at a flow rate of 1 L min$^{-1}$ through a stainless-steel nozzle (inner diameter 4.3 mm).

Line 103:

The detailed dimensions of the SSA simulation chamber are provided in Table S2.

**Table S2.** Dimensions and operating conditions of the SSA simulation chamber.

| Characteristic | Value |
|---|---|
| Nozzle diameter (mm) | 4.3 |
| Seawater depth (cm) | 15 |
| Seawater volume (L) | 9 |
| Headspace depth (cm) | 22 |
| Headspace volume (L) | 15 |
| Zero sweep air (L min$^{-1}$) | 3 |
| Headspace residence time (min) | 5 |
| Plunging jet flow rate (L min$^{-1}$) | 1 |

The reasons for the selection of the relevant parameters are as follows:

1) A stainless steel nozzle with an inner diameter of 4.3 mm is used to generate plunging jets. The nozzle can be changed easily. The size of the nozzle is consistent with that in previous studies (Salter et al., 2014; Sha et al., 2021).

2) We have studied the cases with headspace heights of 14 cm, 19 cm, and 22 cm in our previous work (Liu et al., 2022a) and found that the SSA production was most efficient at a headspace height of 22 cm. Hence, we chose it for SSA generation.

3) We have tried pump flow rates of 0.50 L min$^{-1}$, 0.75 L min$^{-1}$, 1.00 L min$^{-1}$, 1.25 L min$^{-1}$, and 1.50 L min$^{-1}$ in our previous study (Liu et al., 2022a) and found that when the pump flow rate was 1 L min$^{-1}$ or higher, the performance of the SSA generator was good (see Table R1). Therefore, we used a pump flow rate of 1 L min$^{-1}$ to maintain a high SSA production rate and a relatively long working lifetime of the corresponding accessories.

Table R1. Effect of pump flow rate on SSA production.

| Pump flow rate (L min$^{-1}$) | SSA production (particles s$^{-1}$) | Mass concentration (µg m$^{-3}$) |
|:---:|:---:|:---:|
| 0.50 | $5.78 \times 10^6$ | 198.96 |
| 0.75 | $9.90 \times 10^6$ | 317.31 |
| 1.00 | $1.37 \times 10^7$ | 417.86 |
| 1.25 | $1.58 \times 10^7$ | 469.59 |
| 1.50 | $1.63 \times 10^7$ | 495.42 |

4) We have previously studied SSA production at purge air flow rates of 2 L min$^{-1}$, 3 L min$^{-1}$, 4 L min$^{-1}$, 5 L min$^{-1}$, 6 L min$^{-1}$ to simulate the sea breeze (Liu et al., 2022a). We found that the SSA production increased with the purge air flow rate when the flow rate was less than 3 L min$^{-1}$, and decreased with the increase of the purge air when the flow rate was more than 3 L min$^{-1}$.

5) Based on a previous study of the effect of relative humidity on the growth of sea salt particles, the inlet humidity would affect the morphology of SSA (Tang et al., 1997). Hence, we chose to keep the relative humidity at about 40% in our study. Particles will deliquescence at higher humidity (Bryan et al., 2022). In addition, it has been demonstrated that acid has no effect on NaCl deliquescence (Ming and Russell, 2001). Combining the above parameters, we obtained a particle size distribution similar to that from field observations (Quinn et al., 2017; Xu et al., 2022).

[Figure]

**Fig. S2.** Number size distribution of SSA generated with the SSA simulation chamber in this study compared with field studies.

• It is also important to note that seawater temperature has been shown to influence the process of sea spray aerosol formation. Therefore, it would be helpful to know whether the seawater sample temperature in the chamber was controlled or monitored. Additionally, it would be useful to know how the chamber was cleaned between experiments. Furthermore, in the schematic shown in Figure S1, a 'sweep flow' of particle-free air into the chamber is shown, while the only outlet was to the SMPS." The authors did not specify where the excess air went. Did they not operate with an "overflow" exit to ensure that the chamber was always operated at atmospheric pressure? Additionally, was the Dekati LPI connected to the same outlet as the SMPS system or was it connected to another outlet so that both measurements were conducted simultaneously?

**Author reply:**

1) Seawater temperature:

We controlled the laboratory temperature around 22–25°C during the experiments and examined the seawater temperature in the SSA simulation chamber before and after the experiments, and the corresponding results are supplemented in Table S3.

**Table S3.** Summary of experimental conditions.

| Exp. No. | Experiment type | Concentration (mM) | pH | Salinity (psu) | Sampling time (h) | RH (%) | Temperature difference (°C) [a] |
|---|---|---|---|---|---|---|---|
| 1 | SW | 0 | 7.92 | 34.2 | 5 | 35 | 2.0 |
| 2 | SW+benzoic acid | 1 | 7.72 | 34.3 | 5 | 34 | 1.5 |
| 3 | SW+$o$-hydroxybenzoic acid | 1 | 7.60 | 34.5 | 5 | 36 | 1.0 |
| 4 | SW+$m$-hydroxybenzoic acid | 1 | 7.68 | 34.1 | 5 | 40 | 2.0 |
| 5 | SW+$p$-hydroxybenzoic acid | 1 | 7.84 | 34.3 | 5 | 38 | 1.5 |
| 6 | SW+$o$-phthalic acid | 1 | 7.58 | 34.2 | 5 | 36 | 2.0 |
| 7 | SW+$m$-phthalic acid | 1 | 7.80 | 34.5 | 5 | 37 | 2.5 |
| 8 | SW+$p$-phthalic acid | 1 | 7.85 | 34.4 | 5 | 42 | 2.0 |
| 9 | SW+vanillic acid | 1 | 7.81 | 34.2 | 5 | 43 | 3.0 |
| 10 | SW+syringic acid | 1 | 7.84 | 34.3 | 5 | 39 | 2.0 |
| 11 | ASW | 0 | 7.96 | 35.1 | 5 | 33 | 1.5 |
| 12 | ASW+benzoic acid | 1 | 7.68 | 34.6 | 5 | 35 | 1.0 |
| 13 | ASW+$o$-hydroxybenzoic acid | 1 | 7.76 | 34.9 | 5 | 34 | 0.5 |
| 14 | ASW+$m$-hydroxybenzoic acid | 1 | 7.99 | 35.3 | 5 | 36 | 1.5 |
| 15 | ASW+$p$-hydroxybenzoic acid | 1 | 7.85 | 34.7 | 5 | 38 | 2.0 |
| 16 | ASW+$o$-phthalic acid | 1 | 7.93 | 34.5 | 5 | 35 | 1.0 |

| 17 | ASW+*m*-phthalic acid | 1 | 7.88 | 34.9 | 5 | 36 | 1.0 |
|---|---|---|---|---|---|---|---|
| 18 | ASW+*p*-phthalic acid | 1 | 7.97 | 34.6 | 5 | 34 | 1.5 |
| 19 | ASW+vanillic acid | 1 | 7.89 | 35.2 | 5 | 35 | 1.0 |
| 20 | ASW+syringic acid | 1 | 7.99 | 34.8 | 5 | 39 | 1.0 |
| 21 | ASW+benzoic acid+*o*-hydroxybenzoic acid+*o*-phthalic acid+vanillic acid+syringic acid | $10^{-3}$ | 7.95 | 35.1 | 20 | 41 | 3.5 |
| 22 | ASW+benzoic acid+*m*-hydroxybenzoic acid+*m*-phthalic acid+vanillic acid+syringic acid | $10^{-3}$ | 7.98 | 34.6 | 20 | 38 | 1.5 |
| 23 | ASW+benzoic acid+*p*-hydroxybenzoic acid+*p*-phthalic acid+vanillic acid+syringic acid | $10^{-3}$ | 7.88 | 34.9 | 20 | 40 | 2.0 |
| 24 | NaCl | 0 | 7.68 | 35.3 | 5 | 38 | 1.0 |
| 25 | NaCl+*m*-hydroxybenzoic acid | 1 | 7.54 | 34.7 | 5 | 36 | 1.5 |

[a] The temperature difference in the SSA simulation chamber before and after the experiment.

2) SSA simulation chamber cleaning:

Between experiments, the SSA simulation chamber was cleaned with ethanol first, then rinsed with ultrapure water, blown with zero air to reduce the influence of organic acids residue. The above steps run the pump to allow for thorough cleaning of the system.

**S1.** Quality assurance/quality control.

Seawater was collected from the coastal area of Shazikou on March 27, 2023, with a volume of 500 L (Fig. S1). Considering the storage inconvenience caused by huge consumption of seawater, all our seawater was pre-filtered through a polyethersulfone filter (47 mm diameter, 0.2 µm pore size, Supor®-200, Pall Life Sciences, USA) and stored in the dark at 18 °C for less than one month. Quinn et al. (2015) have shown that the fraction that passes through the filter is regarded as dissolved organic carbon and includes colloidal and truly dissolved materials. For each experiment, we measured particle number concentrations generated by filtered seawater and cations concentrations in seawater, and we found good agreement between each set of experiments (see Fig. S2).

In order to avoid the influence of organic matter in quartz fiber filters and access the accuracy of the experiment, pre-baked quartz fiber filters were used in sampling. Before each set of experiments, experimental blanks were conducted using filtered seawater. Experimental blanks were conducted with the same procedure of SSA samples. Seawater and filter samples were stored at -20°C until analyzed. In order to reduce the influence of organic acids residue after each experiment, the SSA simulation chamber was cleaned with ethanol first, then the system was cleaned with ultra-pure water for several times. The above steps also run the pump to allow for thorough cleaning of the system. Thereafter, the system was blown with zero air and sealed for preservation. The Dekati DLPI was also ultrasonicated with methanol and water (V:V=1:1) and dried after the experiment.

Filtered seawater (without added aromatic acid) was used as the experimental blank, and the same experimental and analytical methods were used as those for the experimental samples. As a result, no target aromatic acid was found in both seawater and filters. This may be due to the fact that we did not perform any concentration operation during the seawater sample processing. The standard curves for each aromatic acid are linear, as shown in Fig. S5.

[Figure]

**Fig. S1.** Sampling site at Shazikou along the Yellow Sea coast, Qingdao, China.

[Figure]

**Fig. S2.** Number size distribution of SSA generated with the SSA simulation chamber in this study compared with field studies.

[Figure]

**Fig. S5.** The standard curves for aromatic acids were constructed within a concentration range of 0.01-1000 μM, with more than seven data points.

3) Experimental setup:

We used a three-way valve to allow excess gas to escape, ensuring that the chamber operates at atmospheric pressure. The DeKati DLPI+ was connected to the same outlet as the SMPS system. The stabilization of the particles total number concentration suggests that there is little SSA particle adsorption in the chamber. The decrease in aromatic acid concentration may then be due to the transfer of SSA. Therefore, the sample collection for the SMPS system and the DeKati DLPI+ were conducted separately.

We inserted these sentences and figures in the revised manuscript and supplement to make it clearer.

Lines 111-112:

The outlet was fitted with a three-way valve to transfer the sample airflow and the excess gas.

Lines 128-129:

After the size distribution stabilized, a total of 40 mL of seawater is collected from a tap located 1.5 cm above the bottom of the chamber.

[Figure]

**Fig. S3.** Schematic picture of the plunging jet-sea spray aerosol generator: SMPS sampling (A), single particle sampling (B), and DeKati DLPI+ sampling (C). The red arrows represent the flow direction of seawater, and the purple arrows represent the flow of gases and aerosol particles.

• When evaluating data from sea spray simulation systems, it is important to consider the size of the bubble plume generated by the plunging jet compared to the chamber itself. For instance, did the bubble plume reach the bottom of the chamber? If the surface foam patch is also relatively large in comparison to the surface area of the water, then "wall effects" can occur whereby bubbles reach the side and burst faster than they would if the sides had not affected the bubbles. Therefore, some insight here (perhaps some photographs) would add value to the manuscript.

**Author reply:**

1) The bubble plume reached the bottom of the SSA simulation chamber.

2) The physical diagram of the experiment and the top view of the bubble bursting are shown below. Furthermore, due to the similar enrichment levels of 1 µM and 1 mM aromatic acids in SSA, we speculate that the main factor influencing the enrichment factor is aerosolization rather than the "wall effect".

The following text was added in the revised manuscript.

Line 107-108:

During the experiment, it was assumed that the interaction of the bubble plume generated by the plunging jet with the wall was negligible (Fig. S4).

[Figure]

**Fig. S4.** Physical diagram of the SSA simulation chamber (A) and the top view of the bubble generation in the chamber (B).

• Regarding the use of a Dekati LPI and a TEM sampler to obtain offline samples for subsequent analysis, the authors did not mention obtaining blank samples to check for handling impacts. It is unclear whether blank substrates were loaded into the samplers and then analyzed in the same way as the samples.

**Author reply:**

During the experiment, we used filtered seawater (no aromatic acids added) as the experimental blank and then analyzed in the same way as the samples. We measured the concentration of aromatic acid in filtered seawater and filters all below the detection limit. The TEM images of the filtered seawater (no aromatic acids added) have been shown in Fig. 4A.

[Figure]

**Fig. 4.** Particle morphology observed using TEM of sea salt (A) and mixed particles composed of aromatic acids-coated sea salt particles (B–K).

• When it comes to the results of their experiments, the authors first report the impact of different organic acids on surface tension. However, they did not provide enough information to discern how the experiments were carried out. It is assumed that the authors added or "spiked" a known concentration of the different compounds to the seawater samples they collected from the Yellow Sea. However, the concentration of the different compounds used, whether the seawater samples already contained some or all of the compounds, and the variability of the concentration of the analytes of interest in these seawater samples are not mentioned. Additionally, the presence of other surfactants in the form of organic matter derived from natural and anthropogenic sources, and how they could affect the results, is not discussed. All of this information is critical to helping the reader discern the implications of the results. Since natural seawater has variable amounts of these compounds, it is suggested that the authors could have "spiked" the analytes of interest into artificial seawater to try to negate this

effect.

**Author reply:**

1) The experiment consisted of a total of 25 sets with target compound concentrations of $10^{-3}$ and 1 mM (Table S3). Each aromatic acid was added individually to seawater for the experiment. The SMPS recorded particle number concentrations, and sampling was performed once the size distributions stabilized. Seawater samples, totaling 40 mL, were collected from a tap located 1.5 cm above the bottom of the chamber. After collection, quartz fiber filter (QFF) containing aromatic acids was subjected to a 40-minute ultrasonic extraction in 4 mL of Milli-Q water to dissolve submicron SSA particles. The solution was then filtered through a 0.45 μm syringe filter to eliminate filter fragments, for subsequent concentration calculations and EF analysis. After each experiment, the SSA chamber underwent thorough cleaning as outlined in QA/QC (see S1).

We have inserted the following text in the revised manuscript to make the experimental process clearer.

Lines 176-179:

The experiment consisted of a total of 25 sets with target compound concentrations of $10^{-3}$ and 1 mM (Table S3). Each aromatic acid was added individually to seawater for the experiment. Surface tension of seawater was examined using a surface tensiometer (Sigma 700, Biolin Scientific, Sweden) equipped with a Wihelmy plate, which was calibrated at 25 °C with 30 mL ultrapure water.

Lines 128-129:

After the size distribution stabilized, a total of 40 milliliters of seawater was collected from a tap located 1.5 cm above the bottom of the chamber.

**Table S3.** Summary of experimental conditions.

| Exp. No. | Experiment type | Concentration (mM) | pH | Salinity (psu) | Sampling time (h) | RH (%) | Temperature difference (°C) [a] |
|---|---|---|---|---|---|---|---|
| 1 | SW | 0 | 7.92 | 34.2 | 5 | 35 | 2.0 |
| 2 | SW+benzoic acid | 1 | 7.72 | 34.3 | 5 | 34 | 1.5 |
| 3 | SW+*o*-hydroxybenzoic acid | 1 | 7.60 | 34.5 | 5 | 36 | 1.0 |
| 4 | SW+*m*-hydroxybenzoic acid | 1 | 7.68 | 34.1 | 5 | 40 | 2.0 |
| 5 | SW+*p*-hydroxybenzoic acid | 1 | 7.84 | 34.3 | 5 | 38 | 1.5 |
| 6 | SW+*o*-phthalic acid | 1 | 7.58 | 34.2 | 5 | 36 | 2.0 |
| 7 | SW+*m*-phthalic acid | 1 | 7.80 | 34.5 | 5 | 37 | 2.5 |
| 8 | SW+*p*-phthalic acid | 1 | 7.85 | 34.4 | 5 | 42 | 2.0 |
| 9 | SW+vanillic acid | 1 | 7.81 | 34.2 | 5 | 43 | 3.0 |
| 10 | SW+syringic acid | 1 | 7.84 | 34.3 | 5 | 39 | 2.0 |
| 11 | ASW | 0 | 7.96 | 35.1 | 5 | 33 | 1.5 |
| 12 | ASW+benzoic acid | 1 | 7.68 | 34.6 | 5 | 35 | 1.0 |
| 13 | ASW+*o*-hydroxybenzoic acid | 1 | 7.76 | 34.9 | 5 | 34 | 0.5 |
| 14 | ASW+*m*-hydroxybenzoic acid | 1 | 7.99 | 35.3 | 5 | 36 | 1.5 |
| 15 | ASW+*p*-hydroxybenzoic acid | 1 | 7.85 | 34.7 | 5 | 38 | 2.0 |
| 16 | ASW+*o*-phthalic acid | 1 | 7.93 | 34.5 | 5 | 35 | 1.0 |

| 17 | ASW+$m$-phthalic acid | 1 | 7.88 | 34.9 | 5 | 36 | 1.0 |
|---|---|---|---|---|---|---|---|
| 18 | ASW+$p$-phthalic acid | 1 | 7.97 | 34.6 | 5 | 34 | 1.5 |
| 19 | ASW+vanillic acid | 1 | 7.89 | 35.2 | 5 | 35 | 1.0 |
| 20 | ASW+syringic acid | 1 | 7.99 | 34.8 | 5 | 39 | 1.0 |
| 21 | ASW+benzoic acid+$o$-hydroxybenzoic acid+$o$-phthalic acid+vanillic acid+syringic acid | $10^{-3}$ | 7.95 | 35.1 | 20 | 41 | 3.5 |
| 22 | ASW+benzoic acid+$m$-hydroxybenzoic acid+$m$-phthalic acid+vanillic acid+syringic acid | $10^{-3}$ | 7.98 | 34.6 | 20 | 38 | 1.5 |
| 23 | ASW+benzoic acid+$p$-hydroxybenzoic acid+$p$-phthalic acid+vanillic acid+syringic acid | $10^{-3}$ | 7.88 | 34.9 | 20 | 40 | 2.0 |
| 24 | NaCl | 0 | 7.68 | 35.3 | 5 | 38 | 1.0 |
| 25 | NaCl+$m$-hydroxybenzoic acid | 1 | 7.54 | 34.7 | 5 | 36 | 1.5 |

[a] The temperature difference in the SSA simulation chamber before and after the experiment.

2) During the experimental process, filtered seawater (without the addition of aromatic acids) was employed as a procedural blank. Subsequently, analysis was conducted using the same methods as applied to the samples. However, the targeted aromatic acids were not detected in the seawater, possibly due to their concentrations being below the detection limit. We have added relevant information in the Supplement.

Filtered seawater (without added aromatic acid) was used as the experimental blank, and the same experimental and analytical methods were used as those for the experimental samples. As a result, no aromatic acid was found in both seawater and filtered samples. This may be due to the fact that we did not perform any concentration operation during the water sample processing.

3) ASW were selected as the bulk water to characterize the influence of organic matter in our previous study (Song et al., 2022). We concluded that considering ASW as bulk waters will result in a deviation of SSA production from that produced by seawater. And the particles size distribution obtained by the lognormal-mode-fitting approach in our previous work (Fig. S2) is very similar to the distributions observed in previous studies (Xu et al., 2022; Quinn et al., 2017).

[Figure]

**Fig. S2.** Number size distribution of SSA generated with the SSA simulation chamber in this study compared with field studies.

Additionally, to enhance the rigor of our results, we have "spiked" the target analytes into artificial seawater, and the following text has been added in the revised manuscript.

Lines 177-179:

Each aromatic acid was added individually to seawater and artificial seawater (ASW) for the experiment. Surface tension of seawater was examined using a surface tensiometer (Sigma 700, Biolin Scientific, Sweden) equipped with a Wihelmy plate, which was calibrated at 25 °C with 30 mL ultrapure water.

Lines 181-183:

However, the average surface tension value of ASW was 74.5 mN m$^{-1}$ (Fig. S6), indicating that the presence of surfactants in seawater enhances its surface activity.

[Figure]

**Fig. S6.** Measured surface tension values of artificial seawater (ASW) and aromatic acid-containing ASW.

From the results, we can see that the effect trend of aromatic acids on seawater surface tension is generally similar, not only in seawater but also in artificial seawater.

Furthermore, we also measured the SSA particle number concentration with aromatic acids and the EF of aromatic acids, the results are as follows.

[Figure]

**Fig. S7.** Number concentration distribution of sea salt particles and SSA particles containing benzoic acids (A), benzene dicarboxylic acids (B), hydroxybenzoic acids (C), vanillic acid and syringic acid (D).

From the results, we can see that the effect trend of aromatic acids on SSA production is generally similar, not only in seawater but also in artificial seawater.

[Figure]

**Fig. S8.** Enrichment factors of aromatic acids at different concentrations from artificial seawater to the atmosphere.

Comparing the EF of aromatic acids in SW and ASW, it was observed that the EF trends of benzene dicarboxylic acids in seawater follows the pattern: *o*-phthalic acid < *m*-phthalic acid < *p*-phthalic acid. However, in ASW, the EF of *p*-phthalic acid was lower than that of *m*-phthalic acid. Based on the findings of Li et al. (2023), we hypothesize that in ASW, *p*-phthalic acid acts as an ·OH scavenger to produce TAOH. Hence, the EF of *p*-phthalic acid is lower than that of *m*-phthalic acid. Meanwhile, organic compounds in SW preferentially react with ·OH (Anastasio and Newberg, 2007), thus the EF of *p*-phthalic acid is the highest among the benzene dicarboxylic acids. Furthermore, differences in aromatic acid concentration did not change the enrichment pattern of organic acids.

• The authors then report the impact of different acids on the size and number of particles that are emitted as SSA. It is unclear what the concentrations of the different analytes in these experiments were. It is also unclear whether the same seawater was used for all experiments or whether different seawater samples were spiked. The concentration of the analytes in the seawater sample and the level of organic matter in the seawater are not mentioned.

**Author reply:**

1) We added 1 mM of individual aromatic acids to natural seawater. In artificial seawater, $10^{-3}$ and 1mM aromatic acids were added, and the specific experimental groups can be found in Table S3.

Lines 176-177:

The experiment consisted of a total of 25 sets with target compound concentrations of $10^{-3}$ and 1 mM (Table S3).

2) We used the same seawater for all experiments. Seawater was collected from the coastal area of Shazikou on March 27, 2023, with a volume of 500 L (Fig. S1). Considering the storage inconvenience caused by huge consumption of seawater, all our seawater was pre-filtered through a polyethersulfone filter (47 mm diameter, 0.2 µm pore size, Supor®-200, Pall Life Sciences, USA) and stored in the dark at 18 °C for less than one month. For each experiment, we measured particle number concentrations generated by filtered seawater and cations concentrations in seawater, and we found good agreement between each set of experiments (see Fig. R1). Furthermore, for comparing the properties of SSA particles containing aromatic acids, we normalized the particles size distribution of seawater before adding aromatic acids.

[Figure]

**Fig. R1.** Mean value of total particle number concentration of SSA ($N_{Total}$) and concentration of $Na^+$ ($C(Na^+)$) for each experiment.

3) We measured the standard curve for aromatic acids and it was linear over the measurement range. Therefore, we used the peak area directly for the EF calculation.

[Figure]

**Fig. S5.** Standard curves for aromatic acids were constructed within a concentration range of 0.01-1000 μM, with more than seven data points.

**Table R2.** Peak areas of aromatic acids in seawater and SSA filter samples.

| | Group 1 | | Group 2 | | Group 3 | | Group 4 | |
|---|---|---|---|---|---|---|---|---|
| | SW | SSA | SW | SSA | SW | SSA | SW | SSA |
| benzoic acid | 159632 | 2732.77 | 196860 | 5206.99 | 185135 | 1540.1 | 167892 | 2508.47 |
| *o*-phthalic acid | 123785 | 1226.6 | 168974 | 1802.2 | 156324 | 1888.7 | | |
| *m*-phthalic acid | 466987 | 6606.85 | 458769 | 5432.25 | 491623 | 4880.15 | | |
| *p*-phthalic acid | 454593 | 10626.92 | 496102 | 21471.43 | 445985 | 15614.82 | 471582 | 20110.87 |
| *o*-hydroxybenzoic acid | 257548 | 2482 | 232946 | 3844.08 | 238358 | 5167.06 | | |
| *m*-hydroxybenzoic acid | 170482 | 8532.59 | 160309 | 7072.09 | 190181 | 4732.93 | | |
| *p*-hydroxybenzoic acid | 169753 | 3635.99 | 153257 | 4467.54 | 131186 | 2297.87 | 171062 | 4329.58 |
| vanillic acid | 357000 | 4948.01 | 287470 | 4696.83 | 279462 | 12750.11 | | |
| syringic acid | 486640 | 23833.92 | 467129 | 29328.75 | 491816 | 11993.31 | | |

• Following this, the authors present enrichment factors of the different organic acids on the nascent sea spray aerosol they generate. However, many details are again missing, which makes it impossible to evaluate the results. For example, it is not clear to me whether the presented results are the average over all stages of the impactor or if they only represent a single stage. Furthermore, the authors did not mention how they quantified the sodium ion and the concentration of the organic acids in the same sample. Did the authors extract the substrates in ultrapure water and then subsample these extracts for the different analyses, or did they use a different approach? To generate their enrichment factor estimates, the authors will have used a concentration value for each of the acids in the seawater used to generate the aerosols. Was this value quantified, or did the authors simply use the "spike" concentration they aimed for? This is important given that surface active species, such as the organic acids used, have a tendency to stick to the walls of chambers and the actual concentration of the organic acids in the seawater may well have been well below the "spike" value the authors aimed for. Along the same lines, assuming that the authors did quantify the actual concentration of the organic acids in the seawater, it would be good to know at what point of the experiment they obtained these samples. For example, the authors state that the LPI was run for 5 hours - were water samples taken at the start or end of this period? In similar experiments using perfluoroalkyl acids, Johannsson et al. (2019) observed that the concentration of the substances in the seawater from which they were generating aerosol during their experiment decreased as these substances were "lost" to the aerosol.

**Author reply:**

1) Impactor sampling:

The present results are averaged over all stages of the impactor with particle size less than 1 μm.

Lines 144-145:

Submicron SSA particles were impacted onto 25-mm diameter quartz fiber filters (QFF, 1851–025, Waterman, UK), which were baked in muffle furnace at 450 ℃ for 3 h.

2) Quantification of concentration:

We extracted the substrates in ultrapure water and then subsampled these extracts for analyses. We further measured the standard curve for the aromatic acids and it was linear over the measurement range. Therefore, we used the peak area directly for the subsequent EF calculation.

**Table R2.** Peak areas of aromatic acids in seawater and SSA filter samples.

| | Group 1 | | Group 2 | | Group 3 | | Group 4 | |
|---|---|---|---|---|---|---|---|---|
| | SW | SSA | SW | SSA | SW | SSA | SW | SSA |
| benzoic acid | 159632 | 2732.77 | 196860 | 5206.99 | 185135 | 1540.1 | 167892 | 2508.47 |
| *o*-phthalic acid | 123785 | 1226.6 | 168974 | 1802.2 | 156324 | 1888.7 | | |
| *m*-phthalic acid | 466987 | 6606.85 | 458769 | 5432.25 | 491623 | 4880.15 | | |
| *p*-phthalic acid | 454593 | 10626.92 | 496102 | 21471.43 | 445985 | 15614.82 | 471582 | 20110.87 |
| *o*-hydroxybenzoic acid | 257548 | 2482 | 232946 | 3844.08 | 238358 | 5167.06 | | |
| *m*-hydroxybenzoic acid | 170482 | 8532.59 | 160309 | 7072.09 | 190181 | 4732.93 | | |
| *p*-hydroxybenzoic acid | 169753 | 3635.99 | 153257 | 4467.54 | 131186 | 2297.87 | 171062 | 4329.58 |
| vanillic acid | 357000 | 4948.01 | 287470 | 4696.83 | 279462 | 12750.11 | | |
| syringic acid | 486640 | 23833.92 | 467129 | 29328.75 | 491816 | 11993.31 | | |

3) Sampling time:

We obtained seawater samples after the SMPS-monitored total number concentrations had stabilized, and prior to DLPI+ sampling.

Lines 128-129:

After the size distribution stabilized, a total of 40 milliliters of seawater is collected from a tap located 1.5 cm above the bottom of the chamber.

• On Line 84, the authors introduce the idea of upscaling their measurements using literature values for the seawater concentration of these compounds. However, the reader requires more information at this juncture. Questions arise, such as: What are the typical seawater concentrations of these compounds? How much do they vary

globally? How extensive is the dataset of such measurements? Incorporating this information into the previously introduced table (major point 1) would be beneficial. While the authors eventually provide some literature values in section 3.4 of the manuscript, it remains unclear which specific values were utilised. Was it an average of the literature values or a different approach? Considering the likely high spatial (and potentially temporal) variability in these concentrations, it is crucial to understand how the authors addressed this variability. Presently, it is nearly impossible to evaluate the magnitude of the emissions proposed by the authors due to the uncertainty in the conducted measurements.

**Author reply:**

Based on the concentration values summarized from Table S1 in our study, we calculated the annual emission flux of SSA within the concentration range reported in the literature. For concentration selection, we chose the actual detected concentrations of aromatic acids in seawater rather than the concentrations found in sea algal extracts. The concentrations we used are shown in Table S5. We calculated the range of aromatic acid emissions within the concentration range reported in the literature (Zangrando et al., 2019; Zhao et al., 2019). The following reference has been cited in the revised manuscript.

Lines 351-353:

For example, the seawater concentrations of syringic acid, *p*-hydroxybenzoic acid, benzoic acid are 0.3, 4.58, 34 ng L$^{-1}$, respectively (Zhao et al., 2019).

[Figure]

**Fig. 7.** Estimated annual global aromatic acids emission (tons yr⁻¹) via SSA. Yellow and blue stacked columns represent emissions based on Textor et al. (2006) and Jonas et al. (2021), respectively.

**Minor points**

• Line 30: I don't think "captured" is the right word here. Perhaps "taken up" would be better.

**Author reply:**

We have replaced "captured" by "taken up" in the revised manuscript.

Lines 30-32:

On the one hand, aromatic acids are readily to be taken up by marine organisms, leading to their enrichment within these organisms or transportation to remote areas (Fu et al., 2011; Shariati et al., 2021; Wang and Kawamura, 2006).

• Line 31: Would read better as: "...leading to their enrichment within these organisms or transportation to remote areas (Fu et al.,..."

**Author reply:**

We have modified corresponding sentences.

• Line 31: I would also break this sentence to aid readability, e.g.: "This process poses health risks to the endocrine system of aquatic organisms and the overall marine ecosystem (Saha et al., 2006)."

**Author reply:**

We have modified corresponding sentences. Please, refer to our reply to the first minor comment for the detailed changes.

Lines 32-33:

This process poses health risks to the endocrine system of aquatic organisms and the overall marine ecosystem (Saha et al., 2006).

• Line 35: Would read better as "Previous studies suggest that these organic acids can alter the composition of SSA, subsequently influencing atmospheric processes such as cloud condensation nuclei (CCN) or ice nuclei (IN) activities.

**Author reply:**

We have modified corresponding sentences in the revised manuscript.

Lines 35-38:

Previous studies suggest that these organic acids can alter the composition of SSA, subsequently influencing atmospheric processes such as cloud condensation nuclei (CCN) or ice nuclei (IN) activities.

• Line 36: Needs rephrasing. It is not the "organic acids" that are the source of the aerosol. Rather, it is SSA that may contain organic acids that play a role in Earth's climate.

**Author reply:**

We thank the Reviewer for pointing this out. A new sentence has been shown in the revised manuscript.

Lines 38-39:

SSA may contain organic acids that play a major role in Earth's climate (Moore et al., 2011; Zhu et al., 2019).

• Line 50: Would read better as "...the identification of phthalic acid in organisms raises the possibility of its origin from the ingestion of marine plastics."

**Author reply:**

We have modified corresponding sentences in the revised manuscript.

Lines 49-50:

Importantly, in addition to biodegradation, the identification of phthalic acid in organisms raises the possibility of its origin from the ingestion of marine plastics (Almulhim et al., 2022).

• Line 53: Would read better as "This phenomenon has prompted discussions about the concept of" missing aromatic acids."

**Author reply:**

We have modified corresponding sentences in the revised manuscript.

Lines 53-55:

Recent laboratory studies have shown that personal-care products, especially sunscreen (e.g., $o$-hydroxybenzoic acid), are reduced in levels during algal blooms (Franklin et al., 2022). This phenomenon has prompted discussions about the concept of "missing aromatic acids".

• Line 54: Would read better as "Nevertheless, the currently available data do not provide a conclusive explanation for the existence of these" missing aromatic acids."

**Author reply:**

We have modified corresponding sentences in the revised manuscript.

Lines 55-56:

Nevertheless, the currently available data do not provide a conclusive explanation for the existence of these "missing aromatic acids".

• Line 55: This paragraph could be better linked to the previous discussion. For example,"One possible reason for the disappearance of these aromatic acids is their release into the atmosphere. Existing datasets obtained from remote marine areas offer evidence of the presence of these compounds in the atmosphere (Fu et al., 2010)."

**Author reply:**

We have modified corresponding sentences in the revised manuscript.

Lines 57-59:

One possible reason for the disappearance of these aromatic acids is their release into the atmosphere. Existing datasets obtained from remote marine areas offer evidence of the presence of these compounds in the atmosphere (Fu et al., 2010).

• Line 66: Would read better: "While field studies have demonstrated the presence of aromatic acids in both the ocean and atmosphere (Boreddy et al., 2017), the specific mechanisms influencing their transport at the air-sea interface require further investigation."

**Author reply:**

We have modified corresponding sentences in the revised manuscript.

Lines 67-69:

While field studies have demonstrated the presence of aromatic acids in both the ocean and atmosphere (Boreddy et al., 2017), the specific mechanisms influencing their transport at the air-sea interface require further investigation.

• Line 69: Here the authors introduce the air-sea transport of perfluoroalkyl acids (PFAAs) via sea spray aerosol. Although this is an interesting topic, the link to the work carried out in this study is unclear. I urge the authors to better describe this link or remove this reference.

**Author reply:**

This reference has been removed.

• Line 74: Would read better as "In a previous study, we observed that the transfer of short-chain organic acids between the air and sea through SSA may be influenced by seawater surface tension. This factor could, in turn, impact the enrichment behaviour of organic acids (Song et al., 2022)."

**Author reply:**

We have modified corresponding sentences in the revised manuscript.

Lines 72-74:

In a previous study, we observed that the transfer of short-chain organic acids between the air and sea through SSA may be influenced by seawater surface tension. This factor could, in turn, impact the enrichment behavior of organic acids (Song et al., 2022).

• Line 77: The authors state "...other factors were discussed in our recent review..." What are the other factors? If this discussion is important, then the authors should do it justice and include all relevant information. Otherwise, I see no need to mention this.

**Author reply:**

We have deleted this sentence in the Introduction of the revised manuscript.

• Line 78: There should be a new paragraph here: "In this study..."

**Author reply:**

We have modified it in the revised manuscript.

**References**

Johansson, Jana H., et al. "Global transport of perfluoroalkyl acids via sea spray aerosol." Environmental Science: Processes & Impacts 21.4 (2019): 635-649.

Sha, B., Johansson, J. H., Benskin, J. P., Cousins, I. T., and Salter, M. E.: Influence of water concentrations of perfluoroalkyl acids (PF AAs) on their size-resolved enrichment in nascent sea spray aerosols, Environ. Sci. Technol., 55, 9489–9497, 10.1021/acs.est.0c03804, 2021

**References**

Abdel-Hamid A. Hamdy, N. M. E.-f., A. El-Beih, M. Mohammed, W. Mettwally: Egyptian red sea seagrass as a source of biologically active secondary metabolites, Egyptian Pharmaceutical Journal 19, 224, 10.4103/epj.epj_57_19, 2020.

Al-Zereini, W., Fotso Fondja Yao, C. B., Laatsch, H., and Anke, H.: Aqabamycins A-G: novel nitro maleimides from a marine Vibrio species. I. Taxonomy, fermentation, isolation and biological activities, J. Antibiot., 63, 297–301, 10.1038/ja.2010.34, 2010.

Alygizakis, N. A., Gago-Ferrero, P., Borova, V. L., Pavlidou, A., Hatzianestis, I., and Thomaidis, N. S.: Occurrence and spatial distribution of 158 pharmaceuticals, drugs of abuse and related metabolites in offshore seawater, Sci. Total Environ., 541, 1097–1105, 10.1016/j.scitotenv.2015.09.145, 2016.

Anastasio, C. and Newberg, J. T.: Sources and sinks of hydroxyl radical in sea-salt particles, J. Geophys. Res. Atmos., 112, D10306, 10.1029/2006jd008061, 2007.

Andreae, M. O. and Rosenfeld, D.: Aerosol-cloud-precipitation interactions. Part 1. The nature and sources of cloud-active aerosols, Earth Sci. Rev., 89, 13–41, 10.1016/j.earscirev.2008.03.001, 2008.

Angle, K. J., Crocker, D. R., Simpson, R. M. C., Mayer, K. J., Garofalo, L. A., Moore, A. N., Mora Garcia, S. L., Or, V. W., Srinivasan, S., Farhan, M., Sauer, J. S., Lee, C., Pothier, M. A., Farmer, D. K., Martz, T. R., Bertram, T. H., Cappa, C. D., Prather, K. A., and Grassian, V. H.: Acidity across the interface from the ocean surface to sea spray aerosol, Proc. Natl. Acad. Sci. U.S.A., 118, e2018397118, 10.1073/pnas.2018397118, 2021.

Boreddy, S. K. R., Hegde, P., Arun, B. S., Aswini, A. R., and Babu, S. S.: Molecular composition and light-absorbing properties of organic aerosols from west-coast of tropical India, Sci. Total Environ., 845, 157163, 10.1016/j.scitotenv.2022.157163, 2022.

Boreddy, S. K. R., Mochizuki, T., Kawamura, K., Bikkina, S., and Sarin, M. M.: Homologous series of low molecular weight (C1-C10) monocarboxylic acids, benzoic acid and hydroxyacids in fine-mode (PM2.5) aerosols over the Bay of Bengal: Influence of heterogeneity in air masses and formation pathways, Atmos. Environ., 167, 170–180, 10.1016/j.atmosenv.2017.08.008, 2017.

Bryan, C. R., Knight, A. W., Katona, R. M., Sanchez, A. C., and Schindelholz, E. J.: Physical and chemical properties of sea salt deliquescent brines as a function of temperature and relative humidity, Sci. Total Environ., 824, 154462, 10.1016/j.scitotenv.2022.154462, 2022.

Castillo, A., Celeiro, M., Lores, M., Grgić, K., Banožić, M., Jerković, I., and Jokić, S.: Bioprospecting of Targeted Phenolic Compounds of Dictyota dichotoma, Gongolaria barbata, Ericaria amentacea, Sargassum hornschuchii and Ellisolandia elongata from the Adriatic Sea Extracted by Two Green Methods, Mar. Drugs, 21, 97, 10.3390/md21020097, 2023.

Cochran, R. E., Laskina, O., Jayarathne, T., Laskin, A., Laskin, J., Lin, P., Sultana, C., Lee, C., Moore, K. A., Cappa, C. D., Bertram, T. H., Prather, K. A., Grassian, V. H., and Stone, E. A.: Analysis of organic anionic surfactants in fine and coarse fractions of freshly emitted sea spray aerosol, Environ. Sci. Technol., 50, 2477-2486, 10.1021/acs.est.5b04053, 2016.

Cuadros-Orellana, S., Pohlschröder, M., and Durrant, L. R.: Isolation and characterization of halophilic archaea able to grow in aromatic compounds, Int. Biodeterior. Biodegradation, 57, 151–154, 10.1016/j.ibiod.2005.04.005, 2006.

Dekiff, J. H., Remy, D., Klasmeier, J., and Fries, E.: Occurrence and spatial distribution of microplastics in sediments from Norderney, Environ. Pollut., 186, 248–256, 10.1016/j.envpol.2013.11.019, 2014.

Deshmukh, D. K., Kawamura, K., Lazaar, M., Kunwar, B., and Boreddy, S. K. R.: Dicarboxylic acids, oxoacids, benzoic acid, α-dicarbonyls, WSOC, OC, and ions in spring aerosols from Okinawa Island in

the western North Pacific Rim: size distributions and formation processes, Atmos. Chem. Phys., 16, 5263–5282, 10.5194/acp-16-5263-2016, 2016.

Di Giacinto, F., Di Renzo, L., Mascilongo, G., Notarstefano, V., Gioacchini, G., Giorgini, E., Bogdanović, T., Petričević, S., Listeš, E., Brkljača, M., Conti, F., Profico, C., Zambuchini, B., Di Francesco, G., Giansante, C., Diletti, G., Ferri, N., and Berti, M.: Detection of microplastics, polymers and additives in edible muscle of swordfish (Xiphias gladius) and bluefin tuna (Thunnus thynnus) caught in the Mediterranean Sea, J. Sea Res., 192, 102359, 10.1016/j.seares.2023.102359, 2023.

Di Renzo, L., Mascilongo, G., Berti, M., Bogdanović, T., Listeš, E., Brkljača, M., Notarstefano, V., Gioacchini, G., Giorgini, E., Olivieri, V., Silvestri, C., Matiddi, M., D'Alterio, N., Ferri, N., and Di Giacinto, F.: Potential impact of microplastics and additives on the health status of loggerhead turtles (Caretta caretta) stranded along the central adriatic coast, Water Air Soil Pollut., 232, 98, 10.1007/s11270-021-04994-8, 2021.

Ding, Z., Du, W., Wu, C., Cheng, C., Meng, J., Li, D., Ho, K., Zhang, L., and Wang, G.: Summertime atmospheric dicarboxylic acids and related SOA in the background region of Yangtze River Delta, China: Implications for heterogeneous reaction of oxalic acid with sea salts, Sci. Total Environ., 757, 143741, 10.1016/j.scitotenv.2020.143741, 2021.

Fisch, K., Waniek, J. J., and Schulz-Bull, D. E.: Occurrence of pharmaceuticals and UV-filters in riverine run-offs and waters of the German Baltic Sea, Mar. Pollut. Bull., 124, 388–399, 10.1016/j.marpolbul.2017.07.057, 2017.

Fotso Fondja Yao, C. B., Zereini, W. A., Fotso, S., Anke, H., and Laatsch, H.: Aqabamycins A–G: novel nitro maleimides from a marine Vibrio species: II. Structure elucidation*, The Journal of Antibiotics, 63, 303-308, 10.1038/ja.2010.35, 2010a.

Fotso Fondja Yao, C. B., Zereini, W. A., Fotso, S., Anke, H., and Laatsch, H.: Aqabamycins A–G: novel nitro maleimides from a marine Vibrio species: II. Structure elucidation*, J. Antibiot., 63, 303–308, 10.1038/ja.2010.35, 2010b.

Franklin, E. B., Amiri, S., Crocker, D., Morris, C., Mayer, K., Sauer, J. S., Weber, R. J., Lee, C., Malfatti, F., Cappa, C. D., Bertram, T. H., Prather, K. A., and Goldstein, A. H.: Anthropogenic and biogenic contributions to the organic composition of coastal submicron sea spray aerosol, Environ. Sci. Technol., 56, 16633–16642, 10.1021/acs.est.2c04848, 2022.

Frossard, A. A., Gerard, V., Duplessis, P., Kinsey, J. D., Lu, X., Zhu, Y., Bisgrove, J., Maben, J. R., Long, M. S., Chang, R. Y., Beaupre, S. R., Kieber, D. J., Keene, W. C., Noziere, B., and Cohen, R. C.: Properties of seawater surfactants associated with primary marine aerosol particles produced by bursting bubbles at a model air-sea interface, Environ. Sci. Technol., 53, 9407–9417, 10.1021/acs.est.9b02637, 2019.

Hasenecz, E. S., Jayarathne, T., Pendergraft, M. A., Santander, M. V., Mayer, K. J., Sauer, J., Lee, C., Gibson, W. S., Kruse, S. M., Malfatti, F., Prather, K. A., and Stone, E. A.: Marine bacteria affect saccharide enrichment in sea spray aerosol during a phytoplankton bloom, ACS Earth Space Chem., 4, 1638–1649, 10.1021/acsearthspacechem.0c00167, 2020.

Hawas, U. W. and Abou El-Kassem, L. T.: Thalassiolin D: a new flavone O-glucoside Sulphate from the seagrass Thalassia hemprichii, Nat. Prod. Res., 31, 2369–2374, 10.1080/14786419.2017.1308367, 2017.

Hu, J., Loh, P. S., Chang, Y.-P., and Yang, C.-W.: Multi-proxy records of paleoclimatic changes in sediment core ST2 from the southern Zhejiang-Fujian muddy coastal area since 1650 yr BP, Cont Shelf Res, 239, 104717, 10.1016/j.csr.2022.104717, 2022.

Jingchuan Xue, N. S., Madhavan Elangovan, Guthrie Diamond, Kurunthachalam Kannan: Elevated accumulation of parabens and their metabolites in marine mammals from the United States coastal waters,

Environ. Sci. Technol., 49, 12071–12079, 2015.

Jonas, G., Augustin, M., Michael, S., Andrews, E., Balkanski, Y., Bauer, S. E., Benedictow, A. M. K., Bian, H., Checa-Garcia, R., Chin, M., Ginoux, P., Griesfeller, J. J., Heckel, A., Kipling, Z., Kirkevåg, A., Kokkola, H., Laj, P., Le Sager, P., Lund, M. T., Lund Myhre, C., Matsui, H., Myhre, G., Neubauer, D., van Noije, T., North, P., Olivié, D. J. L., Rémy, S., Sogacheva, L., Takemura, T., Tsigaridis, K., and Tsyro, S. G.: AeroCom phase III multi-model evaluation of the aerosol life cycle and optical properties using ground- and space-based remote sensing as well as surface in situ observations, Atmos. Chem. Phys., 21, 87–128, 10.5194/acp-21-87-2021, 2021.

Kawamur, B. K. a. K.: Seasonal distributions and sources of low molecular weight dicarboxylic acids, ωoxocarboxylic acids, pyruvic acid, α-dicarbonyls and fatty acids in ambient aerosols from subtropical Okinawa in the western Pacific Rim, Environ. Chem., 11, 673–689, 10.1071/EN14097_AC, 2014.

Klejdus, B., Plaza, M., Šnóblová, M., and Lojková, L.: Development of new efficient method for isolation of phenolics from sea algae prior to their rapid resolution liquid chromatographic–tandem mass spectrometric determination, J. Pharm. Biomed. Anal., 135, 87–96, 10.1016/j.jpba.2016.12.015, 2017.

Lee, H. D., Estillore, A. D., Morris, H. S., Ray, K. K., Alejandro, A., Grassian, V. H., and Tivanski, A. V.: Direct surface tension measurements of individual sub-micrometer particles using atomic force microscopy, J. Phys. Chem. A, 121, 8296–8305, 10.1021/acs.jpca.7b04041, 2017.

Li, K., Guo, Y., Nizkorodov, S. A., Rudich, Y., Angelaki, M., Wang, X., An, T., Perrier, S., and George, C.: Spontaneous dark formation of OH radicals at the interface of aqueous atmospheric droplets, Proc. Natl. Acad. Sci. U.S.A., 120, e2220228120, 10.1073/pnas.2220228120, 2023.

Liao, C. and Kannan, K.: Temporal trends of parabens and their metabolites in mollusks from the Chinese Bohai Sea during 2006–2015: Species-specific accumulation and implications for human exposure, Environ. Sci. Technol., 52, 9045–9055, 10.1021/acs.est.8b02750, 2018.

Liao, C., Shi, J., Wang, X., Zhu, Q., and Kannan, K.: Occurrence and distribution of parabens and bisphenols in sediment from northern Chinese coastal areas, Environ. Pollut., 253, 759–767, 10.1016/j.envpol.2019.07.076, 2019.

Liu, L., Du, L., Xu, L., Li, J., and Tsona, N. T.: Molecular size of surfactants affects their degree of enrichment in the sea spray aerosol formation, Environ. Res., 206, 112555, 10.1016/j.envres.2021.112555, 2022a.

Liu, S., Longnecker, K., Kujawinski, E. B., Vergin, K., Bolaños, L. M., Giovannoni, S. J., Parsons, R., Opalk, K., Halewood, E., Hansell, D. A., Johnson, R., Curry, R., and Carlson, C. A.: Linkages among dissolved organic matter export, dissolved metabolites, and associated microbial community structure response in the Northwestern Sargasso Sea on a seasonal scale, Front. Microbiol., 13, 833252, 10.3389/fmicb.2022.833252, 2022b.

Lu, S., Lin, C., Lei, K., Xin, M., Wang, B., Ouyang, W., Liu, X., and He, M.: Endocrine-disrupting chemicals in a typical urbanized bay of Yellow Sea, China: Distribution, risk assessment, and identification of priority pollutants, Environ. Pollut., 287, 117588, 10.1016/j.envpol.2021.117588, 2021.

Lu, S., Wang, J., Wang, B., Xin, M., Lin, C., Gu, X., Lian, M., and Li, Y.: Comprehensive profiling of the distribution, risks and priority of pharmaceuticals and personal care products: A large-scale study from rivers to coastal seas, Water Res., 230, 119591, 10.1016/j.watres.2023.119591, 2023.

Malfatti, F., Lee, C., Tinta, T., Pendergraft, M. A., Celussi, M., Zhou, Y. Y., Sultana, C. M., Rotter, A., Axson, J. L., Collins, D. B., Santander, M. V., Morales, A. L. A., Aluwihare, L. I., Riemer, N., Grassian, V. H., Azam, F., and Prather, K. A.: Detection of active microbial enzymes in nascent sea spray aerosol: Implications for atmospheric chemistry and climate, Environ. Sci. Technol. Lett., 6, 171–177,

10.1021/acs.estlett.8b00699, 2019.

Martí-Quijal, F. J., Tornos, A., Príncep, A., Luz, C., Meca, G., Tedeschi, P., Ruiz, M.-J., and Barba, F. J.: Impact of fermentation on the recovery of antioxidant bioactive compounds from sea bass byproducts, Antioxidants, 9, 239, 10.3390/antiox9030239, 2020.

Ming, Y. and Russell, L. M.: Predicted hygroscopic growth of sea salt aerosol, J. Geophys. Res. Atmos., 106, 28259-28274, 10.1029/2001jd000454, 2001.

Mochizuki, T., Kawamura, K., Aoki, K., and Sugimoto, N.: Long-range atmospheric transport of volatile monocarboxylic acids with Asian dust over a high mountain snow site, central Japan, Atmos. Chem. Phys., 16, 14621–14633, 10.5194/acp-16-14621-2016, 2016.

mostafa, s., Mohamed, H., Ibraheem, I., and Abdel-Raouf, N.: Controlling of microbial growth by using cystoseira barbata extract, Egypt. J. Bot., 57, 469–477, 10.21608/ejbo.2017.911.1071, 2017.

Olson, N. E., Cooke, M. E., Shi, J. H., Birbeck, J. A., Westrick, J. A., and Ault, A. P.: Harmful algal bloom toxins in aerosol generated from inland lake water, Environ. Sci. Technol., 54, 4769–4780, 10.1021/acs.est.9b07727, 2020.

Poznyakovsky, V., Kalenik, T., Wojciech, P., Tabakaeva, O., and Tabakaev, A.: Antioxidant properties of edible sea weed from the Northern Coast of the Sea of Japan, Foods Raw Mater., 9, 262–270, 10.21603/2308-4057-2021-2-262-270, 2021.

Quinn, P. K., Coffman, D. J., Johnson, J. E., Upchurch, L. M., and Bates, T. S.: Small fraction of marine cloud condensation nuclei made up of sea spray aerosol, Nat. Geosci., 10, 674–679, 10.1038/Ngeo3003, 2017.

Quinn, P. K., Collins, D. B., Grassian, V. H., Prather, K. A., and Bates, T. S.: Chemistry and related properties of freshly emitted sea spray aerosol, Chem. Rev., 115, 4383-4399, 10.1021/cr500713g, 2015.

Rastelli, E., Corinaldesi, C., Dell'Anno, A., Lo Martire, M., Greco, S., Cristina Facchini, M., Rinaldi, M., O'Dowd, C., Ceburnis, D., and Danovaro, R.: Transfer of labile organic matter and microbes from the ocean surface to the marine aerosol: an experimental approach, Sci. Rep., 7, 11475, 10.1038/s41598-017-10563-z, 2017.

Ren, L., Weng, L., Chen, D., Hu, H., Jia, Y., and Zhou, J. L.: Bioremediation of PAEs-contaminated saline soil: The application of a marine bacterial strain isolated from mangrove sediment, Marine Pollution Bulletin, 192, 115071, 10.1016/j.marpolbul.2023.115071, 2023a.

Ren, L., Weng, L., Chen, D., Hu, H., Jia, Y., and Zhou, J. L.: Bioremediation of PAEs-contaminated saline soil: The application of a marine bacterial strain isolated from mangrove sediment, Mar. Pollut. Bull., 192, 115071, 10.1016/j.marpolbul.2023.115071, 2023b.

Rukachaisirikul, V., Khamthong, N., Sukpondma, Y., Phongpaichit, S., Hutadilok-Towatana, N., Graidist, P., Sakayaroj, J., and Kirtikara, K.: Cyclohexene, diketopiperazine, lactone and phenol derivatives from the sea fan-derived fungi Nigrospora sp. PSU-F11 and PSU-F12, Arch. Pharm. Res., 33, 375–380, 10.1007/s12272-010-0305-3, 2010.

Salter, M. E., Nilsson, E. D., Butcher, A., and Bilde, M.: On the seawater temperature dependence of the sea spray aerosol generated by a continuous plunging jet, J. Geophys. Res. Atmos., 119, 9052-9072, 10.1002/2013jd021376, 2014.

Sanjuan, O. N., Sait, S. T. L., Gonzalez, S. V., Tomás, J., Raga, J. A., and Asimakopoulos, A. G.: Phthalate metabolites in loggerhead marine turtles (Caretta caretta) from the Mediterranean Sea (East Spain region), Environ. Toxicol. Chem., 5, 178–185, 10.1016/j.enceco.2023.08.003, 2023.

Sha, B., Johansson, J. H., Benskin, J. P., Cousins, I. T., and Salter, M. E.: Influence of water concentrations of perfluoroalkyl acids (PFAAs) on their size-resolved enrichment in nascent sea spray

aerosols, Environ. Sci. Technol., 55, 9489–9497, 10.1021/acs.est.0c03804, 2021.

Shao, C., Guo, Z., Peng, H., Peng, G., Huang, Z., She, Z., Lin, Y., and Zhou, S.: A new isoprenyl phenyl ether compound from mangrove fungus, Chem Nat Compd, 43, 377–380, 10.1007/s10600-007-0142-x, 2007.

Shumilina, E., Skavang, P. K., and Dikiy, A.: Application of NMR spectroscopy for the detection and quantification of phthalic acid in fish muscles: The case of Atlantic Cod from Norwegian Sea, Mar. Environ. Res., 188, 105973, 10.1016/j.marenvres.2023.105973, 2023.

Simoneit, B. R. T.: Biomass burning- a review of organic tracers for smoke from incomplete combustion, Appl. Geochem., 17, 129–162, 10.1016/S0883-2927(01)00061-0, 2022.

Song, Y., Li, J., Tsona, N. T., Liu, L., and Du, L.: Enrichment of short-chain organic acids transferred to submicron sea spray aerosols, Sci. Total Environ., 851, 158122, 10.1016/j.scitotenv.2022.158122, 2022.

Tang, I. N., Tridico, A. C., and Fung, K. H.: Thermodynamic and optical properties of sea salt aerosols, J. Geophys. Res. Atmos., 102, 23269-23275, 10.1029/97jd01806, 1997.

Textor, C., Schulz, M., Guibert, S., Kinne, S., Balkanski, Y., and Bauer, S.: Analysis and quantification of the diversities of aerosol life cycles within AeroCom, Atmos. Chem. Phys., 6, 1777–1813, 10.5194/acp-6-1777-2006, 2006.

Tian, M., Zhu, T., Park, H. E., and Row, K. H.: Purification of 4-hydroxybenzoic acid and 4-hydroxybenzaldehyde from Laminaria japonica aresch using commercial and monolithic sorbent in SPE cartridge, Anal. Lett., 45, 2359–2366, 10.1080/00032719.2012.691590, 2012.

Van Acker, E., Huysman, S., De Rijcke, M., Asselman, J., De Schamphelaere, K. A. C., Vanhaecke, L., and Janssen, C. R.: Phycotoxin-enriched sea spray aerosols: Methods, mechanisms, and human exposure, Environ. Sci. Technol., 55, 6184–6196, 10.1021/acs.est.1c00995, 2021.

Wang, J., Yao, P., Bianchi, T. S., Li, D., Zhao, B., Cui, X., Pan, H., Zhang, T., and Yu, Z.: The effect of particle density on the sources, distribution, and degradation of sedimentary organic carbon in the Changjiang Estuary and adjacent shelf, Chem. Geol., 402, 52–67, 10.1016/j.chemgeo.2015.02.040, 2015.

Xu, F., Jin, H., Ji, Z., Chen, J., and Loh, P. S.: Sources and distribution of sedimentary organic matter along the northern Bering and Chukchi Seas, J Environ Sci (China), 52, 66–75, 10.1016/j.jes.2016.04.003, 2017.

Xu, W., Ovadnevaite, J., Fossum, K. N., Lin, C. S., Huang, R. J., Ceburnis, D., and O'Dowd, C.: Sea spray as an obscured source for marine cloud nuclei, Nat. Geosci., 15, 282–286, 10.1038/s41561-022-00917-2, 2022.

Xuefeng Zhou, X. L., Xieyang Guo, Bin Yang, Xian-Wen Yang and Yonghong Liu: Chemical constituents of the sponge mycale species from South China Sea, Rec. Nat. Prod., 7, 119–123, 2013.

Yang, J., Zhao, W., Wei, L., Zhang, Q., Zhao, Y., Hu, W., Wu, L., Li, X., Pavuluri, C. M., Pan, X., Sun, Y., Wang, Z., Liu, C.-Q., Kawamura, K., and Fu, P.: Molecular and spatial distributions of dicarboxylic acids, oxocarboxylic acids, and α-dicarbonyls in marine aerosols from the South China Sea to the eastern Indian Ocean, Atmos. Chem. Phys., 20, 6841–6860, 10.5194/acp-20-6841-2020, 2020.

Zangrando, R., Corami, F., Barbaro, E., Grosso, A., Barbante, C., Turetta, C., Capodaglio, G., and Gambaro, A.: Free phenolic compounds in waters of the Ross Sea, Sci. Total Environ., 650, 2117–2128, 10.1016/j.scitotenv.2018.09.360, 2019.

Zhao, X., Qiu, W., Zheng, Y., Xiong, J., Gao, C., and Hu, S.: Occurrence, distribution, bioaccumulation, and ecological risk of bisphenol analogues, parabens and their metabolites in the Pearl River Estuary, South China, Ecotoxicol. Environ. Saf., 180, 43–52, 10.1016/j.ecoenv.2019.04.083, 2019.

Zhu, Y., Tilgner, A., Hans Hoffmann, E., Herrmann, H., Kawamura, K., Xue, L., Yang, L., and Wang, W.:

Molecular distributions of dicarboxylic acids, oxocarboxylic acids, and α-dicarbonyls in aerosols over Tuoji Island in the Bohai Sea: Effects of East Asian continental outflow, Atmos Res, 272, 106154, 10.1016/j.atmosres.2022.106154, 2022.

---

## Author Comment (AC3)

Our responses to the comments are as follows: Comments are in black, authors' responses are in blue, and changes to the manuscript are in red color text. Figures prepared for reply are named as Figure R-.

**Community comment:**

Dear Authors, editor, and reviewers,

This manuscript presents insights into the interactions between sea spray aerosols (SSAs) and aromatic acids through a laboratory study. This is an interesting and valuable research. I have some constructive comments on this manuscript.

Specific comments:

Abstract and Introduction:

Line 11, what is the importance of aromatic acids with SSAs in affecting the global radiative balance? In the following section of Introduction (Lines 29-40), the authors had ambiguous presentation in relationship between SSAs, aromatic acids, and global radiative balance.

**Author reply:**

As an important medium for exchange processes between ocean and the atmosphere, SSA directly affects the radiative balance of the Earth by scattering solar radiation and indirectly affects the global climate as a source of cloud condensation nuclei and ice nuclei (Wilson et al., 2015; Quinn et al., 2017; Rosenfeld et al., 2019; Croft et al., 2021). It is well known that the presence of surfactants affects the production of SSA, which in turn may affect the global radiation balance. This indicates that aromatic acids as a surfactant may also affect the global radiation balance. We have updated the Introduction section.

Lines 35-38:

Previous studies suggest that these organic acids can alter the composition of SSA, subsequently influencing atmospheric processes such as cloud condensation nuclei (CCN) or ice nuclei (IN) activities. SSA may contain organic acids that play a major role in Earth's climate (Moore et al., 2011; Zhu et al., 2019).

Lines 41-54: The authors introduced the current studies in seawater source of aromatic acids in this paragraph, that is, the majority is from esters released and biodegraded by algae and the remain is from anthropogenic emissions. Could aromatic acids be derived from precipitation? What are the major types and their proportions and concentrations of aromatic acids in seawater? It should correspond to aromatic acids the authors used in the experiments (lines 90-98, 2 Experiment Section). More detailed description and possible summary are needed.

**Author reply:**

A previous study found benzoic acid in snow pit samples (Mochizuki et al., 2016), while the other aromatic acid samples were not studied in precipitation. The types, sources, and concentrations of the target aromatic acids are summarized in Table S4. The following content has been added to the revised manuscript.

Line 41-45:

In various observations, aromatic acids have been detected in both natural and anthropogenic sources (Zhao et al., 2019; Zangrando et al., 2019; Dekiff et al., 2014). Among them, natural sources of aromatic acids produced by algal releases account for most marine aromatic acids, especially benzoic acid, and most of hydroxybenzoic acids (Mostafa et al., 2017; Fotso Fondja Yao et al., 2010b; Castillo et al., 2023; Abdel-Hamid A. Hamdy, 2020).

**Line 50-53:**

This is consistent with previous researches that phthalic acid is primarily derived from anthropogenic sources (Ren et al., 2023b), whereas hydroxybenzoic acid has both anthropogenic and natural sources (Zhao et al., 2019; Castillo et al., 2023). Line 89-90:

Further details are provided in Table S1, which lists sources and concentrations of these aromatic acid identified in seawater and atmospheric samples over the ocean.

| Aromatic acids  | Natural sources                                                        | Anthropogenic sources                                                |
|-----------------|------------------------------------------------------------------------|----------------------------------------------------------------------|
| benzoic acid    | • sea algae (Abdel-Hamid A. Hamdy, 2020; Al-Zereini et al.,            | • emerging endocrine disrupting compounds (34–491                    |
|                 | 2010; Fotso Fondja Yao et al., 2010a; Liu et al., 2022b)               | ng L -1 ) (Zhao et al., 2019)                             |
|                 | • sedimentary organic matter (10–65 µg g -1 ) (Deshmukh et  | • fuel combustion (Boreddy et al., 2017)                             |
|                 | al., 2016)                                                             | • industrial wastewater, automobile exhaust and                      |
|                 | • bacteria isolated from sea bass viscera (314 ppb) (Martí-            | tobacco smoke (Cuadros-Orellana et al., 2006)                        |
|                 | Quijal et al., 2020)                                                   |                                                                      |
|                 | • snow pit samples (2.11 ng g -1 ) (Mochizuki et al., 2016) |                                                                      |
| o-phthalic acid |                                                                        | • plasticizer (16.7–657 ng g -1 d.w.) (Ren et al., 2023a; |
|                 |                                                                        | Sanjuan et al., 2023);                                               |
|                 |                                                                        | • plastic waste burning (8.3–84.9 ng m -3 ) (Zhu et al.,  |
|                 |                                                                        | 2022)                                                                |
|                 |                                                                        | • the end product of photochemical oxidation of SOA                  |
|                 |                                                                        | (15.5 ng m -3 ) (Ding et al., 2021)                       |
|                 |                                                                        | • biomass burning and fossil fuel combustion sources                 |
|                 |                                                                        | (0.4–7.9 ng m -3 ) (Shumilina et al., 2023; Yang et al.,  |
|                 |                                                                        | 2020; Boreddy et al., 2022)                                          |

Table S1. Sources and concentrations of aromatic acids identified in seawater and atmospheric samples over the ocean.

| m phthalia agid               |                                                                           |   | plasticizor (Pop et al. 2022a)                                     |
|-------------------------------|---------------------------------------------------------------------------|---|--------------------------------------------------------------------|
| m -plitilatic acid     |                                                                           |   | plasticizer (Reli et al., 2023a)                                   |
|                               |                                                                           | • | the end product of photochemical oxidation of SOA                  |
|                               |                                                                           |   | (3.6 ng m -3 ) (Ding et al., 2021)                      |
|                               |                                                                           | • | biomass burning and fossil fuel combustion sources                 |
|                               |                                                                           |   | (0.01–2.3 ng m -3 ) (Yang et al., 2020; Boreddy et al., |
|                               |                                                                           |   | 2022; Kawamur, 2014)                                               |
| p -phthalic acid       |                                                                           | • | plasticizer (0.51–6.8 mg kg -1 ) (Ren et al., 2023a; Di |
|                               |                                                                           |   | Giacinto et al., 2023; Di Renzo et al., 2021)                      |
|                               |                                                                           | • | plastic waste burning (10.8–80.7 ng m -3 ) (Zhu et al., |
|                               |                                                                           |   | 2022); the end product of photochemical oxidation                  |
|                               |                                                                           |   | of SOA (4.3 ng m -3 ) (Ding et al., 2021)               |
|                               |                                                                           | • | biomass burning and fossil fuel combustion sources                 |
|                               |                                                                           |   | (0.05–2.5 ng m -3 ) (Yang et al., 2020; Boreddy et al., |
|                               |                                                                           |   | 2022; Kawamur, 2014)                                               |
| o-hydroxybenzoic acid         | • sea algae (76.8 mg L -1 ) (Castillo et al., 2023; Mostafa et | • | pharmaceuticals and drugs of abuse (0.4–53.3 ng $L^-$              |
|                               | al., 2017; Klejdus et al., 2017)                                          |   | 1 ) (Alygizakis et al., 2016)                           |
| m -hydroxybenzoic acid | • sea algae (Al-Zereini et al., 2010; Castillo et al., 2023)              |   |                                                                    |

| p -hydroxybenzoic acid | • sea algae (57.7 mg L -1 ) (Castillo et al., 2023; Klejdus et al., | • | Pharmaceuticals and personal care products (Lu et              |
|-------------------------------|--------------------------------------------------------------------------------|---|----------------------------------------------------------------|
|                               | 2017; Tian et al., 2012; Hawas and Abou El-Kassem, 2017)                       |   | al., 2023)                                                     |
|                               | • sea fungus (Rukachaisirikul et al., 2010; Shao et al., 2007)                 | • | emerging endocrine disrupting compounds (4.58-                 |
|                               | • sponge Mycale species (Xuefeng Zhou, 2013); metabolite                       |   | 49.9 ng L -1 ) (Zhao et al., 2019; Lu et al., 2021; |
|                               | (Jingchuan Xue, 2015; Liao and Kannan, 2018)                                   |   | Alygizakis et al., 2016)                                       |
|                               | • sediment samples (6.85–437 ng g -1 dw) (Liao et al., 2019)        |   |                                                                |
| vanillic acid                 | • sea algae (3–47 ng L -1 ) (Zangrando et al., 2019; Klejdus et     | • | combustion of both softwood and hardwood                       |
|                               | al., 2017)                                                                     |   | (Simoneit, 2022)                                               |
|                               | • lignin decomposition (Wang et al., 2015; Hu et al., 2022;                    |   |                                                                |
|                               | Xu et al., 2017)                                                               |   |                                                                |
| syringic acid                 | • sea algae $(0.3-0.6 \text{ ng } \text{L}^{-1})$ (Poznyakovsky et al., 2021;  | • | pharmaceuticals (Fisch et al., 2017)                           |
|                               | Zangrando et al., 2019; Klejdus et al., 2017)                                  | • | hardwood burning (Simoneit, 2022)                              |
|                               | • lignin decomposition (Hu et al., 2022; Xu et al., 2017)                      |   |                                                                |

Line 80 Sintered glass filter and wave breaking method can also produce artificial SSAs with different properties of flux, chemical composition, size distribution, and so on. The authors should clarify the relationship between methods (sintered glass filter, plunging jet, and wave breaking) and possible efficiency of aromatic acid transport within SSAs. That is, whether the plunging jet method can effectively reflect the "Role of sea spray aerosol at the air-sea interface in transporting aromatic acids to the atmosphere"?

**Author reply:**

Combining the following parameters, we obtained a particle size distribution similar to that from field observations (Quinn et al., 2017; Xu et al., 2022), which is shown in Fig. S2. The experimental principle aligns with previous successful studies (Sha et al., 2021; Johansson et al., 2019). This enables us to consider that the plunging jet method can effectively reflect the transfer of aromatic acid at the air-sea interface.

The reasons for the selection of the relevant parameters are as follows:

1) A stainless steel nozzle with an inner diameter of 4.3 mm is used to generate plunging jets. The nozzle can be changed easily. The size of the nozzle is consistent with that in previous studies (Salter et al., 2014; Sha et al., 2021).

2) We have studied the cases with headspace heights of 14 cm, 19 cm, and 22 cm in our previous work (Liu et al., 2022a) and found that the SSA production was most efficient at a headspace height of 22 cm. Hence, we chose it for SSA generation.

3) We have tried pump flow rates of 0.50 L min-1, 0.75 L min-1, 1.00 L min-1, 1.25 L min-1, and 1.50 L min-1 in our previous study (Liu et al., 2022a) and found that when the pump flow rate was 1 L min-1 or higher, the performance of the SSA generator was good (see Table R1). Therefore, we used a pump flow rate of 1 L min-1 to maintain a high SSA production rate and a relatively long working lifetime of the corresponding accessories.

| Pump flow rate (L min -1 ) | SSA production (particles s -1 ) | Mass concentration (µg m -3 ) |
|---------------------------------------|---------------------------------------------|------------------------------------------|
| 0.50                                  | $5.78	imes10^{6}$                           | 198.96                                   |
| 0.75                                  | $9.90	imes10^6$                             | 317.31                                   |
| 1.00                                  | $1.37 	imes 10^7$                           | 417.86                                   |
| 1.25                                  | $1.58 	imes 10^7$                           | 469.59                                   |
| 1.50                                  | $1.63 	imes 10^7$                           | 495.42                                   |

Table R1. Effect of pump flow rate on SSA production.

4) We have previously studied SSA production at purge air flow rates of 2 L min-1, 3 L min-1, 4 L min-1, 5 L min-1, 6 L min-1 to simulate the sea breeze (Liu et al., 2022a). We found that the SSA production increased with the purge air flow rate when the flow rate was less than 3 L min-1, and decreased with the increase of the purge air when the flow rate was more than 3 L min-1.

5) Based on a previous study of the effect of relative humidity on the growth of sea salt particles, the inlet humidity would affect the morphology of SSA. (Tang et al., 1997). Hence, we chose to keep the relative humidity at about 40% in our study. Particles will deliquescence at higher humidity (Bryan et al., 2022). In addition, it has been demonstrated that acid has no effect on NaCl deliquescence (Ming and Russell, 2001).

**Fig. S2.** Number size distribution of SSA generated with the SSA simulation chamber in this study compared with field studies.

**Experimental Section**

Lines 90-97 Why do authors chose these types of aromatic acids? It should be clarified.

**Author reply:**

1) Molecular Structure:

Different aromatic acids have distinct properties, including molecular structure, polarity,

and reactivity. We aimed to investigate the position and number of functional groups and the effect of different functional groups on the transport of aromatic acids in SSA at the air-sea interface.

2) Biological Significance:

Most aromatic acids we selected are emerging endocrine-disrupting compounds with the potential for bioaccumulation in living organisms (Zhao et al., 2019). The associated human health risks should be of great concern.

3) Climate-Relevant Properties:

Aromatic acids may change the SSA acidity that make them particularly relevant to climate processes (Angle et al., 2021).

In summary, the selection of specific aromatic acids is likely driven by a combination of their molecular structure, biological significance, relevance to climate processes, and practical considerations for experimentation or observation.

Lines 93-94 The collection procedure of seawater is not clear (site, chemical composition of seawater, contamination control, storge conditions, filtration or not?) Detailed QA/QC description is needed.

**Author reply:**

We have indicated the collection procedure of seawater in the revised Supplement to make the experimental process clearer.

S1. Quality assurance/quality control.

Seawater was collected from the coastal area of Shazikou on March 27, 2023, with a volume of 500 L (Fig. S1). Considering the storage inconvenience caused by huge consumption of seawater, all our seawater was pre-filtered through a polyethersulfone filter (47 mm diameter, 0.2  $\mu$ m pore size, Supor®-200, Pall Life Sciences, USA) and stored in the dark at 18 °C for less than one month. Quinn et al. (2015) have shown that the fraction that passes through the filter is regarded as dissolved organic carbon and includes colloidal and truly dissolved materials. For each experiment, we measured particle number concentrations generated by filtered seawater and cations concentrations in seawater, and we found good agreement between each set of experiments (see Fig. S2).

In order to avoid the influence of organic matter in quartz fiber filters and access the accuracy of the experiment, pre-baked quartz fiber filters were used in sampling. Before each set of experiments, experimental blanks were conducted using filtered seawater. Experimental blanks were conducted with the same procedure of SSA samples. Seawater and filter samples were stored at  $-20^{\circ}$ C until analyzed. In order to reduce the influence of organic acids residue after each experiment, the SSA simulation chamber was cleaned with ethanol first, then the system was cleaned with ultra-pure water for several times. The above steps also run the pump to allow for thorough cleaning of the system. Therefore, the system was blown with zero air and sealed for preservation. The Dekati DLPI was also ultrasonicated with methanol and water (V:V=1:1) and dried after the experiment.

Filtered seawater (without added aromatic acid) was used as the experimental blank, and the same experimental and analytical methods were used as those for the experimental samples. As a result, no target aromatic acid was found in both seawater and filters. This may be due to the fact that we did not perform any concentration operation during the seawater sample processing. The standard curves for each aromatic acid are linear, as shown in Fig. S5.

---

## Author Response (AR3)

We thank the Editor and Reviewers for their insightful comments. We have revised our manuscript according to the suggestions of the Reviewers' comments and our responses to the comments are as follows: Reviewers' comments are in black, authors' responses are in blue, and changes to the manuscript are in red color text.

**Editor decision:**

Comments (line number refer to the latest version of the manuscript w/o track changes):

- Line 36: SSA should be explained here at its first occurrence and not in line 63.

**Author reply:**

We have made the necessary revisions to ensure that SSA is properly explained at its first occurrence.

- Line 36-40: There is some repetition here with respect to CCN.

**Author reply:**

We have carefully reviewed the manuscript to eliminate any unnecessary repetition.

- Line 99: Figure or Table S1?

**Author reply:**

We have revised the manuscript accordingly, and the correct response is "Fig. S1."

- Line 124: Are the RH and T-ranges measured or technically given/fixed?

**Author reply:**

The RH and T ranges are technically given parameters in the experimental setup, and actual measurements fall within these specified ranges during the experiments.

- Line 130: I would give the mean and standard deviation of the measured RH (since you measured it).

**Author reply:**

The measured RH during the experiments was 34.2±3.9%. This has been updated in the revised manuscript.

- Line 133: particles -> particle

**Author reply:**

Revised.

- Line 136: Better "<20 cm$^{-3}$" (remove the hashtag)

**Author reply:**

The hashtag has been removed and replaced with "< 20 cm$^{-3}$".

- Line 155: Add "a" before "muffle"

**Author reply:**

We have added "a" before "muffle" accordingly.

- Beginning of Sect. 3.1: This should be part of the methods and not the result section.

**Author reply:**

We have moved the content to Sect. 2.3 as suggested.

- Table 1 caption: add "the" before "experiments"

**Author reply:**

We have revised the caption of Table 1 accordingly by adding "the" before "experiments".

- Figure 1: Please describe properly the box plot. What are the whiskers showing? How many data points are included in each box? 3 points is not enough (as suggested in the caption).

**Author reply:**

We have provided a more detailed description of the box plot in Fig. 1, specifying that each box contains 14 data points. The whiskers represent the maximum and minimum values, respectively.

Lines 685–688:

Fig. 1. Measured surface tension values of natural seawater and aromatic acid-containing seawater: benzoic acids (A), benzenedicarboxylic acids (B), hydroxybenzoic acids (C), *p*-hydroxybenzoic acid, vanillic acid, and syringic acid (D). The dark spots represent the mean values of at least 9 data points, the boxes represent the ranges of 25th−50th−75th percentiles, and the whiskers represent the maximum and minimum values.

- Figure 2: "SW" is not properly defined. Is it the ambient sea water or the artificial sea water (which in the text is abbreviated as ASW). Please make sure that all abbreviations are properly defined.

**Author reply:**

"SW" refers to the ambient seawater in our study. We apologize for any confusion caused by the abbreviation. We have defined all abbreviations properly throughout the manuscript.

Lines 690–692:

Fig. 2. Number concentration distribution of sea salt particles and SSA particles containing benzoic acids (A), benzenedicarboxylic acids (B), hydroxybenzoic acids (C), *p*-hydroxybenzoic acid, vanillic acid, and syringic acid (D). SW represents natural seawater.

- Figure 3: Define "SW". It is really hard to see difference in the circle size. In the caption, make clear that "numbers above the points give the geometric mean diameter (in nm)". Errorbars are standard deviation?

**Author reply:**

In Figure 3, "SW" refers to seawater. We have made sure to define this abbreviation in the figure caption. Regarding the circle size, we have improved to enhance the visibility of the differences. Additionally, we have clarified in the caption that the numbers below or above the points represent the geometric mean diameter in nanometers (nm). The error bars in the figure represents the standard deviation. We have made these clarifications in the revised manuscript.

Lines 693–697:

[Figure]

Fig. 3. SSA production, particle size, mass concentration distribution of aromatic acids. The symbol size represents the geometric mean diameter of SSA particles, with the numbers below

or above the points giving the geometric mean diameter (in nm), and the error bars are standard deviation. The symbol color indicates the particle mass concentration, with SW representing natural seawater.

- Figure 7: Harmonize if you use "tons" or "tonnes" within the figures and text.

**Author reply:**

We have made the necessary adjustments and made usage of "tons" both within the figures and the text.

Lines 708–711:

[Figure]

Fig. 7. Concentration range of aromatic acids in seawater (A) and the estimated range of annual global aromatic acids emission (tons yr$^{-1}$) via SSA (B). Yellow and blue stacked columns represent emissions based on Textor et al. (2006) and Gliss et al. (2021), respectively.

- Supplement:

o All figures and tables in the SI need to be also referenced in the main text. I could not find S5, S9 and S10 in the main text! Otherwise remove them.

**Author reply:**

We have included references to all SI figures and tables in the main text accordingly:

Line 96: Table S1, Line 112: Table S2, Lines 130–131: Table S3, Line 326: Table S4, Line 364: Table S5, Line 380: Table S6.

Lines 100–101: Fig. S1, Line 109: Fig. S2, Line 117: Fig. S3, Line 168: Fig.S4, Lines 194 and 218: Fig. S5, Line 223: Fig. S6, Line 256: Fig. S7, Lines 282, 299, and 310: Fig. S8, Lines 321 and 345: Fig. S9, Line 361: Fig. S10.

o Figure S5: Please increase font size and check the y-labels.

**Author reply:**

We have increased the font size and reviewed the y-labels.

[Figure]

**Fig. S4.** Standard curves for aromatic acids were constructed within a concentration range of 0.01–1000 µM, with more than seven data points.

o Suggest to use normal page numbers and not S1, S2, etc. to not confuse with the figure and table labelling.

**Author reply:**

We have revised the page numbering to use normal page numbers.

- Data availability: I strongly recommend to make the data publicly available and follow the data policy of ACP (https://www.atmospheric-chemistry-and-physics.net/policies/data_policy.html).

**Author reply:**

We have arranged the data to be made publicly available and to comply with the data policy of ACP.

Lines 408–409:

The data used in this study can be found online at https://doi.org/10.5281/zenodo.10903140 (Song et al., 2024).

**Reviewer #2:**

I want to thank the authors for their persistence with this manuscript. I think their efforts have borne fruit. I believe the manuscript is now close to achieving the quality required for publication in ACP. However, there are still a few presentation issues that the authors would be advised to resolve. For example, there are several instances where the authors should have begun a new paragraph. An example of this can be found in line 55 where the authors should start a new paragraph with "Recent laboratory studies have shown...".

**Author reply:**

We have addressed the presentation issues raised by the Reviewer, including starting new paragraphs at line 55 with "Recent laboratory studies have shown...", at line 69 with "Recent data indicate that the surface activity...", and at line 212 with "Unlike the order of seawater surface tension…".

Also, there are several instances of incorrect referencing. For example, in Line 376 it should be "Gliss et al. (2021)" rather than "Jonas et al. (2021)". I would urge the authors to double-check all references. The authors would be wise to read thoroughly through the entire manuscript and double-check the presentation.

**Author reply:**

We have carefully reviewed the entire manuscript to ensure correct referencing. We have also read through the entire manuscript and double-check the presentation.

**Reviewer #3:**

The revised version has improved a lot and is in my opinion suitable for publication after the following minor issues are addressed:

Line 57 et al.: The concept about the "missing aromatic acids" is not clear to me and is not evident from the part "recent laboratory studies have shown that personal-care products, especially sunscreen (e.g., o-hydroxybenzoic acid), are reduced in levels during algal blooms (Franklin et al., 2022). " Please clarify.

**Author reply:**

Our intention to use "missing aromatic acids" was to highlight that the levels of o-hydroxybenzoic acid, commonly found in sunscreen products, decrease during algal blooms, but the reasons for this reduction are not yet clear. We speculate that these missing aromatic acids may be transported to the atmosphere through SSA.

Lines 56–57:

Recent laboratory studies have shown that personal-care products in SW, especially sunscreen (e.g., $o$-hydroxybenzoic acid), are reduced in levels during algal blooms (Franklin et al., 2022).

The language has strongly improved, however, there are still articles missing (in the newly introduced parts), such as

Line 69: "The" is missing: Moreover, "the" molecular structure

Line 79: to "the" molecular structure

Line 287: of "the" salt particles

Line 342: by "the" compound concent

Please check again carefully.

**Author reply:**

Thank you for pointing out those missing articles. We have revised them in the current version of the manuscript.

**References**

Franklin, E. B., Amiri, S., Crocker, D., Morris, C., Mayer, K., Sauer, J. S., Weber, R. J., Lee, C., Malfatti, F., Cappa, C. D., Bertram, T. H., Prather, K. A., and Goldstein, A. H.: Anthropogenic and biogenic contributions to the organic composition of coastal submicron sea spray aerosol, Environ. Sci. Technol., 56, 16633–16642, 10.1021/acs.est.2c04848, 2022.

Gliss, J., Mortier, A., Schulz, M., Andrews, E., Balkanski, Y., Bauer, S. E., Benedictow, A. M. K., Bian, H., Checa-Garcia, R., Chin, M., Ginoux, P., Griesfeller, J. J., Heckel, A., Kipling, Z., Kirkevåg, A., Kokkola, H., Laj, P., Sager, P. L., Lund, M. T., Myhre, C. L., Matsui, H., Myhre, G., Neubauer, D., Noije, T. v., North, P., Olivié, D. J. L., Rémy, S., Sogacheva, L., Takemura, T., Tsigaridis, K., and Tsyro, S. G.: AeroCom phase III multi-model evaluation of the aerosol life cycle and optical properties using ground- and space-based remote sensing as well as surface in situ observations, Atmos. Chem. Phys., 21, 87–128, 10.5194/acp-21-87-2021, 2021.

Textor, C., Schulz, M., Guibert, S., Kinne, S., Balkanski, Y., Bauer, S., Berntsen, T., Berglen, T., Boucher, O., Chin, M., Dentener, F., Diehl, T., Easter, R., Feichter, H., Fillmore, D., Ghan, S., Ginoux, P., Gong, S., Grini, A., Hendricks, J., Horowitz, L., Huang, P., Isaksen, I., Iversen, T., Kloster, S., Koch, D., Kirkevag, A., Kristjansson, J. E., Krol, M., Lauer, A., Lamarque, J. F., Liu, X., Montanaro, V., Myhre, G., Penner, J., Pitari, G., Reddy, S., Seland, Stier, P., Takemura, T., and Tie, X.: Analysis and quantification of the diversities of aerosol life cycles within AeroCom, Atmos. Chem. Phys., 6, 1777–1813, 10.5194/acp-6-1777-2006, 2006.

---

## Author Response (AR4)

We thank the Reviewers for their insightful comments. We have revised our manuscript according to the suggestions of the Reviewers' comments and our responses to the comments are as follows: Reviewers' comments are in black, authors' responses are in blue, and changes to the manuscript are in red color text. Figures prepared for reply are named as Figure R-.

**Reviewer #1:**

**General comments**

I find this manuscript well-written / well-referenced, scientifically interesting for the SSA community, and the experimental quality is good. I find it great that the authors try connect SSA experiments with functional group level chemistry.

I have some minor concerns, that should be easily to address. I would recommend publication with only minor revisions.

**Author reply:**

We thank the Reviewer for the positive assessment of our manuscript and the constructive comments. We have revised our manuscript according to the suggestions of the Reviewer's comments and our responses to the comments are as follows.

**Specific comments**

1. Real seawater composition. In line 94 you write that you sample and transport seawater to the SSA laboratory. Is it possible to get more details? Conditions at sampling site (is it a productive area?), temperature, duration of storage, volume, was it filtered? Was it sampled on the same or different days? What time of year? You can add this information to the SI or just expand Table S1.

   **Author reply:**

   Based on the global net primary productivity estimated by Dai et al. (2023), the sampling site is a high-productivity area. We have indicated the storage conditions

and time in the revised SI and expanded Table S3 to make the experimental process clearer.

**S1.** Quality assurance/quality control

Seawater was collected from the coastal area of Shazikou on March 27, 2023, with a volume of 500 L (Fig. S1). Considering the storage inconvenience caused by huge consumption of seawater, all our seawater was pre-filtered through a polyethersulfone filter (47 mm diameter, 0.2 μm pore size, Supor®-200, Pall Life Sciences, USA) and stored in the dark at 18 °C for less than one month. Quinn et al. (2015) have shown that the fraction that passes through the filter is regarded as dissolved organic carbon and includes colloidal and truly dissolved materials.

[Figure]

**Fig. S1.** Sampling site at Shazikou along the Yellow Sea coast, Qingdao, China.

**Table S3.** Summary of experimental conditions.

| Exp. No. | Experiment type | Concentration (mM) | pH | Salinity (psu) | Sampling time (h) | RH (%) | Temperature difference (°C) [a] |
|---|---|---|---|---|---|---|---|
| 1 | SW | 0 | 7.92 | 34.2 | 5 | 35 | 2.0 |
| 2 | SW+benzoic acid | 1 | 7.72 | 34.3 | 5 | 34 | 1.5 |
| 3 | SW+$o$-hydroxybenzoic acid | 1 | 7.60 | 34.5 | 5 | 36 | 1.0 |
| 4 | SW+$m$-hydroxybenzoic acid | 1 | 7.68 | 34.1 | 5 | 40 | 2.0 |
| 5 | SW+$p$-hydroxybenzoic acid | 1 | 7.84 | 34.3 | 5 | 38 | 1.5 |
| 6 | SW+$o$-phthalic acid | 1 | 7.58 | 34.2 | 5 | 36 | 2.0 |
| 7 | SW+$m$-phthalic acid | 1 | 7.80 | 34.5 | 5 | 37 | 2.5 |
| 8 | SW+$p$-phthalic acid | 1 | 7.85 | 34.4 | 5 | 42 | 2.0 |
| 9 | SW+vanillic acid | 1 | 7.81 | 34.2 | 5 | 43 | 3.0 |
| 10 | SW+syringic acid | 1 | 7.84 | 34.3 | 5 | 39 | 2.0 |
| 11 | ASW | 0 | 7.96 | 35.1 | 5 | 33 | 1.5 |
| 12 | ASW+benzoic acid | 1 | 7.68 | 34.6 | 5 | 35 | 1.0 |
| 13 | ASW+$o$-hydroxybenzoic acid | 1 | 7.76 | 34.9 | 5 | 34 | 0.5 |
| 14 | ASW+$m$-hydroxybenzoic acid | 1 | 7.99 | 35.3 | 5 | 36 | 1.5 |
| 15 | ASW+$p$-hydroxybenzoic acid | 1 | 7.85 | 34.7 | 5 | 38 | 2.0 |
| 16 | ASW+$o$-phthalic acid | 1 | 7.93 | 34.5 | 5 | 35 | 1.0 |

| 17 | ASW+$m$-phthalic acid | 1 | 7.88 | 34.9 | 5 | 36 | 1.0 |
| 18 | ASW+$p$-phthalic acid | 1 | 7.97 | 34.6 | 5 | 34 | 1.5 |
| 19 | ASW+vanillic acid | 1 | 7.89 | 35.2 | 5 | 35 | 1.0 |
| 20 | ASW+syringic acid | 1 | 7.99 | 34.8 | 5 | 39 | 1.0 |
| 21 | ASW+benzoic acid+$o$-hydroxybenzoic acid+$o$-phthalic acid+vanillic acid+syringic acid | $10^{-3}$ | 7.95 | 35.1 | 20 | 41 | 3.5 |
| 22 | ASW+benzoic acid+$m$-hydroxybenzoic acid+$m$-phthalic acid+vanillic acid+syringic acid | $10^{-3}$ | 7.98 | 34.6 | 20 | 38 | 1.5 |
| 23 | ASW+benzoic acid+$p$-hydroxybenzoic acid+$p$-phthalic acid+vanillic acid+syringic acid | $10^{-3}$ | 7.88 | 34.9 | 20 | 40 | 2.0 |
| 24 | NaCl | 0 | 7.68 | 35.3 | 5 | 38 | 1.0 |
| 25 | NaCl+$m$-hydroxybenzoic acid | 1 | 7.54 | 34.7 | 5 | 36 | 1.5 |

[a] The temperature difference in the SSA simulation chamber before and after the experiment.

o   Also I have had challenges when I sampled fresh real seawater, that the SSA properties (size and number) changed as a function of time in the SSA chamber (due to microbial activity, degassing) – I therefore sometimes prepared artificial seawater from just inorganic sea salts. Could the authors elaborate on how reproducible the experiments are? And would the authors expect the results being similar using artificial inorganic mixture?

**Author reply:**

1) All our seawater was pre-filtered through a polyethersulfone filter (47 mm diameter, 0.2 μm pore size, Supor®-200, Pall Life Sciences, USA) and stored in the dark at 18 °C for less than one month to minimize microbiological effects.

2) We measured the total particle number concentration and concentration of Na$^+$ of seawater before each experiment, and all the experiments showed good repeatability (Fig. R1). Furthermore, for comparing the properties of SSA particles containing aromatic acids, we normalized the particles size distribution of seawater before adding aromatic acids. Therefore, perhaps it is likely that the same trend would exist in artificial seawater.

[Figure]

Fig. R1. Mean value of total particle number concentration of SSA (N$_{Total}$) and concentration of Na$^+$ (C(Na$^+$)) for each experiment.

3) To demonstrate this conclusion more rigorously, we added aromatic acid to the artificial seawater and observed its effect on the particle size distribution of SSA particles. The following text has been added in the revised manuscript.

Lines 241-242:

Moreover, the results showed that the effect trends of aromatic acids on SSA production in ASW were consistent with those observed in seawater (Fig. S7), eliminating the influence of organic matter.

[Figure]

**Fig. S7.** Number concentration distribution of sea salt particles and SSA particles containing benzoic acids (A), benzene dicarboxylic acids (B), hydroxybenzoic acids (C), vanillic acid and syringic acid (D).

From the results, we can see that the effect trend of aromatic acids on SSA production is generally similar, not only in seawater but also in artificial seawater.

Furthermore, we also measured the artificial seawater surface tension with aromatic acids and the EF of aromatic acids, and the results are as follows.

[Figure]

**Fig. S6.** Measured surface tension values of artificial seawater (ASW) and aromatic acid-containing ASW.

From the results, we can see that the effect trend of aromatic acids on seawater surface tension is generally similar, not only in seawater but also in artificial seawater.

[Figure]

**Fig. S8.** Enrichment factors of aromatic acids at different concentrations from artificial seawater to the atmosphere.

Comparing the EF of aromatic acids in SW and ASW, it was observed that the EF trends of benzene dicarboxylic acids in seawater follows the pattern: *o*-phthalic acid < *m*-

phthalic acid $<$ *p*-phthalic acid. However, in ASW, the EF of *p*-phthalic acid was lower than that of *m*-phthalic acid. Based on the findings of Li et al. (2023), we hypothesize that in ASW, *p*-phthalic acid acts as an ·OH scavenger to produce TAOH. Hence, the EF of *p*-phthalic acid is lower than that of *m*-phthalic acid. Meanwhile, organic compounds in SW preferentially react with ·OH (Anastasio and Newberg, 2007), thus the EF of *p*-phthalic acid is the highest among the benzene dicarboxylic acids. Furthermore, differences in aromatic acid concentration did not change the enrichment pattern of organic acids.

2. Experimental Setup. Would be helpful for the reader if Figure S1 was updated to include schematics of the entire setup, e.g. add where the DLPI+ was connected (before or after dryer?), single particle sampler (TEM). The air flow rate into SSA chamber (Line 109). Why does the range span from 3 all the way to 50 L min$^{-1}$? Is it because you have different setups at different times during a single experiment? Could you elaborate more on this.

**Author reply:**

The Dekati DLPI+ and single particle sampler were connected after dryer and sampling was carried out separately. We updated figure S3 to include schematics of the entire setup in the supplement to make it clearer. For the air flow rate, we would like to express that the zero air can be adjusted in the range of 3–50 L min$^{-1}$, e.g. zero air flow rate into the SSA simulation chamber was set at 3 L min$^{-1}$, while the zero air flow rate of the Nafion dryer tube was set at three times the outlet air flow rate when we measured the size distribution of SSA particles. When the total particle concentration has stabilized, we connected the single particle sampler to sample the SSA particles under the same air flow rate. The inlet flow rate for Dekai DLPI+ sampler sampling was set to 10 L min$^{-1}$ to supply its pumping flow. We also updated Figure S3 to include air flow rate in the supplement to make it clearer.

[Figure]

Fig. S3. Schematic picture of the plunging jet-sea spray aerosol generator: SMPS sampling (A), single particle sampling (B), and DeKati DLPI+ sampling (C). The red arrows represent the flow direction of seawater, and the purple arrows represent the flow of gases and aerosol particles.

3. TEM details. I am missing details about TEM instrument and conditions. When you do TEM, wouldn't the organic coating of SSA just vaporize in the vacuum? Also, wouldn't the SSA and aromatic acids be more internally mixed, when use a plunging jet / real bubble bursting?

**Author reply:**

We have added details about TEM instrument and conditions in the manuscript. The copper grid film was placed in a liquid nitrogen-frozen vacuum environment through a TEM holder, thereby inhibiting the evaporation of the organic coating of SSA. According to the classification of aerosol particles by Li et al. (2016), a particle consisting of two or more aerosol components can be defined as an internally mixed particle. Otherwise, it will be regarded as an externally mixed particle. For the mixing state of SSA particles, the OM coating and core-shell mixing structure were considered as internally mixed in previous studies (Li et al., 2016; Li et al., 2021).

Lines 138-141:

TEM was performed using a TEM cryo-mount (Gatan 626) to load the samples, where the TEM grid was immersed in liquid nitrogen and then mounted on the holder by means of a cryo-transfer workstation. TEM with a high-angle annulardark-field detector was used and then TEM images were obtained at an accelerating voltage of 200 kV.

**Technical comments**

Line 58: add mass, so it reads main mass component

**Author reply:**

We simplified the original sentence as follows.

Lines 59-61:

Sea spray aerosols (SSAs), generated by breaking waves and bubble bursting, are one of the major sources of atmospheric particles (Andreae and Rosenfeld, 2008; Angle et al., 2021; Hasenecz et al., 2020; Malfatti et al., 2019).

Line 61: "... disturbing ecological systems... ", sound funny. Maybe change to "further impacting" or "further interaction with"

**Author reply:**

We have rephrased the original text as:

Lines 61-63:

SSAs can act as carrier agents for the vertical transport of much more than just sea salt and often include organic surfactants in the ocean, as already shown by field and laboratory studies (Cochran et al., 2016; Franklin et al., 2022; Rastelli et al., 2017).

Line 163: The **k** look wierd, should be $k_{SSA}$ as in Sha et al, right? Also chemical symbols should be upright, not italic.

**Author reply:**

Yes. We have replaced "**k**" by "$k_{SSA}$" in the revised manuscript.

Line 202: Unit, Part s$^{-1}$, would prefer just s$^{-1}$ or particles s$^{-1}$.

**Author reply:**

We have modified corresponding sentences and figures in the revised manuscript. All the SSA production units previously noted as "part s$^{-1}$" of seawater are now expressed as "particles s$^{-1}$".

Line 207: what does increase in bubble bursting refer to? Is it foam stability or lifetime? Or smaller and more bubbles?

**Author reply:**

The increase in bubble bursting here refers to smaller and more bubbles.

Line 260: change ball to sphere

**Author reply:**

We have modified corresponding sentences in the manuscript.

Lines 275-276:

Notably, it can be seen that the core morphology of salt particles had changed significantly, where the cubic structure has changed into a sphere structure.

Line 323: "lousy" is informal slang, change to "very poor" or just "poor" or "around and just below 1".

**Author reply:**

We thank the Reviewer for pointing this out, a new sentence has been shown in the revised manuscript.

Lines 342-343:

In SSA, $Ca^{2+}$ always exhibited high enrichment (EF > 1), while the EFs for $K^+$ and $Mg^{2+}$ were around and just below 1.

Line 351: … plays a very important role… Tone down, add "might play"

**Author reply:**

We have modified corresponding sentences in the manuscript.

Lines 373-374:

This demonstrates that the EF might play a very important role in the global emission fluxes of organic matter, in addition to concentration.

Figure 1: Maybe same range on y-axis? Easier to compare across subplots. I am colorblind, do you need the colors? If you perfer using colors, then should be the same as in Figure 2.

**Author reply:**

We have modified corresponding figures in the manuscript. We used the same range on y-axis and removed unnecessary colors.

[Figure]

**Fig. 1.** Measured surface tension values of natural seawater and aromatic acid-containing seawater: benzoic acids (A), benzene dicarboxylic acids (B), hydroxybenzoic acids (C), *p*-hydroxybenzoic acid, vanillic acid, and syringic acid (D). The dark spots represent Mean values of at least three measurements and the boxes represent the ranges of 25th−50th−75th percentiles.

Figure 2. I would change the colors. Yellow is difficult to see. Look at this website for inspiration: https://colorbrewer2.org/#type=diverging&scheme=BrBG&n=4

**Author reply:**

We thank the Reviewer for providing us with this useful website. We have changed the colors of Figure 2 in the revised manuscript.

[Figure]

**Fig. 2.** Number concentration distribution of sea salt particles and SSA particles containing benzoic acids (A), benzene dicarboxylic acids (B), hydroxybenzoic acids (C), *p*-hydroxybenzoic acid, vanillic acid, and syringic acid (D).

Figure 3. Seawater = Seawater only. And maybe add some errorbar estimation with respect to SSA production? Here color is okay, but that white shadow looks funny.

**Author reply:**

We have added error bar estimation with respect to SSA production of Figure 3 in the revised manuscript.

[Figure]

**Fig. 3.** SSA production, particle size, mass concentration distribution of aromatic acids. The symbol size represents the geometric mean diameter of SSA particles and is marked with numbers, and the symbol color indicates the particle mass concentration.

Figure 6: Maybe make different symbols and use one color? Feel free to ignore this comment.

**Author reply:**

We have tried to make different symbols in Figure 6 in the revised manuscript. Using the same color makes the symbols overlap. Hence, we used different colors.

[Figure]

**Fig. 6.** Enrichment factors of $K^+$, $Mg^{2+}$, and $Ca^{2+}$ in submicron SSA during the experiment.

Figure 7. The space between the bars are not the same. Also I would change colors (Feel free to ignore this comment)

**Author reply:**
We have made the space between the bars same and changed colors of Figure 7 in the revised manuscript.

[Figure]

**Fig. 7.** Estimated annual global aromatic acids emission (tons yr[-1]) via SSA. Yellow and blue stacked columns represent emissions based on Textor et al. (2006) and Jonas et al. (2021), respectively.

**Reviewer #2:**

Review of "Role of sea spray aerosol at the air-sea interface in transporting aromatic acids to the atmosphere" by Yaru Song et al.

**Summary**

Yaru Song et al. conducted a study to investigate the potential transport of organic acids, such as benzoic acids, from the ocean to the atmosphere via sea spray aerosols. The authors generated nascent sea spray aerosols in a laboratory setting using a sea spray simulation chamber equipped with a plunging water jet, which is a well-established method.

The authors aimed to determine the enrichment of the target organic acids in the aerosols compared to a sea spray tracer ion, sodium. Although the experimental principle aligns with previous successful studies (e.g., Johansson et al., 2019; Sha et al., 2021), the current manuscript lacks crucial details about the experimental procedures and quality control. This omission hinders the ability to assess the measurement quality. The manuscript fails to provide the necessary information to evaluate the reasonableness of the results and the value of the upscaled estimates of global emissions. Consequently, it lacks scientific significance in its present form. Therefore, I recommend rejecting the manuscript for publication in ACP.

To enhance the manuscript's scientific rigor, I suggest the authors make a series of major improvements which I have outlined below. Subsequently, I outline some more minor points for improvement.

**Author reply:**

We thank the Reviewer for the thoughtful comments and valuable suggestions that will contribute without doubt to improve our original manuscript. Based on these major and minor points below, we have revised our manuscript.

Major points

• In their introduction, the authors discuss the concept that the aromatic acids at the centre of their study can be of both natural and anthropogenic origin. Although this is an interesting and important discussion, the authors could expand on the relative importance of the different sources. As it stands, it is unclear whether most of the acids considered in this study are of natural or anthropogenic origin or whether they differ for the different acids investigated. A table summarising this information could help clarify the discussion.

**Author reply:**

For the aromatic acids we studied, benzoic and hydroxybenzoic acids as well as vanillic and syringic acids in seawater have both natural and anthropogenic sources, and benzene dicarboxylic acids are mainly derived from anthropogenic sources. We have rewritten some sentences and summarized in a table as follows.

Line 41-45:

In various observations, aromatic acids have been detected in both natural and anthropogenic sources (Zhao et al., 2019; Zangrando et al., 2019; Dekiff et al., 2014). Among them, natural sources of aromatic acids produced by algal releases account for most of marine aromatic acids, especially benzoic acid, and most of hydroxybenzoic acids (Mostafa et al., 2017; Fotso Fondja Yao et al., 2010a; Castillo et al., 2023; Abdel-Hamid A. Hamdy, 2020).

Line 50-53:

This is consistent with previous researches that phthalic acid is primarily derived from anthropogenic sources (Ren et al., 2023a), whereas hydroxybenzoic acid has both anthropogenic and natural sources (Zhao et al., 2019; Castillo et al., 2023).

Line 89-90:

Further details are provided in Table S1, which lists sources and concentrations of these aromatic acid identified in seawater and atmospheric samples over the ocean.

**Table S1.** Sources and concentrations of aromatic acids identified in seawater and atmospheric samples over the ocean.

| Aromatic acids | Natural sources | Anthropogenic sources |
|---|---|---|
| benzoic acid | <li>sea algae (Abdel-Hamid A. Hamdy, 2020; Al-Zereini et al., 2010; Fotso Fondja Yao et al., 2010b; Liu et al., 2022b)</li><li>sedimentary organic matter (10–65 μg g$^{-1}$) (Deshmukh et al., 2016)</li><li>bacteria isolated from sea bass viscera (314 ppb) (Martí-Quijal et al., 2020)</li><li>snow pit samples (2.11 ng g$^{-1}$) (Mochizuki et al., 2016)</li> | <li>emerging endocrine disrupting compounds (34–491 ng L$^{-1}$) (Zhao et al., 2019)</li><li>fuel combustion (Boreddy et al., 2017)</li><li>industrial wastewater, automobile exhaust and tobacco smoke (Cuadros-Orellana et al., 2006)</li> |
| *o*-phthalic acid | | <li>plasticizer (16.7–657 ng g$^{-1}$ d.w.) (Ren et al., 2023b; Sanjuan et al., 2023);</li><li>plastic waste burning (8.3–84.9 ng m$^{-3}$) (Zhu et al., 2022)</li><li>the end product of photochemical oxidation of SOA (15.5 ng m$^{-3}$) (Ding et al., 2021)</li><li>biomass burning and fossil fuel combustion sources (0.4–7.9 ng m$^{-3}$) (Shumilina et al., 2023; Yang et al., 2020; Boreddy et al., 2022)</li> |

| | | |
|---|---|---|
| *m*-phthalic acid | | • plasticizer (Ren et al., 2023b)
• the end product of photochemical oxidation of SOA ($3.6 \text{ ng m}^{-3}$) (Ding et al., 2021)
• biomass burning and fossil fuel combustion sources ($0.01–2.3 \text{ ng m}^{-3}$) (Yang et al., 2020; Boreddy et al., 2022; Kawamur, 2014) |
| *p*-phthalic acid | | • plasticizer ($0.51–6.8 \text{ mg kg}^{-1}$) (Ren et al., 2023b; Di Giacinto et al., 2023; Di Renzo et al., 2021)
• plastic waste burning ($10.8–80.7 \text{ ng m}^{-3}$) (Zhu et al., 2022); the end product of photochemical oxidation of SOA ($4.3 \text{ ng m}^{-3}$) (Ding et al., 2021)
• biomass burning and fossil fuel combustion sources ($0.05–2.5 \text{ ng m}^{-3}$) (Yang et al., 2020; Boreddy et al., 2022; Kawamur, 2014) |
| *o*-hydroxybenzoic acid | • sea algae ($76.8 \text{ mg L}^{-1}$) (Castillo et al., 2023; Mostafa et al., 2017; Klejdus et al., 2017) | • pharmaceuticals and drugs of abuse ($0.4–53.3 \text{ ng L}^{-1}$) (Alygizakis et al., 2016) |
| *m*-hydroxybenzoic acid | • sea algae (Al-Zereini et al., 2010; Castillo et al., 2023) | |

| | | |
|---|---|---|
| *p*-hydroxybenzoic acid | • sea algae (57.7 mg $L^{-1}$) (Castillo et al., 2023; Klejdus et al., 2017; Tian et al., 2012; Hawas and Abou El-Kassem, 2017)
• sea fungus (Rukachaisirikul et al., 2010; Shao et al., 2007)
• sponge Mycale species (Xuefeng Zhou, 2013); metabolite (Jingchuan Xue, 2015; Liao and Kannan, 2018)
• sediment samples (6.85–437 ng $g^{-1}$ dw) (Liao et al., 2019) | • Pharmaceuticals and personal care products (Lu et al., 2023)
• emerging endocrine disrupting compounds (4.58–49.9 ng $L^{-1}$) (Zhao et al., 2019; Lu et al., 2021; Alygizakis et al., 2016) |
| vanillic acid | • sea algae (3–47 ng $L^{-1}$) (Zangrando et al., 2019; Klejdus et al., 2017)
• lignin decomposition (Wang et al., 2015; Hu et al., 2022; Xu et al., 2017) | • combustion of both softwood and hardwood (Simoneit, 2022) |
| syringic acid | • sea algae (0.3–0.6 ng $L^{-1}$) (Poznyakovsky et al., 2021; Zangrando et al., 2019; Klejdus et al., 2017)
• lignin decomposition (Hu et al., 2022; Xu et al., 2017) | • pharmaceuticals (Fisch et al., 2017)
• hardwood burning (Simoneit, 2022) |

• The description of the process of sea spray aerosol formation could be improved. For example, the initial description on line 57 is rather cumbersome and lacks accuracy. The authors can draw inspiration from the wealth of literature available on this topic to improve this section.

**Author reply:**

We have modified the original text as follows.

Line 59-67:

Sea spray aerosols (SSAs), generated by breaking waves and bubble bursting, are one of the major sources of atmospheric particles (Andreae and Rosenfeld, 2008; Angle et al., 2021; Hasenecz et al., 2020; Malfatti et al., 2019). SSAs can act as carrier agents for the vertical transport of much more than just sea salt and often include organic surfactants in the ocean, as already shown by field and laboratory studies (Cochran et al., 2016; Franklin et al., 2022; Rastelli et al., 2017). Recent data indicated that the surface activity and octanol-water partitioning coefficients ($K$ow) of organic compounds may affect their transport efficiency from the water phase to the atmosphere (Olson et al., 2020). Moreover, molecular structure may induce changes of organic acids properties (i.e., surface tension, toxicity), which further affect their transport potential and global emission flux (Lee et al., 2017; Rastelli et al., 2017; Frossard et al., 2019; Van Acker et al., 2021).

• The authors used a common approach to mimic the process of sea spray aerosol formation in the laboratory, namely a sea spray simulation chamber. Although such systems have become quite common, there is diversity among the systems in use to the extent that it is important to be very precise in describing the particulars of the system in use. Since the characteristics of the different systems used can influence the properties of the aerosols generated, certain information must be conveyed to the reader. For example, the authors used a plunging jet-type system. It is important to include all the relevant details of the system used, such as the diameter of the nozzle through which the seawater flowed into the chamber, the type of pump used to generate the plunging jet, the distance between the exit of the nozzle and the surface of the seawater in the chamber, and the material from which the chamber itself was fabricated.

**Author reply:**

We have added more contents to make the manuscript more complete.

Lines 99-100:

A jet-based laboratory SSA simulation chamber, made of 316L stainless steel material, was used to mimic the SSA generation (Fig.S2).

Lines 104-106:

When used, plunging jets were cycled in seawater by the pump (Shenchen V6-6, China) at a flow rate of 1 L min$^{-1}$ through a stainless-steel nozzle (inner diameter 4.3 mm).

Line 103:

The detailed dimensions of the SSA simulation chamber are provided in Table S2.

**Table S2.** Dimensions and operating conditions of the SSA simulation chamber.

| Characteristic | Value |
|---|---|
| Nozzle diameter (mm) | 4.3 |
| Seawater depth (cm) | 15 |
| Seawater volume (L) | 9 |
| Headspace depth (cm) | 22 |
| Headspace volume (L) | 15 |
| Zero sweep air (L min$^{-1}$) | 3 |
| Headspace residence time (min) | 5 |
| Plunging jet flow rate (L min$^{-1}$) | 1 |

The reasons for the selection of the relevant parameters are as follows:

1) A stainless steel nozzle with an inner diameter of 4.3 mm is used to generate plunging jets. The nozzle can be changed easily. The size of the nozzle is consistent with that in previous studies (Salter et al., 2014; Sha et al., 2021).

2) We have studied the cases with headspace heights of 14 cm, 19 cm, and 22 cm in our previous work (Liu et al., 2022a) and found that the SSA production was most efficient at a headspace height of 22 cm. Hence, we chose it for SSA generation.

3) We have tried pump flow rates of 0.50 L min$^{-1}$, 0.75 L min$^{-1}$, 1.00 L min$^{-1}$, 1.25 L min$^{-1}$, and 1.50 L min$^{-1}$ in our previous study (Liu et al., 2022a) and found that when the pump flow rate was 1 L min$^{-1}$ or higher, the performance of the SSA generator was good (see Table R1). Therefore, we used a pump flow rate of 1 L min$^{-1}$ to maintain a high SSA production rate and a relatively long working lifetime of the corresponding accessories.

Table R1. Effect of pump flow rate on SSA production.

| Pump flow rate (L min$^{-1}$) | SSA production (particles s$^{-1}$) | Mass concentration (μg m$^{-3}$) |
|---|---|---|
| 0.50 | $5.78 \times 10^6$ | 198.96 |
| 0.75 | $9.90 \times 10^6$ | 317.31 |
| 1.00 | $1.37 \times 10^7$ | 417.86 |
| 1.25 | $1.58 \times 10^7$ | 469.59 |
| 1.50 | $1.63 \times 10^7$ | 495.42 |

4) We have previously studied SSA production at purge air flow rates of 2 L min$^{-1}$, 3 L min$^{-1}$, 4 L min$^{-1}$, 5 L min$^{-1}$, 6 L min$^{-1}$ to simulate the sea breeze (Liu et al., 2022a). We found that the SSA production increased with the purge air flow rate when the flow rate was less than 3 L min$^{-1}$, and decreased with the increase of the purge air when the flow rate was more than 3 L min$^{-1}$.

5) Based on a previous study of the effect of relative humidity on the growth of sea salt particles, the inlet humidity would affect the morphology of SSA (Tang et al., 1997). Hence, we chose to keep the relative humidity at about 40% in our study. Particles will deliquescence at higher humidity (Bryan et al., 2022). In addition, it has been demonstrated that acid has no effect on NaCl deliquescence (Ming and Russell, 2001). Combining the above parameters, we obtained a particle size distribution similar to that from field observations (Quinn et al., 2017; Xu et al., 2022).

[Figure]

**Fig. S2.** Number size distribution of SSA generated with the SSA simulation chamber in this study compared with field studies.

• It is also important to note that seawater temperature has been shown to influence the process of sea spray aerosol formation. Therefore, it would be helpful to know whether the seawater sample temperature in the chamber was controlled or monitored. Additionally, it would be useful to know how the chamber was cleaned between experiments. Furthermore, in the schematic shown in Figure S1, a 'sweep flow' of particle-free air into the chamber is shown, while the only outlet was to the SMPS." The authors did not specify where the excess air went. Did they not operate with an "overflow" exit to ensure that the chamber was always operated at atmospheric pressure? Additionally, was the Dekati LPI connected to the same outlet as the SMPS system or was it connected to another outlet so that both measurements were conducted simultaneously?

**Author reply:**

1) Seawater temperature:

We controlled the laboratory temperature around 22–25°C during the experiments and examined the seawater temperature in the SSA simulation chamber before and after the experiments, and the corresponding results are supplemented in Table S3.

**Table S3.** Summary of experimental conditions.

| Exp. No. | Experiment type | Concentration (mM) | pH | Salinity (psu) | Sampling time (h) | RH (%) | Temperature difference (°C) [a] |
|---|---|---|---|---|---|---|---|
| 1 | SW | 0 | 7.92 | 34.2 | 5 | 35 | 2.0 |
| 2 | SW+benzoic acid | 1 | 7.72 | 34.3 | 5 | 34 | 1.5 |
| 3 | SW+$o$-hydroxybenzoic acid | 1 | 7.60 | 34.5 | 5 | 36 | 1.0 |
| 4 | SW+$m$-hydroxybenzoic acid | 1 | 7.68 | 34.1 | 5 | 40 | 2.0 |
| 5 | SW+$p$-hydroxybenzoic acid | 1 | 7.84 | 34.3 | 5 | 38 | 1.5 |
| 6 | SW+$o$-phthalic acid | 1 | 7.58 | 34.2 | 5 | 36 | 2.0 |
| 7 | SW+$m$-phthalic acid | 1 | 7.80 | 34.5 | 5 | 37 | 2.5 |
| 8 | SW+$p$-phthalic acid | 1 | 7.85 | 34.4 | 5 | 42 | 2.0 |
| 9 | SW+vanillic acid | 1 | 7.81 | 34.2 | 5 | 43 | 3.0 |
| 10 | SW+syringic acid | 1 | 7.84 | 34.3 | 5 | 39 | 2.0 |
| 11 | ASW | 0 | 7.96 | 35.1 | 5 | 33 | 1.5 |
| 12 | ASW+benzoic acid | 1 | 7.68 | 34.6 | 5 | 35 | 1.0 |
| 13 | ASW+$o$-hydroxybenzoic acid | 1 | 7.76 | 34.9 | 5 | 34 | 0.5 |
| 14 | ASW+$m$-hydroxybenzoic acid | 1 | 7.99 | 35.3 | 5 | 36 | 1.5 |
| 15 | ASW+$p$-hydroxybenzoic acid | 1 | 7.85 | 34.7 | 5 | 38 | 2.0 |
| 16 | ASW+$o$-phthalic acid | 1 | 7.93 | 34.5 | 5 | 35 | 1.0 |

| 17 | ASW+$m$-phthalic acid | 1 | 7.88 | 34.9 | 5 | 36 | 1.0 |
|---|---|---|---|---|---|---|---|
| 18 | ASW+$p$-phthalic acid | 1 | 7.97 | 34.6 | 5 | 34 | 1.5 |
| 19 | ASW+vanillic acid | 1 | 7.89 | 35.2 | 5 | 35 | 1.0 |
| 20 | ASW+syringic acid | 1 | 7.99 | 34.8 | 5 | 39 | 1.0 |
| 21 | ASW+benzoic acid+$o$-hydroxybenzoic acid+$o$-phthalic acid+vanillic acid+syringic acid | $10^{-3}$ | 7.95 | 35.1 | 20 | 41 | 3.5 |
| 22 | ASW+benzoic acid+$m$-hydroxybenzoic acid+$m$-phthalic acid+vanillic acid+syringic acid | $10^{-3}$ | 7.98 | 34.6 | 20 | 38 | 1.5 |
| 23 | ASW+benzoic acid+$p$-hydroxybenzoic acid+$p$-phthalic acid+vanillic acid+syringic acid | $10^{-3}$ | 7.88 | 34.9 | 20 | 40 | 2.0 |
| 24 | NaCl | 0 | 7.68 | 35.3 | 5 | 38 | 1.0 |
| 25 | NaCl+$m$-hydroxybenzoic acid | 1 | 7.54 | 34.7 | 5 | 36 | 1.5 |

[a] The temperature difference in the SSA simulation chamber before and after the experiment.

2) SSA simulation chamber cleaning:

Between experiments, the SSA simulation chamber was cleaned with ethanol first, then rinsed with ultrapure water, blown with zero air to reduce the influence of organic acids residue. The above steps run the pump to allow for thorough cleaning of the system.

**S1.** Quality assurance/quality control.

Seawater was collected from the coastal area of Shazikou on March 27, 2023, with a volume of 500 L (Fig. S1). Considering the storage inconvenience caused by huge consumption of seawater, all our seawater was pre-filtered through a polyethersulfone filter (47 mm diameter, 0.2 µm pore size, Supor®-200, Pall Life Sciences, USA) and stored in the dark at 18 °C for less than one month. Quinn et al. (2015) have shown that the fraction that passes through the filter is regarded as dissolved organic carbon and includes colloidal and truly dissolved materials. For each experiment, we measured particle number concentrations generated by filtered seawater and cations concentrations in seawater, and we found good agreement between each set of experiments (see Fig. S2).

In order to avoid the influence of organic matter in quartz fiber filters and access the accuracy of the experiment, pre-baked quartz fiber filters were used in sampling. Before each set of experiments, experimental blanks were conducted using filtered seawater. Experimental blanks were conducted with the same procedure of SSA samples. Seawater and filter samples were stored at -20°C until analyzed. In order to reduce the influence of organic acids residue after each experiment, the SSA simulation chamber was cleaned with ethanol first, then the system was cleaned with ultra-pure water for several times. The above steps also run the pump to allow for thorough cleaning of the system. Thereafter, the system was blown with zero air and sealed for preservation. The Dekati DLPI was also ultrasonicated with methanol and water (V:V=1:1) and dried after the experiment.

Filtered seawater (without added aromatic acid) was used as the experimental blank, and the same experimental and analytical methods were used as those for the experimental samples. As a result, no target aromatic acid was found in both seawater and filters. This may be due to the fact that we did not perform any concentration operation during the seawater sample processing. The standard curves for each aromatic acid are linear, as shown in Fig. S5.

[Figure]

**Fig. S1.** Sampling site at Shazikou along the Yellow Sea coast, Qingdao, China.

[Figure]

**Fig. S2.** Number size distribution of SSA generated with the SSA simulation chamber in this study compared with field studies.

[Figure]

**Fig. S5.** The standard curves for aromatic acids were constructed within a concentration range of 0.01-1000 μM, with more than seven data points.

3) Experimental setup:

We used a three-way valve to allow excess gas to escape, ensuring that the chamber operates at atmospheric pressure. The DeKati DLPI+ was connected to the same outlet as the SMPS system. The stabilization of the particles total number concentration suggests that there is little SSA particle adsorption in the chamber. The decrease in aromatic acid concentration may then be due to the transfer of SSA. Therefore, the sample collection for the SMPS system and the DeKati DLPI+ were conducted separately.

We inserted these sentences and figures in the revised manuscript and supplement to make it clearer.

Lines 111-112:

The outlet was fitted with a three-way valve to transfer the sample airflow and the excess gas.

Lines 128-129:

After the size distribution stabilized, a total of 40 mL of seawater is collected from a tap located 1.5 cm above the bottom of the chamber.

[Figure]

**Fig. S3.** Schematic picture of the plunging jet-sea spray aerosol generator: SMPS sampling (A), single particle sampling (B), and DeKati DLPI+ sampling (C). The red arrows represent the flow direction of seawater, and the purple arrows represent the flow of gases and aerosol particles.

• When evaluating data from sea spray simulation systems, it is important to consider the size of the bubble plume generated by the plunging jet compared to the chamber itself. For instance, did the bubble plume reach the bottom of the chamber? If the surface foam patch is also relatively large in comparison to the surface area of the water, then "wall effects" can occur whereby bubbles reach the side and burst faster than they would if the sides had not affected the bubbles. Therefore, some insight here (perhaps some photographs) would add value to the manuscript.

**Author reply:**

1) The bubble plume reached the bottom of the SSA simulation chamber.

2) The physical diagram of the experiment and the top view of the bubble bursting are shown below. Furthermore, due to the similar enrichment levels of 1 µM and 1 mM aromatic acids in SSA, we speculate that the main factor influencing the enrichment factor is aerosolization rather than the "wall effect".

The following text was added in the revised manuscript.

Line 107-108:

During the experiment, it was assumed that the interaction of the bubble plume generated by the plunging jet with the wall was negligible (Fig. S4).

[Figure]

**Fig. S4.** Physical diagram of the SSA simulation chamber (A) and the top view of the bubble generation in the chamber (B).

• Regarding the use of a Dekati LPI and a TEM sampler to obtain offline samples for subsequent analysis, the authors did not mention obtaining blank samples to check for handling impacts. It is unclear whether blank substrates were loaded into the samplers and then analyzed in the same way as the samples.

**Author reply:**

During the experiment, we used filtered seawater (no aromatic acids added) as the experimental blank and then analyzed in the same way as the samples. We measured the concentration of aromatic acid in filtered seawater and filters all below the detection limit. The TEM images of the filtered seawater (no aromatic acids added) have been shown in Fig. 4A.

[Figure]

**Fig. 4.** Particle morphology observed using TEM of sea salt (A) and mixed particles composed of aromatic acids-coated sea salt particles (B–K).

• When it comes to the results of their experiments, the authors first report the impact of different organic acids on surface tension. However, they did not provide enough information to discern how the experiments were carried out. It is assumed that the authors added or "spiked" a known concentration of the different compounds to the seawater samples they collected from the Yellow Sea. However, the concentration of the different compounds used, whether the seawater samples already contained some or all of the compounds, and the variability of the concentration of the analytes of interest in these seawater samples are not mentioned. Additionally, the presence of other surfactants in the form of organic matter derived from natural and anthropogenic sources, and how they could affect the results, is not discussed. All of this information is critical to helping the reader discern the implications of the results. Since natural seawater has variable amounts of these compounds, it is suggested that the authors could have "spiked" the analytes of interest into artificial seawater to try to negate this

effect.

**Author reply:**

1) The experiment consisted of a total of 25 sets with target compound concentrations of $10^{-3}$ and 1 mM (Table S3). Each aromatic acid was added individually to seawater for the experiment. The SMPS recorded particle number concentrations, and sampling was performed once the size distributions stabilized. Seawater samples, totaling 40 mL, were collected from a tap located 1.5 cm above the bottom of the chamber. After collection, quartz fiber filter (QFF) containing aromatic acids was subjected to a 40-minute ultrasonic extraction in 4 mL of Milli-Q water to dissolve submicron SSA particles. The solution was then filtered through a 0.45 μm syringe filter to eliminate filter fragments, for subsequent concentration calculations and EF analysis. After each experiment, the SSA chamber underwent thorough cleaning as outlined in QA/QC (see S1).

We have inserted the following text in the revised manuscript to make the experimental process clearer.

Lines 176-179:

The experiment consisted of a total of 25 sets with target compound concentrations of $10^{-3}$ and 1 mM (Table S3). Each aromatic acid was added individually to seawater for the experiment. Surface tension of seawater was examined using a surface tensiometer (Sigma 700, Biolin Scientific, Sweden) equipped with a Wihelmy plate, which was calibrated at 25 °C with 30 mL ultrapure water.

Lines 128-129:

After the size distribution stabilized, a total of 40 milliliters of seawater was collected from a tap located 1.5 cm above the bottom of the chamber.

**Table S3.** Summary of experimental conditions.

| Exp. No. | Experiment type | Concentration (mM) | pH | Salinity (psu) | Sampling time (h) | RH (%) | Temperature difference (°C) [a] |
|---|---|---|---|---|---|---|---|
| 1 | SW | 0 | 7.92 | 34.2 | 5 | 35 | 2.0 |
| 2 | SW+benzoic acid | 1 | 7.72 | 34.3 | 5 | 34 | 1.5 |
| 3 | SW+$o$-hydroxybenzoic acid | 1 | 7.60 | 34.5 | 5 | 36 | 1.0 |
| 4 | SW+$m$-hydroxybenzoic acid | 1 | 7.68 | 34.1 | 5 | 40 | 2.0 |
| 5 | SW+$p$-hydroxybenzoic acid | 1 | 7.84 | 34.3 | 5 | 38 | 1.5 |
| 6 | SW+$o$-phthalic acid | 1 | 7.58 | 34.2 | 5 | 36 | 2.0 |
| 7 | SW+$m$-phthalic acid | 1 | 7.80 | 34.5 | 5 | 37 | 2.5 |
| 8 | SW+$p$-phthalic acid | 1 | 7.85 | 34.4 | 5 | 42 | 2.0 |
| 9 | SW+vanillic acid | 1 | 7.81 | 34.2 | 5 | 43 | 3.0 |
| 10 | SW+syringic acid | 1 | 7.84 | 34.3 | 5 | 39 | 2.0 |
| 11 | ASW | 0 | 7.96 | 35.1 | 5 | 33 | 1.5 |
| 12 | ASW+benzoic acid | 1 | 7.68 | 34.6 | 5 | 35 | 1.0 |
| 13 | ASW+$o$-hydroxybenzoic acid | 1 | 7.76 | 34.9 | 5 | 34 | 0.5 |
| 14 | ASW+$m$-hydroxybenzoic acid | 1 | 7.99 | 35.3 | 5 | 36 | 1.5 |
| 15 | ASW+$p$-hydroxybenzoic acid | 1 | 7.85 | 34.7 | 5 | 38 | 2.0 |
| 16 | ASW+$o$-phthalic acid | 1 | 7.93 | 34.5 | 5 | 35 | 1.0 |

| | | | | | | | |
|---|---|---|---|---|---|---|---|
| 17 | ASW+*m*-phthalic acid | 1 | 7.88 | 34.9 | 5 | 36 | 1.0 |
| 18 | ASW+*p*-phthalic acid | 1 | 7.97 | 34.6 | 5 | 34 | 1.5 |
| 19 | ASW+vanillic acid | 1 | 7.89 | 35.2 | 5 | 35 | 1.0 |
| 20 | ASW+syringic acid | 1 | 7.99 | 34.8 | 5 | 39 | 1.0 |
| 21 | ASW+benzoic acid+*o*-hydroxybenzoic acid+*o*-phthalic acid+vanillic acid+syringic acid | $10^{-3}$ | 7.95 | 35.1 | 20 | 41 | 3.5 |
| 22 | ASW+benzoic acid+*m*-hydroxybenzoic acid+*m*-phthalic acid+vanillic acid+syringic acid | $10^{-3}$ | 7.98 | 34.6 | 20 | 38 | 1.5 |
| 23 | ASW+benzoic acid+*p*-hydroxybenzoic acid+*p*-phthalic acid+vanillic acid+syringic acid | $10^{-3}$ | 7.88 | 34.9 | 20 | 40 | 2.0 |
| 24 | NaCl | 0 | 7.68 | 35.3 | 5 | 38 | 1.0 |
| 25 | NaCl+*m*-hydroxybenzoic acid | 1 | 7.54 | 34.7 | 5 | 36 | 1.5 |

[a] The temperature difference in the SSA simulation chamber before and after the experiment.

2) During the experimental process, filtered seawater (without the addition of aromatic acids) was employed as a procedural blank. Subsequently, analysis was conducted using the same methods as applied to the samples. However, the targeted aromatic acids were not detected in the seawater, possibly due to their concentrations being below the detection limit. We have added relevant information in the Supplement.

Filtered seawater (without added aromatic acid) was used as the experimental blank, and the same experimental and analytical methods were used as those for the experimental samples. As a result, no aromatic acid was found in both seawater and filtered samples. This may be due to the fact that we did not perform any concentration operation during the water sample processing.

3) ASW were selected as the bulk water to characterize the influence of organic matter in our previous study (Song et al., 2022). We concluded that considering ASW as bulk waters will result in a deviation of SSA production from that produced by seawater. And the particles size distribution obtained by the lognormal-mode-fitting approach in our previous work (Fig. S2) is very similar to the distributions observed in previous studies (Xu et al., 2022; Quinn et al., 2017).

[Figure]

**Fig. S2.** Number size distribution of SSA generated with the SSA simulation chamber in this study compared with field studies.

Additionally, to enhance the rigor of our results, we have "spiked" the target analytes into artificial seawater, and the following text has been added in the revised manuscript. Lines 177-179:

Each aromatic acid was added individually to seawater and artificial seawater (ASW) for the experiment. Surface tension of seawater was examined using a surface tensiometer (Sigma 700, Biolin Scientific, Sweden) equipped with a Wihelmy plate, which was calibrated at 25 °C with 30 mL ultrapure water.

Lines 181-183:

However, the average surface tension value of ASW was 74.5 mN m$^{-1}$ (Fig. S6), indicating that the presence of surfactants in seawater enhances its surface activity.

[Figure]

**Fig. S6.** Measured surface tension values of artificial seawater (ASW) and aromatic acid-containing ASW.

From the results, we can see that the effect trend of aromatic acids on seawater surface tension is generally similar, not only in seawater but also in artificial seawater.

Furthermore, we also measured the SSA particle number concentration with aromatic acids and the EF of aromatic acids, the results are as follows.

[Figure]

**Fig. S7.** Number concentration distribution of sea salt particles and SSA particles containing benzoic acids (A), benzene dicarboxylic acids (B), hydroxybenzoic acids (C), vanillic acid and syringic acid (D).

From the results, we can see that the effect trend of aromatic acids on SSA production is generally similar, not only in seawater but also in artificial seawater.

[Figure]

**Fig. S8.** Enrichment factors of aromatic acids at different concentrations from artificial seawater to the atmosphere.

Comparing the EF of aromatic acids in SW and ASW, it was observed that the EF trends of benzene dicarboxylic acids in seawater follows the pattern: *o*-phthalic acid < *m*-phthalic acid < *p*-phthalic acid. However, in ASW, the EF of *p*-phthalic acid was lower than that of *m*-phthalic acid. Based on the findings of Li et al. (2023), we hypothesize that in ASW, *p*-phthalic acid acts as an ·OH scavenger to produce TAOH. Hence, the EF of *p*-phthalic acid is lower than that of *m*-phthalic acid. Meanwhile, organic compounds in SW preferentially react with ·OH (Anastasio and Newberg, 2007), thus the EF of *p*-phthalic acid is the highest among the benzene dicarboxylic acids. Furthermore, differences in aromatic acid concentration did not change the enrichment pattern of organic acids.

• The authors then report the impact of different acids on the size and number of particles that are emitted as SSA. It is unclear what the concentrations of the different analytes in these experiments were. It is also unclear whether the same seawater was used for all experiments or whether different seawater samples were spiked. The concentration of the analytes in the seawater sample and the level of organic matter in the seawater are not mentioned.

**Author reply:**

1) We added 1 mM of individual aromatic acids to natural seawater. In artificial seawater, $10^{-3}$ and 1mM aromatic acids were added, and the specific experimental

groups can be found in Table S3.

Lines 176-177:

The experiment consisted of a total of 25 sets with target compound concentrations of $10^{-3}$ and 1 mM (Table S3).

2) We used the same seawater for all experiments. Seawater was collected from the coastal area of Shazikou on March 27, 2023, with a volume of 500 L (Fig. S1). Considering the storage inconvenience caused by huge consumption of seawater, all our seawater was pre-filtered through a polyethersulfone filter (47 mm diameter, 0.2 μm pore size, Supor®-200, Pall Life Sciences, USA) and stored in the dark at 18 °C for less than one month. For each experiment, we measured particle number concentrations generated by filtered seawater and cations concentrations in seawater, and we found good agreement between each set of experiments (see Fig. R1). Furthermore, for comparing the properties of SSA particles containing aromatic acids, we normalized the particles size distribution of seawater before adding aromatic acids.

[Figure]

**Fig. R1.** Mean value of total particle number concentration of SSA ($N_{Total}$) and concentration of $Na^+$ ($C(Na^+)$) for each experiment.

3) We measured the standard curve for aromatic acids and it was linear over the

measurement range. Therefore, we used the peak area directly for the EF calculation.

[Figure]

**Fig. S5.** Standard curves for aromatic acids were constructed within a concentration range of 0.01-1000 μM, with more than seven data points.

**Table R2.** Peak areas of aromatic acids in seawater and SSA filter samples.

| | Group 1 | | Group 2 | | Group 3 | | Group 4 | |
|---|---|---|---|---|---|---|---|---|
| | SW | SSA | SW | SSA | SW | SSA | SW | SSA |
| benzoic acid | 159632 | 2732.77 | 196860 | 5206.99 | 185135 | 1540.1 | 167892 | 2508.47 |
| *o*-phthalic acid | 123785 | 1226.6 | 168974 | 1802.2 | 156324 | 1888.7 | | |
| *m*-phthalic acid | 466987 | 6606.85 | 458769 | 5432.25 | 491623 | 4880.15 | | |
| *p*-phthalic acid | 454593 | 10626.92 | 496102 | 21471.43 | 445985 | 15614.82 | 471582 | 20110.87 |
| *o*-hydroxybenzoic acid | 257548 | 2482 | 232946 | 3844.08 | 238358 | 5167.06 | | |
| *m*-hydroxybenzoic acid | 170482 | 8532.59 | 160309 | 7072.09 | 190181 | 4732.93 | | |
| *p*-hydroxybenzoic acid | 169753 | 3635.99 | 153257 | 4467.54 | 131186 | 2297.87 | 171062 | 4329.58 |
| vanillic acid | 357000 | 4948.01 | 287470 | 4696.83 | 279462 | 12750.11 | | |
| syringic acid | 486640 | 23833.92 | 467129 | 29328.75 | 491816 | 11993.31 | | |

• Following this, the authors present enrichment factors of the different organic acids on the nascent sea spray aerosol they generate. However, many details are again missing, which makes it impossible to evaluate the results. For example, it is not clear to me whether the presented results are the average over all stages of the impactor or if they only represent a single stage. Furthermore, the authors did not mention how they quantified the sodium ion and the concentration of the organic acids in the same sample. Did the authors extract the substrates in ultrapure water and then subsample these extracts for the different analyses, or did they use a different approach? To generate their enrichment factor estimates, the authors will have used a concentration value for each of the acids in the seawater used to generate the aerosols. Was this value quantified, or did the authors simply use the "spike" concentration they aimed for? This is important given that surface active species, such as the organic acids used, have a tendency to stick to the walls of chambers and the actual concentration of the organic acids in the seawater may well have been well below the "spike" value the authors aimed for. Along the same lines, assuming that the authors did quantify the actual concentration of the organic acids in the seawater, it would be good to know at what point of the experiment they obtained these samples. For example, the authors state that the LPI was run for 5 hours - were water samples taken at the start or end of this period? In similar experiments using perfluoroalkyl acids, Johannsson et al. (2019) observed that the concentration of the substances in the seawater from which they were generating aerosol during their experiment decreased as these substances were "lost" to the aerosol.

**Author reply:**

1) Impactor sampling:

The present results are averaged over all stages of the impactor with particle size less than 1 μm.

Lines 144-145:

Submicron SSA particles were impacted onto 25-mm diameter quartz fiber filters (QFF, 1851–025, Waterman, UK), which were baked in muffle furnace at 450 °C for 3 h.

2) Quantification of concentration:

We extracted the substrates in ultrapure water and then subsampled these extracts for analyses. We further measured the standard curve for the aromatic acids and it was linear over the measurement range. Therefore, we used the peak area directly for the subsequent EF calculation.

**Table R2.** Peak areas of aromatic acids in seawater and SSA filter samples.

| | Group 1 | | Group 2 | | Group 3 | | Group 4 | |
|---|---|---|---|---|---|---|---|---|
| | SW | SSA | SW | SSA | SW | SSA | SW | SSA |
| benzoic acid | 159632 | 2732.77 | 196860 | 5206.99 | 185135 | 1540.1 | 167892 | 2508.47 |
| *o*-phthalic acid | 123785 | 1226.6 | 168974 | 1802.2 | 156324 | 1888.7 | | |
| *m*-phthalic acid | 466987 | 6606.85 | 458769 | 5432.25 | 491623 | 4880.15 | | |
| *p*-phthalic acid | 454593 | 10626.92 | 496102 | 21471.43 | 445985 | 15614.82 | 471582 | 20110.87 |
| *o*-hydroxybenzoic acid | 257548 | 2482 | 232946 | 3844.08 | 238358 | 5167.06 | | |
| *m*-hydroxybenzoic acid | 170482 | 8532.59 | 160309 | 7072.09 | 190181 | 4732.93 | | |
| *p*-hydroxybenzoic acid | 169753 | 3635.99 | 153257 | 4467.54 | 131186 | 2297.87 | 171062 | 4329.58 |
| vanillic acid | 357000 | 4948.01 | 287470 | 4696.83 | 279462 | 12750.11 | | |
| syringic acid | 486640 | 23833.92 | 467129 | 29328.75 | 491816 | 11993.31 | | |

3) Sampling time:

We obtained seawater samples after the SMPS-monitored total number concentrations had stabilized, and prior to DLPI+ sampling.

Lines 128-129:

After the size distribution stabilized, a total of 40 milliliters of seawater is collected from a tap located 1.5 cm above the bottom of the chamber.

• On Line 84, the authors introduce the idea of upscaling their measurements using literature values for the seawater concentration of these compounds. However, the reader requires more information at this juncture. Questions arise, such as: What are the typical seawater concentrations of these compounds? How much do they vary globally? How extensive is the dataset of such measurements? Incorporating this information into the previously introduced table (major point 1) would be beneficial. While the authors eventually provide some literature values in section 3.4 of the manuscript, it remains unclear which specific values were utilised. Was it an average of the literature values or a different approach? Considering the likely high spatial (and potentially temporal) variability in these concentrations, it is crucial to understand how the authors addressed this variability. Presently, it is nearly impossible to evaluate the magnitude of the emissions proposed by the authors due to the uncertainty in the conducted measurements.

**Author reply:**

Based on the concentration values summarized from Table S1 in our study, we calculated the annual emission flux of SSA within the concentration range reported in the literature. For concentration selection, we chose the actual detected concentrations of aromatic acids in seawater rather than the concentrations found in sea algal extracts. The concentrations we used are shown in Table S5. We calculated the range of aromatic acid emissions within the concentration range reported in the literature (Zangrando et al., 2019; Zhao et al., 2019). The following reference has been cited in the revised manuscript.

Lines 351-353:

For example, the seawater concentrations of syringic acid, *p*-hydroxybenzoic acid, benzoic acid are 0.3, 4.58, 34 ng L$^{-1}$, respectively (Zhao et al., 2019).

[Figure]

**Fig. 7.** Estimated annual global aromatic acids emission (tons yr$^{-1}$) via SSA. Yellow and blue stacked columns represent emissions based on Textor et al. (2006) and Jonas et al. (2021), respectively.

**Minor points**

• Line 30: I don't think "captured" is the right word here. Perhaps "taken up" would be better.

Author reply:

We have replaced "captured" by "taken up" in the revised manuscript.

Lines 30-32:

On the one hand, aromatic acids are readily to be taken up by marine organisms, leading to their enrichment within these organisms or transportation to remote areas (Fu et al., 2011; Shariati et al., 2021; Wang and Kawamura, 2006).

• Line 31: Would read better as: "...leading to their enrichment within these organisms or transportation to remote areas (Fu et al.,..."

Author reply:

We have modified corresponding sentences.

• Line 31: I would also break this sentence to aid readability, e.g.: "This process poses health risks to the endocrine system of aquatic organisms and the overall marine ecosystem (Saha et al., 2006)."

Author reply:

We have modified corresponding sentences. Please, refer to our reply to the first minor comment for the detailed changes.

Lines 32-33:

This process poses health risks to the endocrine system of aquatic organisms and the overall marine ecosystem (Saha et al., 2006).

• Line 35: Would read better as "Previous studies suggest that these organic acids can alter the composition of SSA, subsequently influencing atmospheric processes such as cloud condensation nuclei (CCN) or ice nuclei (IN) activities.

Author reply:

We have modified corresponding sentences in the revised manuscript.

Lines 35-38:

Previous studies suggest that these organic acids can alter the composition of SSA, subsequently influencing atmospheric processes such as cloud condensation nuclei

(CCN) or ice nuclei (IN) activities.

• Line 36: Needs rephrasing. It is not the "organic acids" that are the source of the aerosol. Rather, it is SSA that may contain organic acids that play a role in Earth's climate.

**Author reply:**

We thank the Reviewer for pointing this out. A new sentence has been shown in the revised manuscript.

Lines 38-39:

SSA may contain organic acids that play a major role in Earth's climate (Moore et al., 2011; Zhu et al., 2019).

• Line 50: Would read better as "...the identification of phthalic acid in organisms raises the possibility of its origin from the ingestion of marine plastics."

**Author reply:**

We have modified corresponding sentences in the revised manuscript.

Lines 49-50:

Importantly, in addition to biodegradation, the identification of phthalic acid in organisms raises the possibility of its origin from the ingestion of marine plastics (Almulhim et al., 2022).

• Line 53: Would read better as "This phenomenon has prompted discussions about the concept of" missing aromatic acids."

**Author reply:**

We have modified corresponding sentences in the revised manuscript.

Lines 53-55:

Recent laboratory studies have shown that personal-care products, especially sunscreen (e.g., $o$-hydroxybenzoic acid), are reduced in levels during algal blooms (Franklin et al., 2022). This phenomenon has prompted discussions about the concept of "missing aromatic acids".

• Line 54: Would read better as "Nevertheless, the currently available data do not provide a conclusive explanation for the existence of these" missing aromatic acids."

**Author reply:**

We have modified corresponding sentences in the revised manuscript.

Lines 55-56:

Nevertheless, the currently available data do not provide a conclusive explanation for the existence of these "missing aromatic acids".

• Line 55: This paragraph could be better linked to the previous discussion. For example,"One possible reason for the disappearance of these aromatic acids is their release into the atmosphere. Existing datasets obtained from remote marine areas offer evidence of the presence of these compounds in the atmosphere (Fu et al., 2010)."

**Author reply:**

We have modified corresponding sentences in the revised manuscript.

Lines 57-59:

One possible reason for the disappearance of these aromatic acids is their release into the atmosphere. Existing datasets obtained from remote marine areas offer evidence of the presence of these compounds in the atmosphere (Fu et al., 2010).

• Line 66: Would read better: "While field studies have demonstrated the presence of aromatic acids in both the ocean and atmosphere (Boreddy et al., 2017), the specific mechanisms influencing their transport at the air-sea interface require further investigation."

**Author reply:**

We have modified corresponding sentences in the revised manuscript.

Lines 67-69:

While field studies have demonstrated the presence of aromatic acids in both the ocean and atmosphere (Boreddy et al., 2017), the specific mechanisms influencing their transport at the air-sea interface require further investigation.

• Line 69: Here the authors introduce the air-sea transport of perfluoroalkyl acids (PFAAs) via sea spray aerosol. Although this is an interesting topic, the link to the work carried out in this study is unclear. I urge the authors to better describe this link or remove this reference.

**Author reply:**

This reference has been removed.

• Line 74: Would read better as "In a previous study, we observed that the transfer of short-chain organic acids between the air and sea through SSA may be influenced by seawater surface tension. This factor could, in turn, impact the enrichment behaviour of organic acids (Song et al., 2022)."

**Author reply:**

We have modified corresponding sentences in the revised manuscript.

Lines 72-74:

In a previous study, we observed that the transfer of short-chain organic acids between the air and sea through SSA may be influenced by seawater surface tension. This factor could, in turn, impact the enrichment behavior of organic acids (Song et al., 2022).

• Line 77: The authors state "...other factors were discussed in our recent review..." What are the other factors? If this discussion is important, then the authors should do it justice and include all relevant information. Otherwise, I see no need to mention this.

**Author reply:**

We have deleted this sentence in the Introduction of the revised manuscript.

• Line 78: There should be a new paragraph here: "In this study..."

**Author reply:**

We have modified it in the revised manuscript.


**Community comment:**

Dear Authors, editor, and reviewers,

This manuscript presents insights into the interactions between sea spray aerosols (SSAs) and aromatic acids through a laboratory study. This is an interesting and valuable research. I have some constructive comments on this manuscript.

Specific comments:

Abstract and Introduction:

Line 11, what is the importance of aromatic acids with SSAs in affecting the global radiative balance? In the following section of Introduction (Lines 29-40), the authors had ambiguous presentation in relationship between SSAs, aromatic acids, and global radiative balance.

**Author reply:**

As an important medium for exchange processes between ocean and the atmosphere, SSA directly affects the radiative balance of the Earth by scattering solar radiation and indirectly affects the global climate as a source of cloud condensation nuclei and ice nuclei (Wilson et al., 2015; Quinn et al., 2017; Rosenfeld et al., 2019; Croft et al., 2021). It is well known that the presence of surfactants affects the production of SSA, which in turn may affect the global radiation balance. This indicates that aromatic acids as a surfactant may also affect the global radiation balance. We have updated the Introduction section.

Lines 35-38:

Previous studies suggest that these organic acids can alter the composition of SSA, subsequently influencing atmospheric processes such as cloud condensation nuclei (CCN) or ice nuclei (IN) activities. SSA may contain organic acids that play a major role in Earth's climate (Moore et al., 2011; Zhu et al., 2019).

Lines 41-54: The authors introduced the current studies in seawater source of aromatic acids in this paragraph, that is, the majority is from esters released and biodegraded by algae and the remain is from anthropogenic emissions. Could aromatic acids be derived from precipitation? What are the major types and their proportions and concentrations

of aromatic acids in seawater? It should correspond to aromatic acids the authors used in the experiments (lines 90-98, 2 Experiment Section). More detailed description and possible summary are needed.

**Author reply:**

A previous study found benzoic acid in snow pit samples (Mochizuki et al., 2016), while the other aromatic acid samples were not studied in precipitation. The types, sources, and concentrations of the target aromatic acids are summarized in Table S4. The following content has been added to the revised manuscript.

Line 41-45:

In various observations, aromatic acids have been detected in both natural and anthropogenic sources (Zhao et al., 2019; Zangrando et al., 2019; Dekiff et al., 2014). Among them, natural sources of aromatic acids produced by algal releases account for most marine aromatic acids, especially benzoic acid, and most of hydroxybenzoic acids (Mostafa et al., 2017; Fotso Fondja Yao et al., 2010a; Castillo et al., 2023; Abdel-Hamid A. Hamdy, 2020).

Line 50-53:

This is consistent with previous researches that phthalic acid is primarily derived from anthropogenic sources (Ren et al., 2023a), whereas hydroxybenzoic acid has both anthropogenic and natural sources (Zhao et al., 2019; Castillo et al., 2023).

Line 89-90:

Further details are provided in Table S1, which lists sources and concentrations of these aromatic acid identified in seawater and atmospheric samples over the ocean.

**Table S1.** Sources and concentrations of aromatic acids identified in seawater and atmospheric samples over the ocean.

| Aromatic acids | Natural sources | Anthropogenic sources |
|---|---|---|
| benzoic acid | • sea algae (Abdel-Hamid A. Hamdy, 2020; Al-Zereini et al., 2010; Fotso Fondja Yao et al., 2010b; Liu et al., 2022b)
• sedimentary organic matter (10–65 µg g$^{-1}$) (Deshmukh et al., 2016)
• bacteria isolated from sea bass viscera (314 ppb) (Martí-Quijal et al., 2020)
• snow pit samples (2.11 ng g$^{-1}$) (Mochizuki et al., 2016) | • emerging endocrine disrupting compounds (34–491 ng L$^{-1}$) (Zhao et al., 2019)
• fuel combustion (Boreddy et al., 2017)
• industrial wastewater, automobile exhaust and tobacco smoke (Cuadros-Orellana et al., 2006) |
| *o*-phthalic acid | | • plasticizer (16.7–657 ng g$^{-1}$ d.w.) (Ren et al., 2023b; Sanjuan et al., 2023);
• plastic waste burning (8.3–84.9 ng m$^{-3}$) (Zhu et al., 2022)
• the end product of photochemical oxidation of SOA (15.5 ng m$^{-3}$) (Ding et al., 2021)
• biomass burning and fossil fuel combustion sources (0.4–7.9 ng m$^{-3}$) (Shumilina et al., 2023; Yang et al., 2020; Boreddy et al., 2022) |

| | | |
|---|---|---|
| *m*-phthalic acid | | • plasticizer (Ren et al., 2023b)
• the end product of photochemical oxidation of SOA (3.6 ng m$^{-3}$) (Ding et al., 2021)
• biomass burning and fossil fuel combustion sources (0.01–2.3 ng m$^{-3}$) (Yang et al., 2020; Boreddy et al., 2022; Kawamur, 2014) |
| *p*-phthalic acid | | • plasticizer (0.51–6.8 mg kg$^{-1}$) (Ren et al., 2023b; Di Giacinto et al., 2023; Di Renzo et al., 2021)
• plastic waste burning (10.8–80.7 ng m$^{-3}$) (Zhu et al., 2022); the end product of photochemical oxidation of SOA (4.3 ng m$^{-3}$) (Ding et al., 2021)
• biomass burning and fossil fuel combustion sources (0.05–2.5 ng m$^{-3}$) (Yang et al., 2020; Boreddy et al., 2022; Kawamur, 2014) |
| *o*-hydroxybenzoic acid | • sea algae (76.8 mg L$^{-1}$) (Castillo et al., 2023; Mostafa et al., 2017; Klejdus et al., 2017) | • pharmaceuticals and drugs of abuse (0.4–53.3 ng L$^{-1}$) (Alygizakis et al., 2016) |
| *m*-hydroxybenzoic acid | • sea algae (Al-Zereini et al., 2010; Castillo et al., 2023) | |

| | | |
|---|---|---|
| *p*-hydroxybenzoic acid | ● sea algae (57.7 mg $L^{-1}$) (Castillo et al., 2023; Klejdus et al., 2017; Tian et al., 2012; Hawas and Abou El-Kassem, 2017)
● sea fungus (Rukachaisirikul et al., 2010; Shao et al., 2007)
● sponge Mycale species (Xuefeng Zhou, 2013); metabolite (Jingchuan Xue, 2015; Liao and Kannan, 2018)
● sediment samples (6.85–437 ng $g^{-1}$ dw) (Liao et al., 2019) | ● Pharmaceuticals and personal care products (Lu et al., 2023)
● emerging endocrine disrupting compounds (4.58–49.9 ng $L^{-1}$) (Zhao et al., 2019; Lu et al., 2021; Alygizakis et al., 2016) |
| vanillic acid | ● sea algae (3–47 ng $L^{-1}$) (Zangrando et al., 2019; Klejdus et al., 2017)
● lignin decomposition (Wang et al., 2015; Hu et al., 2022; Xu et al., 2017) | ● combustion of both softwood and hardwood (Simoneit, 2022) |
| syringic acid | ● sea algae (0.3–0.6 ng $L^{-1}$) (Poznyakovsky et al., 2021; Zangrando et al., 2019; Klejdus et al., 2017)
● lignin decomposition (Hu et al., 2022; Xu et al., 2017) | ● pharmaceuticals (Fisch et al., 2017)
● hardwood burning (Simoneit, 2022) |

Line 80 Sintered glass filter and wave breaking method can also produce artificial SSAs with different properties of flux, chemical composition, size distribution, and so on. The authors should clarify the relationship between methods (sintered glass filter, plunging jet, and wave breaking) and possible efficiency of aromatic acid transport within SSAs. That is, whether the plunging jet method can effectively reflect the "Role of sea spray aerosol at the air-sea interface in transporting aromatic acids to the atmosphere"?

**Author reply:**

Combining the following parameters, we obtained a particle size distribution similar to that from field observations (Quinn et al., 2017; Xu et al., 2022), which is shown in Fig. S2. The experimental principle aligns with previous successful studies (Sha et al., 2021; Johansson et al., 2019). This enables us to consider that the plunging jet method can effectively reflect the transfer of aromatic acid at the air-sea interface.

The reasons for the selection of the relevant parameters are as follows:

1) A stainless steel nozzle with an inner diameter of 4.3 mm is used to generate plunging jets. The nozzle can be changed easily. The size of the nozzle is consistent with that in previous studies (Salter et al., 2014; Sha et al., 2021).

2) We have studied the cases with headspace heights of 14 cm, 19 cm, and 22 cm in our previous work (Liu et al., 2022a) and found that the SSA production was most efficient at a headspace height of 22 cm. Hence, we chose it for SSA generation.

3) We have tried pump flow rates of 0.50 L min$^{-1}$, 0.75 L min$^{-1}$, 1.00 L min$^{-1}$, 1.25 L min$^{-1}$, and 1.50 L min$^{-1}$ in our previous study (Liu et al., 2022a) and found that when the pump flow rate was 1 L min$^{-1}$ or higher, the performance of the SSA generator was good (see Table R1). Therefore, we used a pump flow rate of 1 L min$^{-1}$ to maintain a high SSA production rate and a relatively long working lifetime of the corresponding accessories.

Table R1. Effect of pump flow rate on SSA production.

| Pump flow rate (L min$^{-1}$) | SSA production (particles s$^{-1}$) | Mass concentration (μg m$^{-3}$) |
|---|---|---|
| 0.50 | $5.78 \times 10^6$ | 198.96 |
| 0.75 | $9.90 \times 10^6$ | 317.31 |
| 1.00 | $1.37 \times 10^7$ | 417.86 |
| 1.25 | $1.58 \times 10^7$ | 469.59 |
| 1.50 | $1.63 \times 10^7$ | 495.42 |

4) We have previously studied SSA production at purge air flow rates of 2 L min$^{-1}$, 3 L min$^{-1}$, 4 L min$^{-1}$, 5 L min$^{-1}$, 6 L min$^{-1}$ to simulate the sea breeze (Liu et al., 2022a). We found that the SSA production increased with the purge air flow rate when the flow rate was less than 3 L min$^{-1}$, and decreased with the increase of the purge air when the flow rate was more than 3 L min$^{-1}$.

5) Based on a previous study of the effect of relative humidity on the growth of sea salt particles, the inlet humidity would affect the morphology of SSA. (Tang et al., 1997). Hence, we chose to keep the relative humidity at about 40% in our study. Particles will deliquescence at higher humidity (Bryan et al., 2022). In addition, it has been demonstrated that acid has no effect on NaCl deliquescence (Ming and Russell, 2001).

[Figure]

**Fig. S2.** Number size distribution of SSA generated with the SSA simulation chamber in this study compared with field studies.

Experimental Section

Lines 90-97 Why do authors chose these types of aromatic acids? It should be clarified.
**Author reply:**
1) Molecular Structure:
Different aromatic acids have distinct properties, including molecular structure, polarity,

and reactivity. We aimed to investigate the position and number of functional groups and the effect of different functional groups on the transport of aromatic acids in SSA at the air-sea interface.

2) Biological Significance:

Most aromatic acids we selected are emerging endocrine-disrupting compounds with the potential for bioaccumulation in living organisms (Zhao et al., 2019). The associated human health risks should be of great concern.

3) Climate-Relevant Properties:

Aromatic acids may change the SSA acidity that make them particularly relevant to climate processes (Angle et al., 2021).

In summary, the selection of specific aromatic acids is likely driven by a combination of their molecular structure, biological significance, relevance to climate processes, and practical considerations for experimentation or observation.

Lines 93-94 The collection procedure of seawater is not clear (site, chemical composition of seawater, contamination control, storge conditions, filtration or not?) Detailed QA/QC description is needed.

**Author reply:**

We have indicated the collection procedure of seawater in the revised Supplement to make the experimental process clearer.

S1. Quality assurance/quality control.

Seawater was collected from the coastal area of Shazikou on March 27, 2023, with a volume of 500 L (Fig. S1). Considering the storage inconvenience caused by huge consumption of seawater, all our seawater was pre-filtered through a polyethersulfone filter (47 mm diameter, 0.2 μm pore size, Supor®-200, Pall Life Sciences, USA) and stored in the dark at 18 °C for less than one month. Quinn et al. (2015) have shown that the fraction that passes through the filter is regarded as dissolved organic carbon and includes colloidal and truly dissolved materials. For each experiment, we measured particle number concentrations generated by filtered seawater and cations concentrations in seawater, and we found good agreement between each set of experiments (see Fig. S2).

In order to avoid the influence of organic matter in quartz fiber filters and access the accuracy of the experiment, pre-baked quartz fiber filters were used in sampling. Before each set of experiments, experimental blanks were conducted using filtered

seawater. Experimental blanks were conducted with the same procedure of SSA samples. Seawater and filter samples were stored at -20°C until analyzed. In order to reduce the influence of organic acids residue after each experiment, the SSA simulation chamber was cleaned with ethanol first, then the system was cleaned with ultra-pure water for several times. The above steps also run the pump to allow for thorough cleaning of the system. Therefore, the system was blown with zero air and sealed for preservation. The Dekati DLPI was also ultrasonicated with methanol and water (V:V=1:1) and dried after the experiment.

Filtered seawater (without added aromatic acid) was used as the experimental blank, and the same experimental and analytical methods were used as those for the experimental samples. As a result, no target aromatic acid was found in both seawater and filters. This may be due to the fact that we did not perform any concentration operation during the seawater sample processing. The standard curves for each aromatic acid are linear, as shown in Fig. S5.

[Figure]

**Fig. S1.** Sampling site at Shazikou along the Yellow Sea coast, Qingdao, China.

[Figure]

**Fig. S2.** Number size distribution of SSA generated with the SSA simulation chamber in this study compared with field studies.

[Figure]

**Fig. S5.** The standard curves for aromatic acids were constructed within a concentration

range of 0.01-1000 μM, with more than seven data points.

Line 98: What is the mass concentration of typical aromatic acids in true seawater, which determines whether the concentration of aromatic acid used in the experiment (1 mM) is reasonable.

**Author reply:**

The concentration of aromatic acids in seawater ranges from 1 pM to 0.5 mM. To investigate the influence of concentration, we introduced a set of experiments using artificial seawater with a concentration of 1 μM for aromatic acids, and the specific experimental groups can be found in Table S3.

Lines 176-177:

The experiment consisted of a total of 25 sets with target compound concentrations of $10^{-3}$ and 1 mM (Table S3).

**Table S3.** Summary of experimental conditions.

| Exp. No. | Experiment type | Concentration (mM) | pH | Salinity (psu) | Sampling time (h) | RH (%) | Temperature difference (°C) [a] |
|---|---|---|---|---|---|---|---|
| 1 | SW | 0 | 7.92 | 34.2 | 5 | 35 | 2.0 |
| 2 | SW+benzoic acid | 1 | 7.72 | 34.3 | 5 | 34 | 1.5 |
| 3 | SW+$o$-hydroxybenzoic acid | 1 | 7.60 | 34.5 | 5 | 36 | 1.0 |
| 4 | SW+$m$-hydroxybenzoic acid | 1 | 7.68 | 34.1 | 5 | 40 | 2.0 |
| 5 | SW+$p$-hydroxybenzoic acid | 1 | 7.84 | 34.3 | 5 | 38 | 1.5 |
| 6 | SW+$o$-phthalic acid | 1 | 7.58 | 34.2 | 5 | 36 | 2.0 |
| 7 | SW+$m$-phthalic acid | 1 | 7.80 | 34.5 | 5 | 37 | 2.5 |
| 8 | SW+$p$-phthalic acid | 1 | 7.85 | 34.4 | 5 | 42 | 2.0 |
| 9 | SW+vanillic acid | 1 | 7.81 | 34.2 | 5 | 43 | 3.0 |
| 10 | SW+syringic acid | 1 | 7.84 | 34.3 | 5 | 39 | 2.0 |
| 11 | ASW | 0 | 7.96 | 35.1 | 5 | 33 | 1.5 |
| 12 | ASW+benzoic acid | 1 | 7.68 | 34.6 | 5 | 35 | 1.0 |
| 13 | ASW+$o$-hydroxybenzoic acid | 1 | 7.76 | 34.9 | 5 | 34 | 0.5 |
| 14 | ASW+$m$-hydroxybenzoic acid | 1 | 7.99 | 35.3 | 5 | 36 | 1.5 |
| 15 | ASW+$p$-hydroxybenzoic acid | 1 | 7.85 | 34.7 | 5 | 38 | 2.0 |
| 16 | ASW+$o$-phthalic acid | 1 | 7.93 | 34.5 | 5 | 35 | 1.0 |

| 17 | ASW+$m$-phthalic acid | 1 | 7.88 | 34.9 | 5 | 36 | 1.0 |
| 18 | ASW+$p$-phthalic acid | 1 | 7.97 | 34.6 | 5 | 34 | 1.5 |
| 19 | ASW+vanillic acid | 1 | 7.89 | 35.2 | 5 | 35 | 1.0 |
| 20 | ASW+syringic acid | 1 | 7.99 | 34.8 | 5 | 39 | 1.0 |
| 21 | ASW+benzoic acid+$o$-hydroxybenzoic acid+$o$-phthalic acid+vanillic acid+syringic acid | $10^{-3}$ | 7.95 | 35.1 | 20 | 41 | 3.5 |
| 22 | ASW+benzoic acid+$m$-hydroxybenzoic acid+$m$-phthalic acid+vanillic acid+syringic acid | $10^{-3}$ | 7.98 | 34.6 | 20 | 38 | 1.5 |
| 23 | ASW+benzoic acid+$p$-hydroxybenzoic acid+$p$-phthalic acid+vanillic acid+syringic acid | $10^{-3}$ | 7.88 | 34.9 | 20 | 40 | 2.0 |
| 24 | NaCl | 0 | 7.68 | 35.3 | 5 | 38 | 1.0 |
| 25 | NaCl+$m$-hydroxybenzoic acid | 1 | 7.54 | 34.7 | 5 | 36 | 1.5 |

[a] The temperature difference in the SSA simulation chamber before and after the experiment.

[Figure]

**Fig. S8.** Enrichment factors of aromatic acids at different concentrations from artificial seawater to the atmosphere.

Comparing the EF of aromatic acids in SW and ASW, it was observed that the EF trends of benzene dicarboxylic acids in seawater follows the pattern: *o*-phthalic acid < *m*-phthalic acid < *p*-phthalic acid. However, in ASW, the EF of *p*-phthalic acid was lower than that of *m*-phthalic acid. Based on the findings of Li et al. (2023), we hypothesize that in ASW, *p*-phthalic acid acts as an ·OH scavenger to produce TAOH. Hence, the EF of *p*-phthalic acid is lower than that of *m*-phthalic acid. Meanwhile, organic compounds in SW preferentially react with ·OH (Anastasio and Newberg, 2007), thus the EF of *p*-phthalic acid is the highest among the benzene dicarboxylic acids. Furthermore, differences in aromatic acid concentration did not change the enrichment pattern of organic acids.

Line 135 Can this flow rate of 1 mL min$^{-1}$ for 1 h collect particles? If so, possible sampling artifacts should be considered. I think this flow rate is too small and cannot achieve/tune with DKL-2.

**Author reply:**

This has been corrected. The sampling flow rate of the single particle sampler was set to 1 L min$^{-1}$, which was previously written incorrectly.

Lines 141-142:

The SSA particles impacted onto copper grids films (T11023, Tianld, China) with a

flow rate 1 L min⁻¹ for 1 h.

Results and discussion

Section 3.2.2 and Fig. 4

How quantitatively determine the thickness of organic coating of collected individual particles and what is the basis of the order? Can thickness of organic coating of selected individual particles represent the whole populations? Is there a statistic bias using only one typical TEM image?

**Author reply:**

We measured the thickness of most of the organic coating in the field of view, and the ordering was based only on the position of the functional groups of the aromatic acids, not on the thickness. We have modified corresponding figures in the revised manuscript.

[Figure]

**Fig. 4.** Particle morphology observed using TEM of sea salt (A) and mixed particles composed of aromatic acids-coated sea salt particles (B–K).

Section 3.3

How many samples were used to calculate the enrichment factors of aromatic acids and cations (both in Figures 5 and 6)? It should be summarized as a new table. The authors performed size-resolved filter-based experiments, what is the level of enrichment factors of aromatic acids in different size?

**Author reply:**

1) At least three filter samples should be used to calculate the enrichment factor of aromatic acids and cations. We have provided a summary in a new table.

2) In SSA, the ratios of inorganic salts and organic compounds vary with the particle size, which provides an insight into the size resolved enrichment of particulate mass (Bertram et al., 2018; Quinn et al., 2014). Sea salts account for a large fraction of super-micron aerosol particles. With the decrease of particle size, not only does the concentration of organic matter in submicron aerosol particles become higher, but the atmospheric residence time becomes longer (Ault et al., 2013; O'Dowd et al., 2004; Triesch et al., 2021). The global emission of submicron SSA is about 24 Tg y$^{-1}$, of which organic matter accounts for 8.2 Tg y$^{-1}$ (Meskhidze et al., 2011; Vignati et al., 2010). Although the understanding of organic compounds in submicron SSA has improved considerably in recent years, the transport process of organic acids as typical organic compounds in seawater remains poorly investigated. Then, we just collected all submicron SSA to calculate the enrichment factor.

**Table R3.** Enrichment factors for aromatic acids in submicron SSA.

| Aromatic acid | EF$_1$ | EF$_2$ | EF$_3$ | EF$_4$ | Average EF | Standard deviation |
|---|---|---|---|---|---|---|
| benzoic acid | 8.14 | 17.59 | 4.34 | 8.19 | 9.57 | 4.89 |
| o-hydroxybenzoic acid | 5.13 | 6.55 | 8.55 | | 6.74 | 1.72 |
| m-hydroxybenzoic acid | 29.72 | 29.13 | 15.21 | | 24.69 | 8.21 |
| p-hydroxybenzoic acid | 10.87 | 14.81 | 9.22 | 13.71 | 12.15 | 2.56 |
| o-phthalic acid | 4.89 | 6.14 | 6.89 | | 5.97 | 0.83 |
| m-phthalic acid | 7.80 | 6.70 | 5.68 | | 6.73 | 0.86 |
| p-phthalic acid | 11.53 | 15.28 | 12.75 | 16.80 | 14.09 | 2.39 |
| vanillic acid | 6.69 | 8.87 | 24.22 | | 13.26 | 2.55 |
| syringic acid | 23.50 | 34.38 | 15.51 | | 24.46 | 4.47 |

The authors hypothesize that the binding effect of $Ca^{2+}$ with aromatic acid is a reason for observed positive enrichment of aromatic acids. In fact, the positive calcium enrichment is also observed using artificial seawater without organic matters (Salter et al., 2016). Therefore, this hypothesis may need additional control experiments, for example, using pure NaCl solution with addition of aromatic acids, to excluded the effect of $Ca^{2+}$.

**Author reply:**

We supplemented two sets of control experiments by adding *m*-hydroxybenzoic acid to ASW and NaCl solution to exclude the effect of $Ca^{2+}$.

Lines 345-347:

Taking *m*-hydroxybenzoic acid as an example, we calculated the enrichment factors of $Ca^{2+}$ and *m*-hydroxybenzoic acid in NaCl solution, ASW and SW, and found that both follow the pattern: $EF_{NaCl} < EF_{ASW} < EF_{SW}$.

[Figure]

**Fig. S9**. Enrichment factors of $Ca^{2+}$ and *m*-hydroxybenzoic acid in submicron SSA with SW, ASW, and NaCl solution.

Lines 313-326 I found a missing detail in authors evaluate the cation enrichments, that is, the detailed mass concentration ratio of cations to $Na^+$, for example of $Ca^{2+}/Na^+$, in both seawater and aerosol samples is missing. I therefore suggest the authors supplement more details in mass concentration ratio of aromatic acids and cations in seawater and aerosol samples as a new table.

**Author reply:**

We give the values of enrichment factors for cations and aromatic acids, as well as their

peak areas. Since the standard curves are all linear, we directly used the peak area ratios to derive the enrichment factor values. We have also supplemented the standard curve as follows.

**Table R4.** Enrichment factors for cations in submicron SSA.

| Aromatic acid | Cation | $EF_1$ | $EF_2$ | $EF_3$ | $EF_4$ |
|---|---|---|---|---|---|
| benzoic acid | K | 0.56 | 0.57 | 0.56 | 0.54 |
| | Mg | 0.99 | 0.84 | 0.87 | 1.05 |
| | Ca | 1.33 | 1.27 | 1.08 | 1.04 |
| o-phthalic acid | K | 0.67 | 0.91 | 0.94 | |
| | Mg | 1.34 | 0.73 | 0.57 | |
| | Ca | 0.96 | 1.26 | 1.13 | |
| m-phthalic acid | K | 0.87 | 0.82 | 0.85 | |
| | Mg | 0.81 | 0.76 | 0.78 | |
| | Ca | 1.20 | 1.32 | 1.29 | |
| p-phthalic acid | K | 0.55 | 0.77 | 1.10 | 0.88 |
| | Mg | 0.68 | 0.73 | 0.93 | 0.86 |
| | Ca | 1.74 | 1.18 | 1.17 | 1.49 |
| o-hydroxybenzoic acid | K | 0.52 | 0.40 | 0.88 | |
| | Mg | 0.77 | 1.05 | 0.98 | |
| | Ca | 0.83 | 1.01 | 1.71 | |
| m-hydroxybenzoic acid | K | 0.90 | 0.71 | 1.03 | |
| | Mg | 0.77 | 0.75 | 0.78 | |
| | Ca | 1.97 | 1.74 | 2.00 | |
| p-hydroxybenzoic acid | K | 0.65 | 0.67 | 0.72 | 0.76 |
| | Mg | 0.88 | 0.81 | 0.74 | 0.76 |
| | Ca | 1.10 | 1.00 | 0.84 | 1.16 |
| vanillic acid | K | 0.91 | 0.63 | 0.77 | |
| | Mg | 0.79 | 0.78 | 1.01 | |
| | Ca | 1.95 | 1.64 | 0.94 | |
| syringic acid | K | 0.95 | 0.99 | 0.74 | |
| | Mg | 0.92 | 0.94 | 0.99 | |
| | Ca | 1.79 | 1.55 | 1.83 | |

**Table R5.** Peak areas of cations in seawater and SSA filter samples.

|  |  | SW/10000 | SSA/10 | SW/10000 | SSA/10 | SW/10000 | SSA/10 | SW/10000 | SSA/10 |
|---|---|---|---|---|---|---|---|---|---|
| benzoic acid | Na | 2.11 | 4.43 | 2.01 | 3.01 | 2.42 | 4.64 | 2.41 | 4.40 |
|  | K | 0.09 | 0.11 | 0.17 | 0.15 | 0.09 | 0.10 | 0.09 | 0.09 |
|  | Mg | 0.54 | 1.12 | 0.57 | 0.72 | 0.60 | 1.00 | 0.51 | 0.98 |
|  | Ca | 0.31 | 0.86 | 0.39 | 0.73 | 0.32 | 0.67 | 0.30 | 0.57 |
| o-phthalic acid | Na | 2.39 | 4.85 | 2.57 | 4.46 | 2.40 | 4.21 |  |  |
|  | K | 0.09 | 0.12 | 0.08 | 0.13 | 0.09 | 0.15 |  |  |
|  | Mg | 0.55 | 1.49 | 0.52 | 0.66 | 0.56 | 0.56 |  |  |
|  | Ca | 0.31 | 0.60 | 0.33 | 0.72 | 0.36 | 0.71 |  |  |
| m-phthalic acid | Na | 2.47 | 4.49 | 2.55 | 4.51 | 2.52 | 4.40 |  |  |
|  | K | 0.10 | 0.16 | 0.10 | 0.14 | 0.10 | 0.15 |  |  |
|  | Mg | 0.60 | 0.87 | 0.63 | 0.85 | 0.57 | 0.78 |  |  |
|  | Ca | 0.33 | 0.73 | 0.31 | 0.72 | 0.38 | 0.87 |  |  |
| p-phthalic acid | Na | 2.53 | 5.13 | 2.67 | 7.57 | 2.43 | 6.67 | 2.57 | 6.52 |
|  | K | 0.12 | 0.13 | 0.17 | 0.38 | 0.10 | 0.30 | 0.09 | 0.20 |
|  | Mg | 0.60 | 0.82 | 0.63 | 1.30 | 0.59 | 1.50 | 0.60 | 1.32 |
|  | Ca | 0.48 | 1.70 | 0.48 | 1.59 | 0.40 | 1.28 | 0.40 | 1.52 |
| o-hydroxybenzoic acid | Na | 2.96 | 5.57 | 2.70 | 6.80 | 2.72 | 6.90 |  |  |

| | | | | | | | | |
|---|---|---|---|---|---|---|---|---|
| | K | 0.09 | 0.09 | 0.11 | 0.11 | 0.11 | 0.25 | | |
| | Mg | 0.56 | 0.81 | 0.63 | 1.68 | 0.62 | 1.54 | | |
| | Ca | 0.30 | 0.47 | 0.32 | 0.82 | 0.29 | 1.24 | | |
| *m*-hydroxybenzoic acid | Na | 2.58 | 4.34 | 2.55 | 3.87 | 2.67 | 4.37 | | |
| | K | 0.10 | 0.15 | 0.10 | 0.11 | 0.10 | 0.17 | | |
| | Mg | 0.59 | 0.76 | 0.65 | 0.74 | 0.60 | 0.76 | | |
| | Ca | 0.38 | 1.27 | 0.36 | 0.94 | 0.36 | 1.19 | | |
| *p*-hydroxybenzoic acid | Na | 2.45 | 4.82 | 2.54 | 4.99 | 2.63 | 4.99 | 2.49 | 4.59 |
| | K | 0.09 | 0.12 | 0.10 | 0.13 | 0.09 | 0.12 | 0.10 | 0.14 |
| | Mg | 0.61 | 1.06 | 0.61 | 0.97 | 0.62 | 0.86 | 0.60 | 0.84 |
| | Ca | 0.48 | 1.03 | 0.49 | 0.96 | 0.46 | 0.74 | 0.41 | 0.88 |
| vanillic acid | Na | 2.53 | 5.23 | 2.75 | 5.06 | 2.68 | 5.06 | | |
| | K | 0.10 | 0.19 | 0.12 | 0.14 | 0.14 | 0.20 | | |
| | Mg | 0.56 | 0.92 | 0.69 | 0.99 | 0.59 | 1.13 | | |
| | Ca | 0.25 | 1.01 | 0.50 | 1.50 | 0.40 | 0.71 | | |
| syringic acid | Na | 2.39 | 4.98 | 2.81 | 5.14 | 2.99 | 4.70 | | |
| | K | 0.09 | 0.17 | 0.08 | 0.15 | 0.09 | 0.10 | | |
| | Mg | 0.51 | 0.99 | 0.41 | 0.71 | 0.58 | 0.91 | | |
| | Ca | 0.28 | 1.04 | 0.25 | 0.70 | 0.37 | 1.06 | | |

**Table R2.** Peak areas of aromatic acids in seawater and SSA filter samples.

| | SW$_1$ | SSA$_1$ | SW$_2$ | SSA$_2$ | SW$_3$ | SSA$_3$ | SW$_4$ | SSA$_4$ |
|---|---|---|---|---|---|---|---|---|
| benzoic acid | 159632 | 2732.77 | 196860 | 5206.99 | 185135 | 1540.1 | 167892 | 2508.47 |
| *o*-phthalic acid | 123785 | 1226.6 | 168974 | 1802.2 | 156324 | 1888.7 | | |
| *m*-phthalic acid | 466987 | 6606.85 | 458769 | 5432.25 | 491623 | 4880.15 | | |
| *p*-phthalic acid | 454593 | 10626.92 | 496102 | 21471.43 | 445985 | 15614.82 | 471582 | 20110.87 |
| *o*-hydroxybenzoic acid | 257548 | 2482 | 232946 | 3844.08 | 238358 | 5167.06 | | |
| *m*-hydroxybenzoic acid | 170482 | 8532.59 | 160309 | 7072.09 | 190181 | 4732.93 | | |
| *p*-hydroxybenzoic acid | 169753 | 3635.99 | 153257 | 4467.54 | 131186 | 2297.87 | 171062 | 4329.58 |
| vanillic acid | 357000 | 4948.01 | 287470 | 4696.83 | 279462 | 12750.11 | | |
| syringic acid | 486640 | 23833.92 | 467129 | 29328.75 | 491816 | 11993.31 | | |

[Figure]

**Fig. S5.** The standard curves for aromatic acids were constructed within a concentration range of 0.01-1000 μM, with more than seven data points.

Minor comments:

Lines 57: Please clarify "a potential route".

**Author reply:**

In our response to other previous Reviewers' comments, we have updated the related sentence. Lines 57-59:

One possible reason for the disappearance of these aromatic acids is their release into the atmosphere. Existing datasets obtained from remote marine areas offer evidence of the presence of these compounds in the atmosphere (Fu et al., 2010).

Lines 59-62: This sentence is hard to follow. What is that sea-salt included organic surfactants can thereby further disturbing ecological systems? The cited references also seem inappropriate.

**Author reply:**

We have rewritten this sentence.
Lines 61-63:

SSAs can act as carrier agents for the vertical transport of much more than just sea salt and often include organic surfactants in the ocean, as already shown by field and laboratory studies (Cochran et al., 2016; Franklin et al., 2022; Rastelli et al., 2017).

Line 109 What is the air flow rate? It should be different in different experiment procedures.

**Author reply:**

We have updated Fig. S2 to provide a clearer figure of the air flow rate during the experiment.

[Figure]

**Fig. S2.** Schematic picture of the plunging jet-sea spray aerosol generator: SMPS sampling (A), single particle sampling (B), and DeKati DLPI+ sampling (C). The red arrows represent the flow direction of seawater, and the purple arrows represent the flow of gases and aerosol particles.

Fig.3 It is hard to follow the differences in particle size, because these circles are almost identical.

**Author reply:**

We have labelled the particle sizes in Figure 3.

[Figure]

**Fig. 3.** SSA production, particle size, mass concentration distribution of aromatic acids. The symbol size represents the geometric mean diameter of SSA particles and is marked with numbers, and the symbol color indicates the particle mass concentration.

Fig.4 It is hard to follow the colors of subplots A, B, and C. The color span between groups should be larger.

**Author reply:**

We have updated the figure to expand the color span between the groups.

[Figure]

**Fig. 5.** Enrichment factors of benzene dicarboxylic acids (A), hydroxybenzoic acids (B), *p*-hydroxybenzoic acid, vanillic acid, and syringic acid (C) from seawater to the atmosphere.


To this end, we developed a sea spray aerosol simulation chamber using the plunging jet method, providing the closest proxy to natural SSA currently available in a controlled laboratory monitoring environment, and used it to probe the transport process of aromatic acids.

The surface tension of seawater was examined using a surface tensiometer (Sigma 700, Biolin Scientific, Sweden) equipped with a Wilhelmy plate, calibrated at 25 °C with 30 mL of ultrapure water.

Seawater containing p-phthalic acid ($73.92\pm0.14$ mN m$^{-1}$) exhibited the highest surface tension among the isomers of benzene dicarboxylic acids.

The number concentration of SSA particles with added benzene dicarboxylic acids was much higher than that with added benzoic acids.

The distributions show a decrease in the particle number concentrations when adding hydroxybenzoic acids, compared with the system with added benzoic acids, leading to the conclusion that hydroxybenzoic acids, acting as surfactants, could inhibit SSA production (Fig. 2C).

Importantly, Fig. 2D shows that the particle number concentration decreased proportionally with the increase of the number of methoxy groups, providing further ground that organic matter with hydrophobic functional groups have preferential atomization ability.

Notably, it is evident that the core morphology of salt particles underwent a significant change, with the cubic structure being transformed into a sphere structure.

Lines 301–303:

Figs. 4B show the morphology images of SSA particles containing hydroxybenzoic acids, where the core maintained the crystalline phase of sea salt with its cubic structure and the coatings mainly consisted of hydroxybenzoic acids.

Lines 323–324:

The EFs order of benzene dicarboxylic acids is opposite to that of surfactants, probably due to the amphiphilicity of *o*-phthalic acid.

2) In the Introduction section, the abbreviations are explained.

Lines 63–65:

Sea spray aerosols (SSAs), generated by breaking waves and bubble bursting, are one of the major sources of atmospheric particles (Andreae and Rosenfeld, 2008, Angle et al., 2021, Hasenecz et al., 2020, Malfatti et al., 2019).

Lines 36–40:

It has been shown that the addition of oleic acid as well as surfactants in artificial seawater can significantly enhance cloud condensation nuclei (CCN) activity (Moore et al., 2011). Previous studies suggested that these organic acids can alter the composition of SSA, subsequently influencing atmospheric processes such as CCN or ice nuclei (IN) activities (Moore et al., 2011, Zhu et al., 2019).

2) Structural Improvements:

The overall structure of the writing requires significant improvement. Taking the first paragraph as an example, the complexity of the topic results in a dense paragraph that is challenging to digest. Scientific writing often benefits from a one-idea-per-paragraph approach. Breaking down this paragraph into smaller, focused sections could enhance readability without sacrificing the depth of information. For instance, dividing it into two paragraphs—one emphasising the impact of acids on the environment and humans, and the other focusing on their potential climate impacts—would improve overall clarity. This approach should be taken throughout the manuscript

**Author reply:**

We thank the Reviewer for the very constructive comment. We have divided some of the paragraphs into two to improve the clarity. Lines 36–43, 116–126, and 224–238 were divided into three new paragraphs, each representing the influence of other factors on climate, the measurement of experimental basic parameters, and the effect of functional group position on SSA generation, respectively.

3) Figure Presentation:

The presentation of figures, such as Figure 7, falls below the expected standard. Figures should ideally be standalone, conveying information without heavy reliance on captions. In this case, the significance of the colour bars is unclear from both the plot and the caption, indicating a need for improved figure design and clarity. Readers need to understand the visual elements independently to enhance the overall effectiveness of the figures.

**Author reply:**

Thanks for the insightful comments on the figure presentation. Firstly, we have added Table 1 in an attempt to enhance the logical structure of the paper, providing readers with a clearer understanding of the structure of the article.

Lines 91–94:

Nine aromatic acids, including benzoic acid, phthalic acids (*o*-, *m*-, and *p*-isomers), hydroxybenzoic acids (*o*-, *m*-, and *p*-isomers), vanillic acid, and syringic acid, were investigated to determine the influence of functional group position and quantity on the transmission of aromatic acids at the sea-air interface (Table 1).

**Table 1.** Summary of aromatic acids used in experiments.

| Aromatic acids | | | |
|---|---|---|---|
| benzoic acid | *o*-phthalic acid | *m*-phthalic acid | *p*-phthalic acid |
| **Position of –COOH** | | | |
| benzoic acid | *o*-hydroxybenzoic acid | *m*-hydroxybenzoic acid | *p*-hydroxybenzoic acid |
| **Position of –OH** | | | |
| *p*-hydroxybenzoic acid | vanillic acid | syringic acid | |
| **Number of –CH₃O** | | | |

Furthermore, we have modified the corresponding figures in the manuscript. Firstly, we modified the TEM imagines to succinctly illustrate the influence of functional group positions and quantities on the coating of organic layers.

[Figure]

**Fig. 4. Particle morphology observed using TEM of sea salt particles, and benzenedicarboxylic acids- (A), hydroxybenzoic acids- (B), *p*-hydroxybenzoic acid-, vanillic acid-, and syringic acid-coated sea salt particles (C).**

Secondly, we revised the enrichment factor figure for aromatic acids by incorporating chemical structural formulas into the figures, aiming to illustrate more clearly the impact of aromatic acid structures on their enrichment at the sea-air interface.

[Figure]

**Fig. 5.** Enrichment factors of benzenedicarboxylic acids (A), hydroxybenzoic acids (B), *p*-hydroxybenzoic acid, vanillic acid, and syringic acid (C) from seawater to the atmosphere.

Then, we transformed the enrichment factor figure of cations into bar graphs to better illustrate the trend of each ion's variation.

[Figure]

**Fig. 6.** Enrichment factors of $K^+$, $Mg^{2+}$, and $Ca^{2+}$ in submicron SSA during the experiment.

Finally, we revised Fig. 7 by separating the concentration range figure of aromatic acids from the emission flux estimates, employing colors judiciously to present the content more intuitively to the readers.

[Figure]

**Fig. 7. Concentration range of aromatic acids in seawater (A) and the estimated range of annual global aromatic acids emission (tons yr⁻¹) via SSA (B). Yellow and blue stacked columns represent emissions based on Textor et al. (2006) and Jonas et al. (2021), respectively.**

In light of these issues, I regret to inform you that I cannot recommend the publication of this manuscript at this time. Addressing the highlighted concerns, particularly improving writing quality, enhancing structural organisation, and refining figure presentation, is crucial to meet what I view as the required standards for acceptance in ACP.

**Author reply:**

We thank the Reviewer for the constructive comments. We have comprehensively adjusted the writing quality, organizational structure, and graphical representation of the manuscript according to the reviewer's suggestions. We believe that the quality of the manuscript has significantly improved.

**Reviewer #3:**

I have the feeling that the authors answered many points from the critical review (2) quite good. The sea spray generation system and quality control are much better explained and the relevance of the aromatic amino acids is much clearer now. However, I have two big issues, that were also raised in the critical review:

1. I am surprized that no amino acids were originally detected in the seawater. I beliefe this is due to the insensitive analytical approach (LC-MS, without enrichment or sample preparation steps). Hence the water was spiked with a very high concentration of amino acids (1 mM) that is unrealistically high regarding seawater (typical concentrations of amino acids are in the low ng/L range). The fact that the experiments were done with very high spiked concentration is clearly a limitation of the study and should be pointed out more precisely.

**Author reply:**

We thank the Reviewer for the thoughtful comment. The absence of detection of aromatic acids in seawater is indeed attributed to the lack of enrichment of the seawater samples. The concentration of aromatic acids in seawater ranges from 1 pM to 0.5 mM (Table S1). To investigate the influence of concentration, we introduced a set of experiments using artificial seawater with a concentration of 1 µM for aromatic acids, and the specific experimental groups can be found in Table S3. We found a consistent enrichment trend of aromatic acids at both concentrations (Fig. S8).

And we have now pointed out this limitation in our manuscript.

Lines 103–106:

For simplified consideration, aromatic acids were added separately to artificial seawater to achieve concentrations of 1 µM and 1 mM, in order to verify the effects of background systems and concentrations on the EF of aromatic acids, which may not accurately reflect realistic conditions but provide an approximated trend of EFs instead.

**Table S1.** Sources and concentrations of aromatic acids identified in seawater and atmospheric samples over the ocean.

| Aromatic acids | Natural sources | Anthropogenic sources |
|---|---|---|
| benzoic acid | • sea algae (Abdel-Hamid A. Hamdy, 2020; Al-Zereini et al., 2010; Fotso Fondja Yao et al., 2010; Liu et al., 2022b)
• sedimentary organic matter (10–65 µg g$^{-1}$) (Deshmukh et al., 2016)
• bacteria isolated from sea bass viscera (0.3 µM) (Martí-Quijal et al., 2020)
• snow pit samples (2.11 ng g$^{-1}$) (Mochizuki et al., 2016) | • emerging endocrine disrupting compounds (0.3–4.0 nM) (Zhao et al., 2019)
• fuel combustion (Boreddy et al., 2017)
• industrial wastewater, automobile exhaust and tobacco smoke (Cuadros-Orellana et al., 2006) |
| *o*-phthalic acid | | • plasticizer (16.7–657 ng g$^{-1}$ d.w.) (Ren et al., 2023; Sanjuan et al., 2023);
• plastic waste burning (8.3–84.9 ng m$^{-3}$) (Zhu et al., 2022)
• the end product of photochemical oxidation of SOA (15.5 ng m$^{-3}$) (Ding et al., 2021)
• biomass burning and fossil fuel combustion sources (0.4–7.9 ng m$^{-3}$) (Shumilina et al., 2023; Yang et al., 2020; Boreddy et al., 2022) |
| *m*-phthalic acid | | • plasticizer (Ren et al., 2023)
• the end product of photochemical oxidation of SOA (3.6 ng m$^{-3}$) (Ding et al., 2021) |

| | | |
|---|---|---|
| | | • biomass burning and fossil fuel combustion sources (0.01–2.3 ng m$^{-3}$) (Yang et al., 2020; Boreddy et al., 2022; Kawamur, 2014) |
| *p*-phthalic acid | | • plasticizer (0.51–6.8 mg kg$^{-1}$) (Ren et al., 2023; Di Giacinto et al., 2023; Di Renzo et al., 2021)
• plastic waste burning (10.8–80.7 ng m$^{-3}$) (Zhu et al., 2022); the end product of photochemical oxidation of SOA (4.3 ng m$^{-3}$) (Ding et al., 2021)
• biomass burning and fossil fuel combustion sources (0.05–2.5 ng m$^{-3}$) (Yang et al., 2020; Boreddy et al., 2022; Kawamur, 2014) |
| *o*-hydroxybenzoic acid | • sea algae (0.5 mM) (Castillo et al., 2023; Mostafa et al., 2017; Klejdus et al., 2017) | • pharmaceuticals and drugs of abuse (2.8–385.9 pM) (Alygizakis et al., 2016) |
| *m*-hydroxybenzoic acid | • sea algae (Al-Zereini et al., 2010; Castillo et al., 2023) | |
| *p*-hydroxybenzoic acid | • sea algae (0.4 mM) (Castillo et al., 2023; Klejdus et al., 2017; Tian et al., 2012; Hawas and Abou El-Kassem, 2017)
• sea fungus (Rukachaisirikul et al., 2010; Shao et al., 2007)
• sponge Mycale species (Xuefeng Zhou, 2013); metabolite (Jingchuan Xue, 2015; Liao and Kannan, 2018)
• sediment samples (6.85–437 ng g$^{-1}$ dw) (Liao et al., 2019) | • Pharmaceuticals and personal care products (Lu et al., 2023)
• emerging endocrine disrupting compounds (0.03–0.4 nM) (Zhao et al., 2019; Lu et al., 2021; Alygizakis et al., 2016) |
| vanillic acid | • sea algae (0.02–0.3 nM) (Zangrando et al., 2019; Klejdus | • combustion of both softwood and hardwood |

| | | |
|---|---|---|
| | et al., 2017)

● lignin decomposition (Wang et al., 2015; Hu et al., 2022; Xu et al., 2017) | (Simoneit, 2022) |
| syringic acid | ● sea algae (1.5–3 pM) (Poznyakovsky et al., 2021; Zangrando et al., 2019; Klejdus et al., 2017)

● lignin decomposition (Hu et al., 2022; Xu et al., 2017) | ● pharmaceuticals (Fisch et al., 2017)

● hardwood burning (Simoneit, 2022) |

**Table S3.** Summary of experimental conditions.

| Exp. No. | Experiment type | Concentration (mM) | pH | Salinity (psu) | Sampling time (h) | RH (%) | Temperature difference (°C) [a] |
|---|---|---|---|---|---|---|---|
| 1 | SW | 0 | 7.92 | 34.2 | 5 | 35 | 2.0 |
| 2 | SW+benzoic acid | 1 | 7.72 | 34.3 | 5 | 34 | 1.5 |
| 3 | SW+*o*-hydroxybenzoic acid | 1 | 7.60 | 34.5 | 5 | 36 | 1.0 |
| 4 | SW+*m*-hydroxybenzoic acid | 1 | 7.68 | 34.1 | 5 | 40 | 2.0 |
| 5 | SW+*p*-hydroxybenzoic acid | 1 | 7.84 | 34.3 | 5 | 38 | 1.5 |
| 6 | SW+*o*-phthalic acid | 1 | 7.58 | 34.2 | 5 | 36 | 2.0 |
| 7 | SW+*m*-phthalic acid | 1 | 7.80 | 34.5 | 5 | 37 | 2.5 |
| 8 | SW+*p*-phthalic acid | 1 | 7.85 | 34.4 | 5 | 42 | 2.0 |
| 9 | SW+vanillic acid | 1 | 7.81 | 34.2 | 5 | 43 | 3.0 |
| 10 | SW+syringic acid | 1 | 7.84 | 34.3 | 5 | 39 | 2.0 |
| 11 | ASW | 0 | 7.96 | 35.1 | 5 | 33 | 1.5 |
| 12 | ASW+benzoic acid | 1 | 7.68 | 34.6 | 5 | 35 | 1.0 |
| 13 | ASW+*o*-hydroxybenzoic acid | 1 | 7.76 | 34.9 | 5 | 34 | 0.5 |
| 14 | ASW+*m*-hydroxybenzoic acid | 1 | 7.99 | 35.3 | 5 | 36 | 1.5 |
| 15 | ASW+*p*-hydroxybenzoic acid | 1 | 7.85 | 34.7 | 5 | 38 | 2.0 |
| 16 | ASW+*o*-phthalic acid | 1 | 7.93 | 34.5 | 5 | 35 | 1.0 |
| 17 | ASW+*m*-phthalic acid | 1 | 7.88 | 34.9 | 5 | 36 | 1.0 |

| 18 | ASW+$p$-phthalic acid | 1 | 7.97 | 34.6 | 5 | 34 | 1.5 |
|----|---|---|---|---|---|---|---|
| 19 | ASW+vanillic acid | 1 | 7.89 | 35.2 | 5 | 35 | 1.0 |
| 20 | ASW+syringic acid | 1 | 7.99 | 34.8 | 5 | 39 | 1.0 |
| 21 | ASW+benzoic acid+$o$-hydroxybenzoic acid+$o$-phthalic acid+vanillic acid+syringic acid | $10^{-3}$ | 7.95 | 35.1 | 20 | 41 | 3.5 |
| 22 | ASW+benzoic acid+$m$-hydroxybenzoic acid+$m$-phthalic acid+vanillic acid+syringic acid | $10^{-3}$ | 7.98 | 34.6 | 20 | 38 | 1.5 |
| 23 | ASW+benzoic acid+$p$-hydroxybenzoic acid+$p$-phthalic acid+vanillic acid+syringic acid | $10^{-3}$ | 7.88 | 34.9 | 20 | 40 | 2.0 |
| 24 | NaCl | 0 | 7.68 | 35.3 | 5 | 38 | 1.0 |
| 25 | NaCl+$m$-hydroxybenzoic acid | 1 | 7.54 | 34.7 | 5 | 36 | 1.5 |

[a] The temperature difference in the SSA simulation chamber before and after the experiment.

[Figure]

**Fig. S8.** Enrichment factors of aromatic acids at different concentrations from artificial seawater to the atmosphere.

2. As mentioned by the reviewer, the extrapolation to annual global amino acid emissions is very crude and also in the revised form, the effect of e.g. seawater temperature is not considered. This part should be overworked again.

**Author reply:**

We thank the Reviewer for the valuable comment and we agree that temperature has an impact on SSA emission fluxes. In this study, we have roughly estimated the annual emissions of aromatic acids at experimental temperature. Firstly, we greatly appreciate your input, however, due to the current study focuses on the enrichment behavior of structurally different aromatic acids under room temperature conditions and the influencing factors of their enrichment trend, I'm very sorry that we did not add additional experimental content at this time. Secondly, the enrichment process in field studies is much more complicated than in laboratory experiments. In field measurements for instance, seawater temperature affects SSA emission, while sea surface wind speed may affect SSA particle release and size, etc. Finally, the concentration of target compounds in seawater is a more significant contributor to the estimates of $k_{SSA}$ (Sha et al., 2021). Therefore, the effect of temperature is not discussed in detail in the current manuscript. Nevertheless, we will carefully consider your suggestions and explore them to the best of our ability in future work. And we point out in the manuscript the limitations of temperature in this manuscript.

Lines 369–373:

Furthermore, the enrichment process in the field is much more complicated than in controlled laboratory experiments. For example, seawater temperature affects SSA release, while wind speed at the sea surface may influence the amount and size of SSA particles emitted, etc. As such, further research on the environmental enrichment mechanism of aromatic acids in SSA is required to reduce the uncertainty in the estimation of aromatic acid emissions.

Finally, some more recent studies on amino acids in the marine environment could be included in this work.

**Author reply:**

We added some recent studies on aromatic acids in the marine and atmosphere environment in our manuscript.

Lines 40–41:

In addition, organic acids, such as carboxylic acids, aromatic acids also contribute significantly to ocean acidification (Kumari et al., 2022).

Lines 51–55:

This is consistent with previous studies that phthalic acid is primarily derived from anthropogenic sources, such as plasticizer, biomass burning and fossil fuel combustion (Boreddy et al., 2022, Ding et al., 2021, Ren et al., 2023, Sanjuan et al., 2023, Shumilina et al., 2023, Yang et al., 2020, Zhu et al., 2022), whereas hydroxybenzoic acid has both anthropogenic and natural sources (Castillo et al., 2023, Liao et al., 2019, Lu et al., 2021, Lu et al., 2023, Zhao et al., 2019).

Lines 61–62:

Although recent studies outline the critical sources of aromatic acids, much less is known about its transportation via SSA (Castillo et al., 2023, Hu et al., 2022).

Altogether I think that the manuscript could be published if these points are addressed (once more). I hope this helps, due to time constrains I am currently not able to provide a more detailed review.

**Author reply:**

Thank you for the very helpful comments from the reviewer. We have made corresponding revisions in the manuscript.

**References**

[revised manuscript text omitted]

We thank the Editor and Reviewers for their insightful comments. We have revised our manuscript according to the suggestions of the Reviewers' comments and our responses to the comments are as follows: Reviewers' comments are in black, authors' responses are in blue, and changes to the manuscript are in red color text.

**Editor decision:**

Comments (line number refer to the latest version of the manuscript w/o track changes):

- Line 36: SSA should be explained here at its first occurrence and not in line 63.

**Author reply:**

We have made the necessary revisions to ensure that SSA is properly explained at its first occurrence.

- Line 36-40: There is some repetition here with respect to CCN.

**Author reply:**

We have carefully reviewed the manuscript to eliminate any unnecessary repetition.

- Line 99: Figure or Table S1?

**Author reply:**

We have revised the manuscript accordingly, and the correct response is "Fig. S1."

- Line 124: Are the RH and T-ranges measured or technically given/fixed?

**Author reply:**

The RH and T ranges are technically given parameters in the experimental setup, and actual measurements fall within these specified ranges during the experiments.

- Line 130: I would give the mean and standard deviation of the measured RH (since you measured it).

**Author reply:**

The measured RH during the experiments was 34.2±3.9%. This has been updated in the revised manuscript.

- Line 133: particles -> particle

**Author reply:**

Revised.

- Line 136: Better "<20 cm$^{-3}$" (remove the hashtag)

**Author reply:**

The hashtag has been removed and replaced with "< 20 cm$^{-3}$".

- Line 155: Add "a" before "muffle"

**Author reply:**

We have added "a" before "muffle" accordingly.

- Beginning of Sect. 3.1: This should be part of the methods and not the result section.

**Author reply:**

We have moved the content to Sect. 2.3 as suggested.

- Table 1 caption: add "the" before "experiments"

**Author reply:**

We have revised the caption of Table 1 accordingly by adding "the" before "experiments".

- Figure 1: Please describe properly the box plot. What are the whiskers showing? How many data points are included in each box? 3 points is not enough (as suggested in the caption).

**Author reply:**

We have provided a more detailed description of the box plot in Fig. 1, specifying that each box contains 14 data points. The whiskers represent the maximum and minimum values, respectively.

Lines 685–688:

Fig. 1. Measured surface tension values of natural seawater and aromatic acid-containing seawater: benzoic acids (A), benzenedicarboxylic acids (B), hydroxybenzoic acids (C), *p*-hydroxybenzoic acid, vanillic acid, and syringic acid (D). The dark spots represent the mean values of at least 9 data points, the boxes represent the ranges of 25th−50th−75th percentiles, and the whiskers represent the maximum and minimum values.

- Figure 2: "SW" is not properly defined. Is it the ambient sea water or the artificial sea

water (which in the text is abbreviated as ASW). Please make sure that all abbreviations are properly defined.

**Author reply:**

"SW" refers to the ambient seawater in our study. We apologize for any confusion caused by the abbreviation. We have defined all abbreviations properly throughout the manuscript.

Lines 690–692:

Fig. 2. Number concentration distribution of sea salt particles and SSA particles containing benzoic acids (A), benzenedicarboxylic acids (B), hydroxybenzoic acids (C), *p*-hydroxybenzoic acid, vanillic acid, and syringic acid (D). SW represents natural seawater.

- Figure 3: Define "SW". It is really hard to see difference in the circle size. In the caption, make clear that "numbers above the points give the geometric mean diameter (in nm)". Errorbars are standard deviation?

**Author reply:**

In Figure 3, "SW" refers to seawater. We have made sure to define this abbreviation in the figure caption. Regarding the circle size, we have improved to enhance the visibility of the differences. Additionally, we have clarified in the caption that the numbers below or above the points represent the geometric mean diameter in nanometers (nm). The error bars in the figure represents the standard deviation. We have made these clarifications in the revised manuscript.

Lines 693–697:

[Figure]

Fig. 3. SSA production, particle size, mass concentration distribution of aromatic acids. The symbol size represents the geometric mean diameter of SSA particles, with the numbers below or above the points giving the geometric mean diameter (in nm), and the error bars are standard deviation. The symbol color indicates the particle mass concentration, with SW representing natural seawater.

- Figure 7: Harmonize if you use "tons" or "tonnes" within the figures and text.

**Author reply:**

We have made the necessary adjustments and made usage of "tons" both within the figures and the text.

Lines 708–711:

[Figure]

Fig. 7. Concentration range of aromatic acids in seawater (A) and the estimated range of annual global aromatic acids emission (tons yr$^{-1}$) via SSA (B). Yellow and blue stacked columns represent emissions based on Textor et al. (2006) and Gliss et al. (2021), respectively.

- Supplement:

o All figures and tables in the SI need to be also referenced in the main text. I could not find S5, S9 and S10 in the main text! Otherwise remove them.

**Author reply:**

We have included references to all SI figures and tables in the main text accordingly:

Line 96: Table S1, Line 112: Table S2, Lines 130–131: Table S3, Line 326: Table S4, Line 364: Table S5, Line 380: Table S6.

Lines 100–101: Fig. S1, Line 109: Fig. S2, Line 117: Fig. S3, Line 168: Fig.S4, Lines 194 and 218: Fig. S5, Line 223: Fig. S6, Line 256: Fig. S7, Lines 282, 299, and 310: Fig. S8, Lines 321 and 345: Fig. S9, Line 361: Fig. S10.

o Figure S5: Please increase font size and check the y-labels.

**Author reply:**

We have increased the font size and reviewed the y-labels.

[Figure]

**Fig. S4.** Standard curves for aromatic acids were constructed within a concentration range of 0.01–1000 μM, with more than seven data points.

o Suggest to use normal page numbers and not S1, S2, etc. to not confuse with the figure and table labelling.

**Author reply:**

We have revised the page numbering to use normal page numbers.

- Data availability: I strongly recommend to make the data publicly available and follow the data policy of ACP (https://www.atmospheric-chemistry-and-physics.net/policies/data_policy.html).

**Author reply:**

We have arranged the data to be made publicly available and to comply with the data policy of ACP.

Lines 408–409:

The data used in this study can be found online at https://doi.org/10.5281/zenodo.10903140 (Song et al., 2024).

**Reviewer #2:**

I want to thank the authors for their persistence with this manuscript. I think their efforts have borne fruit. I believe the manuscript is now close to achieving the quality required for publication in ACP. However, there are still a few presentation issues that the authors would be advised to resolve. For example, there are several instances where the authors should have begun a new paragraph. An example of this can be found in line 55 where the authors should start a new paragraph with "Recent laboratory studies have shown...".

**Author reply:**

We have addressed the presentation issues raised by the Reviewer, including starting new paragraphs at line 55 with "Recent laboratory studies have shown...", at line 69 with "Recent data indicate that the surface activity...", and at line 212 with "Unlike the order of seawater surface tension…".

Also, there are several instances of incorrect referencing. For example, in Line 376 it should be "Gliss et al. (2021)" rather than "Jonas et al. (2021)". I would urge the authors to double-check all references. The authors would be wise to read thoroughly through the entire manuscript and double-check the presentation.

**Author reply:**

We have carefully reviewed the entire manuscript to ensure correct referencing. We have also read through the entire manuscript and double-check the presentation.

**Reviewer #3:**

The revised version has improved a lot and is in my opinion suitable for publication after the following minor issues are addressed:

Line 57 et al.: The concept about the "missing aromatic acids" is not clear to me and is not evident from the part "recent laboratory studies have shown that personal-care products, especially sunscreen (e.g., o-hydroxybenzoic acid), are reduced in levels during algal blooms (Franklin et al., 2022). " Please clarify.

**Author reply:**

Our intention to use "missing aromatic acids" was to highlight that the levels of o-hydroxybenzoic acid, commonly found in sunscreen products, decrease during algal blooms, but the reasons for this reduction are not yet clear. We speculate that these missing aromatic acids may be transported to the atmosphere through SSA.

Lines 56–57:

Recent laboratory studies have shown that personal-care products in SW, especially sunscreen (e.g., *o*-hydroxybenzoic acid), are reduced in levels during algal blooms (Franklin et al., 2022).

The language has strongly improved, however, there are still articles missing (in the newly introduced parts), such as

Line 69: "The" is missing: Moreover, "the" molecular structure

Line 79: to "the" molecular structure

Line 287: of "the" salt particles

Line 342: by "the" compound concent

Please check again carefully.

**Author reply:**

Thank you for pointing out those missing articles. We have revised them in the current version of the manuscript.